# NOT SEARCH, BUT SCAN: BENCHMARKING MLLMS ON SCAN-ORIENTED ACADEMIC PAPER REASONING

**Rongjin Li, Zichen Tang, Xianghe Wang, Xinyi Hu, Zhengyu Wang, Zhengyu Lu, Yiling Huang, Jiayuan Chen, Weisheng Tan, Jiacheng Liu, Zhongjun Yang, Haihong E**[*]
Beijing University of Posts and Telecommunications

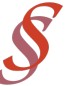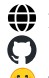

🌐 https://bupt-reasoning-lab.github.io/ScholScan
🔘 https://github.com/BUPT-Reasoning-Lab/ScholScan
🤗 https://huggingface.co/datasets/BUPT-Reasoning-Lab/ScholScan

## ABSTRACT

With the rapid progress of multimodal large language models (MLLMs), AI already performs well at literature retrieval and certain reasoning tasks, serving as a capable assistant to human researchers, yet it remains far from autonomous research. The fundamental reason is that current work on academic paper reasoning is largely confined to a search-oriented paradigm centered on pre-specified targets, with reasoning grounded in relevance retrieval, which struggles to support researcher-style full-document understanding, reasoning, and verification. To bridge this gap, we propose **ScholScan**, a new benchmark for academic paper reasoning. ScholScan introduces a scan-oriented task setting that asks models to read and cross-check entire papers like human researchers, scanning the document to identify consistency issues. The benchmark comprises 1,800 carefully annotated questions drawn from nine error categories across 13 natural-science domains and 715 papers, and provides detailed annotations for evidence localization and reasoning traces, together with a unified evaluation protocol. We assessed 15 models across 24 input configurations and conducted a fine-grained analysis of MLLM capabilities for all error categories. Across the board, retrieval-augmented generation (RAG) methods yield no significant improvements, revealing systematic deficiencies of current MLLMs on scan-oriented tasks and underscoring the challenge posed by ScholScan. We expect ScholScan to be the leading and representative work of the scan-oriented task paradigm.

## 1 INTRODUCTION

Enabling multimodal large language models (MLLMs) (OpenAI, 2025a; Anthropic, 2025; ByteDance Seed Team, 2025; Meta, 2025; xAI, 2025) to conduct comprehensive understanding and generation based on academic literature is the ultimate goal of Deep Research (Comanici et al., 2025), and a critical milestone on the path toward artificial general intelligence (AGI) (Ge et al., 2023; Morris et al., 2024; Phan et al., 2026). With rapid advances, MLLMs are increasingly capable of supporting academic workflows through retrieval, reading, and writing. For example, PaSa (He et al., 2025) can invoke a series of tools to answer complex academic queries with high-quality results, while Google Deep Research (Comanici et al., 2025) is capable of producing human-level research reports based on specific queries.

However, most of the existing work still follows *a search-oriented paradigm*, where models retrieve a few relevant passages and reason over local evidence based on pre-specified targets (Gao et al., 2024; Lou et al., 2025). Such methods are effective for tasks with clearly pre-defined targets, but struggle with researcher-style full-document reasoning and verification (Zhou et al., 2024). ***To function as researchers, models must move beyond reactive question answering and toward proactive discovery of implicit problems.***

To fill this gap, as shown in Figure 1, we introduce *a scan-oriented paradigm*, where models **address queries with absent targets**, requiring them to **exhaustively scan papers, actively construct**

---
[*]Corresponding author.

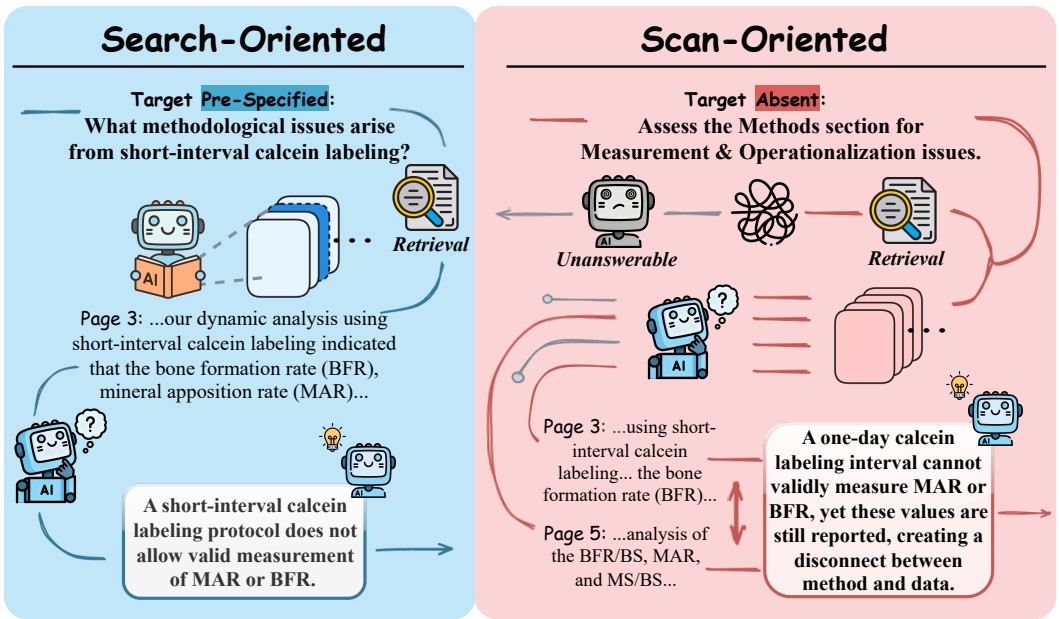

Figure 1: A comparison between *search-oriented* and *scan-oriented* task paradigms. Unlike the former, the scan-oriented paradigm provides no pre-specified targets, requiring the model to actively scan the entire paper and construct a document-level evidence view.

**a document-level evidence view, and perform evidence-based reasoning**. In contrast to search-oriented tasks that assess the model's ability to identify and reason over *relevant* fragments, scan-oriented tasks emphasize *consistency*. *Instead of relying on pre-specified targets or hints, models must derive all necessary concepts and inferences solely from given documents.*

We instantiate this setting via scientific error detection, as it naturally demands discovering non-obvious flaws without target cues, and present **ScholScan**, a new multimodal benchmark for academic reasoning. ScholScan features the following key highlights:

- **Scan-Oriented Task Paradigm.** In ScholScan, models receive one or more complete academic papers with target-absent queries, undergoing rigorous evidence-based reasoning evaluation. The benchmark comprises 715 papers spanning 13 natural science disciplines.
- **Comprehensive Error Taxonomy.** ScholScan covers nine categories of scientific errors throughout the research workflow, including citation and reference errors, rigorously evaluating models' cross-source reasoning abilities.
- **Process-Aware Evaluation Framework.** ScholScan provides fine-grained annotations for both evidence location and reasoning steps, enabling a comprehensive evaluation framework that assesses model performance in terms of both process and outcome.

We evaluated 15 models across 24 input configurations and 8 retrieval-augmented generation (RAG) frameworks. All models exhibited limited performance, and none of the RAG methods delivered significant improvements. These results highlight the inadequacy of search-oriented frameworks when applied to scan-oriented tasks and underscore both the challenges and the potential of enabling MLLMs to perform reliable document-level reasoning over full academic papers.

## 2 RELATED WORK

### 2.1 MULTIMODAL LARGE LANGUAGE MODELS

With the rapid progress of MLLMs, the models have evolved beyond perception tasks (e.g., image recognition and explanation) (Liu et al., 2024) toward a deep understanding of structured multimodal long documents. Their strengths lie in the ability to integrate cross-modal information and perform multi-hop reasoning over extended contexts. These capabilities are not only valuable for specific

question answering or instruction-following tasks (Yue et al., 2024) but are particularly well suited to simulating human thought processes and generating explainable reasoning trajectories (Zheng et al., 2023). Consequently, achieving a comprehensive understanding of entire documents has emerged as a core challenge that MLLMs are inherently equipped to address.

## 2.2 DOCUMENT UNDERSTANDING BENCHMARK

Document understanding tasks challenge models to identify the relevant context and perform accurate reasoning grounded in that information. Progress in document understanding benchmarks has followed two main axes. Across the input dimension, it has evolved from short to long contents, from everyday to specialized domains, and from plain text to multimodal formats (Chen et al., 2021; Yang et al., 2018; Mathew et al., 2021; Deng et al., 2025). Across the output dimension, it has shifted from limited-output formats to more open-ended responses (Pramanick et al., 2024). DocMath-Eval$_{CompLong}$ (Zhao et al., 2024) evaluates numerical reasoning on long specialized documents, while MMLongBench-Doc (Ma et al., 2024) builds a multimodal benchmark with layout-rich documents. The recently proposed FinMMDocR (Tang et al., 2025) pioneers the integration of scenario awareness, document understanding, and multi-step reasoning in financial scenarios. However, a comprehensive benchmark that integrates all the above challenges has yet to be introduced.

## 2.3 ACADEMIC PAPER UNDERSTANDING BENCHMARK

Compared with general documents, academic papers are distinguished by their rich domain knowledge and logical rigor. Reasoning over papers has emerged as a major challenge in recent research. Some studies request local elements such as charts or snippets, leveraging their internal complexity, but neglect the need for cross-source integration and domain-specific interpretation within the full document (Wang et al., 2024; Li et al., 2024). Recent studies extends to document-level inputs using image-based formats for real-world reading scenarios (Auer et al., 2023; Yan et al., 2025). However, benchmarks based on the QA paradigm face inherent limitations, as they typically presuppose the existence of answers and embed explicit cues in the question itself, reducing the need for comprehensive understanding and information organization. Moreover, mainstream evaluation protocols focus on the final outcome, with limited assessment of whether intermediate reasoning is evidentially grounded and logically valid. More examples and analysis are shown in Appendix B.

## 3 SCHOLSCAN BENCHMARK

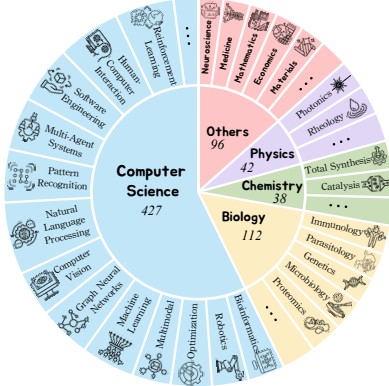

| Benchmark | Mod. | Para. | Eval. | # Dom. |
|---|---|---|---|---|
| *Document Understanding* | | | | |
| DocMath-Eval$_{CompLong}$ | T+TD | Search | O | N/A |
| FinMMDocR | T+MD | Search | O | N/A |
| MMLongBench-Doc | T+MD | Search | O | N/A |
| LongDocURL | T+MD | Search | O | N/A |
| SlideVQA | T+MD | Search | O | N/A |
| *Academic Paper Understanding* | | | | |
| CharXiv | T+I | Search | O | 8 |
| ArXivQA | T+I | Search | O | 10 |
| MMCR | T+MD | Search | O | CS |
| AAAR-1.0 | T+MD | Search | O | CS |
| **ScholScan (ours)** | **T+MD** | **Scan** | **P+O** | **13** |

Figure 2: **Left**: Overview of ScholScan. **Right**: Comparison to related benchmarks. **Mod.**: Modalities; **Para.**: Task Paradigm; **Eval.**: Evaluation Focus; **T**: Text; **I**: Image; **TD**: Text Document; **MD**: Multimodal Document; **P**: Process; **O**: Outcome; **Dom.**: Academic Domain Coverage.

## 3.1 OVERVIEW OF SCHOLSCAN

We introduce ScholScan, a benchmark designed to comprehensively evaluate MLLMs' ability to detect scientific flaws in academic papers under scan-oriented task settings. As illustrated in Figure 2,

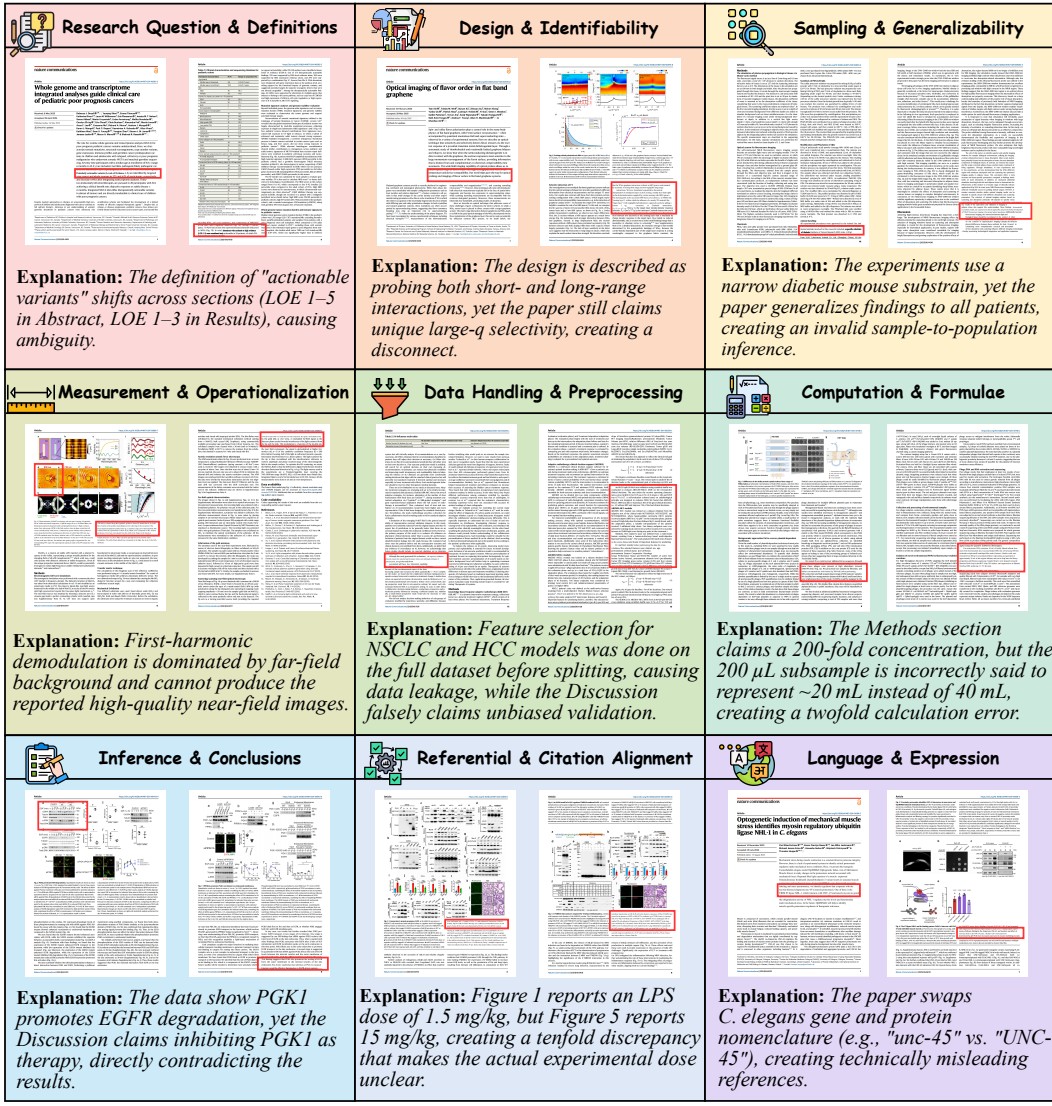

Figure 3: Sampled ScholScan examples with 9 error categories, covering the whole process of scientific research, each requiring the model to perform thorough cross-source evidence-based reasoning.

ScholScan spans 13 disciplines in the natural sciences, including physics, chemistry, and computer science, and encompasses over 100 subfields such as immunology, total synthesis, and machine learning. The benchmark comprises 1,800 questions derived from 715 real academic papers and covers nine major error categories commonly observed in real-world research scenarios (see examples in Figure 3; more examples in Appendix D). These include issues in numerical and formulaic computation, experimental design, inference and conclusion, and citation misuse, among others. Figure 2 also compares ScholScan with existing document and paper understanding benchmarks.

## 3.2 DATA CURATION AND QUESTION GENERATION

We curated papers from ICLR 2024[1] and 2025[2], as well as Nature Communications[3], collecting public reviews for the former. Questions were constructed based on two dimensions, where the source is either generated or sampled, and the context is either within-paper or cross-paper.

---

[1] https://openreview.net/group?id=ICLR.cc/2024/Conference
[2] https://openreview.net/group?id=ICLR.cc/2025/Conference
[3] https://www.nature.com/ncomms/

**Generation.** On high-quality accepted papers, we prompt Gemini 2.5 Pro to perform coordinated sentence-level edits spanning multiple sections or pages. It then synthesizes composite errors and generates the corresponding question along with an explanation grounded in the edited context.

**Sampling.** From rejected ICLR submissions and their public reviews, we prompt Gemini 2.5 Pro to extract explicit, falsifiable scientific errors and convert them into questions with initial explanations. Subjective remarks on novelty or writing quality are excluded.

**Within-Paper.** This setting focuses on verifiable facts and internal consistency within a single paper, and supports both **Generation** and **Sampling**.

**Cross-Paper.** This setting examines citation consistency across papers. For each instance, Gemini 2.5 Pro receives an accepted paper and one of its cited sources, then edits the accepted paper to introduce paraphrases or reasoning errors about the citation. As public reviews mainly address nonfalsifiable aspects, such as appropriateness, all cross-paper instances are constructed exclusively using the generation method.

The detailed prompt templates and specific instructions used are provided in Appendix A.

### 3.3 Quality Control and Annotation

Despite explicit instructions, initial outputs exhibited substantial hallucinations, logical inconsistencies, and low-quality questions. To ensure quality, 10 domain experts conducted a rigorous annotation process. Each instance underwent independent dual review, and disagreements were resolved by a third expert. Among the 3,500 initial candidates, 1,700 were discarded, and 1,541 of the remaining were revised, including 535 question rewrites, 1,207 explanation edits, and 1,141 corrections to error categories or metadata. Appendix C details data sourcing and validation protocols.

## 4 Experiments

### 4.1 Experimental Setup

**Models.** We benchmark a total of 24 input configurations by feeding academic papers as images or OCR-based text using the Tesseract engine (Smith, 2007), covering 15 mainstream models (Yang et al., 2025; Bai et al., 2025; DeepSeek-AI et al., 2025; Guo et al., 2025; OpenAI et al., 2025).

**Evaluation Protocol.** Inspired by MMLongBench-Doc (Ma et al., 2024), we prompt the models to generate the necessary reasoning chains from evidence to detected anomalies without constraining the output format. This design assesses evidence-based reasoning ability rather than basic instruction following. For open-ended responses, we use GPT-4.1 (OpenAI, 2025b) to extract cited evidence and reasoning steps, and quantify alignment with annotated explanations. Human evaluation confirms high agreement between our pipeline and expert annotations. Human validation results and evaluator robustness checks are detailed in Appendix E.

**Metrics.** We define a structured evaluation framework by parsing the model response $a$ into a tuple:

$$\Psi(a) \implies \left( \mathbf{1}_{\text{exist}}, \mathbf{1}_{\text{contain}}, \widehat{\mathcal{E}}, \widehat{\mathcal{R}}, n \right). \tag{1}$$

Here, $\mathbf{1}_{\text{exist}}$ and $\mathbf{1}_{\text{contain}}$ are binary indicators for whether output contains any error and includes the annotated target error; $\widehat{\mathcal{E}}, \widehat{\mathcal{R}}$ and $\mathcal{E}^*, \mathcal{R}^*$ are the predicted and gold evidence sets and reasoning chains; $\hat{g} = \text{prefix\_match}(\widehat{\mathcal{R}}, \mathcal{R}^*)$ counts matched reasoning steps; $n \in \mathbb{N}$ is the number of unrelated errors. $\mathbf{1}_{\text{contain}}$ is 1 if the output contains any predicted error, and 0 otherwise. Based on $\Psi(a)$, we define an end-to-end score $S(a) \in [0, 1]$ that combines all aspects of prediction quality:

*(i) Error Detection Score.* We consider the error detected only if the model identifies the target error:

$$S_{\text{detection}} = \mathbf{1}_{\text{exist}} \cdot \mathbf{1}_{\text{contain}}. \tag{2}$$

*(ii) Evidence Location Score.* Even when the target error is identified, the cited evidence may be incomplete or noisy. We compute a Dice score with a squared penalty for over-reporting:

$$S_{\text{location}} = \max\left\{ 0, \frac{2\left|\widehat{\mathcal{E}} \cap \mathcal{E}^*\right| + \mathbf{1}\left\{ |\widehat{\mathcal{E}}| + |\mathcal{E}^*| = 0 \right\}}{\max\left( |\widehat{\mathcal{E}}| + |\mathcal{E}^*|, 1 \right)} - 0.8 \left( \frac{|\widehat{\mathcal{E}} \setminus \mathcal{E}^*|}{\max\left( |\widehat{\mathcal{E}}|, 1 \right)} \right)^2 \right\}. \tag{3}$$

*(iii) Reasoning Process Score.* Even if the target error is detected, the reasoning may diverge from the gold chain. We use prefix match to assess reasoning completeness:

$$S_{\text{reasoning}} = \mathbf{1}\{\,|R^*| = 0\,\} + \mathbf{1}\{\,|R^*| > 0\,\} \left(\frac{\hat{g}}{|R^*|}\right)^2. \tag{4}$$

*(iv) Unrelated-Error Penalty.* Models may list unrelated items to inflate recall at the cost of precision. We penalize this with a rapidly increasing function of unrelated error count:

$$P_{\text{unrelated\_err}}(n) = 0.9^{\min(n,2)} \exp\!\Big(-0.6 \left[\max(n-2,0)\right]^{1.5}\Big). \tag{5}$$

*(v) Overall Score.* The final score for response $a$ integrates detection accuracy, evidence quality, and reasoning faithfulness:

$$S(a) = S_{\text{detection}} \cdot \sqrt{S_{\text{location}} \cdot S_{\text{reasoning}}} \cdot P_{\text{unrelated\_err}}(n). \tag{6}$$

Table 1: Model performance across 9 error categories (scaled by 100). **RQD**: Research Question & Definitions; **DI**: Design & Identifiability; **SG**: Sampling & Generalizability; **MO**: Measurement & Operationalization; **DHP**: Data Handling & Preprocessing; **CF**: Computation & Formulae; **IC**: Inference & Conclusions; **RCA**: Referential & Citation Alignment; **LE**: Language & Expression.

| Model | Avg. | RQD | DI | SG | MO | DHP | CF | IC | RCA | LE |
|---|---|---|---|---|---|---|---|---|---|---|
| **MLLM (Image Input)** | | | | | | | | | | |
| *Proprietary MLLMs* | | | | | | | | | | |
| GPT-5 | **19.2** | 10.1 | 9.7 | 28.2 | **14.6** | 26.6 | **13.8** | 25.3 | 25.3 | 6.9 |
| Gemini 2.5 Pro | 15.6 | **11.9** | **12.6** | 35.7 | 12.3 | **27.0** | 4.6 | 14.7 | 15.2 | **7.4** |
| Doubao-Seed-1.6-thinking | 10.2 | 3.4 | 3.5 | 22.3 | 7.5 | 15.1 | 10.2 | 12.2 | 10.9 | 3.3 |
| Doubao-Seed-1.6 | 9.9 | 3.0 | 4.4 | 29.2 | 4.9 | 15.0 | 6.3 | 17.9 | 8.0 | 3.9 |
| Grok 4 | 4.0 | 0.0 | 1.9 | 16.7 | 3.2 | 7.4 | 0.7 | 1.9 | 3.6 | 0.0 |
| *Open-Source LLMs* | | | | | | | | | | |
| Llama 4 Maverick | 7.0 | 7.0 | 7.3 | 9.4 | 4.5 | 4.0 | 6.5 | 6.7 | 8.8 | 3.0 |
| Mistral Small 3.1 | 3.3 | 0.1 | 2.0 | 2.0 | 1.5 | 0.1 | 1.0 | 2.2 | 8.6 | 1.0 |
| Gemma 3 27B | 1.7 | 0.5 | 2.7 | 2.3 | 1.7 | 1.0 | 1.0 | 1.3 | 2.6 | 0.0 |
| Qwen2.5-VL-72B | 0.1 | 0.0 | 0.7 | 0.0 | 0.0 | 0.0 | 0.0 | 0.0 | 0.2 | 0.0 |
| **OCR + LLM (Text Input)** | | | | | | | | | | |
| *Proprietary LLMs* | | | | | | | | | | |
| Gemini 2.5 Pro | **30.3** | **21.5** | **34.2** | **44.3** | **27.6** | **56.6** | 10.3 | 28.8 | **35.6** | **8.1** |
| GPT-5 | 22.5 | 16.1 | 21.4 | 26.0 | 20.3 | 36.7 | 4.7 | **29.8** | 30.0 | 2.6 |
| Grok 4 | 20.8 | 9.3 | 7.7 | 37.4 | 12.3 | 34.4 | 9.0 | 20.0 | 31.2 | 7.2 |
| Doubao-Seed-1.6-thinking | 15.3 | 8.2 | 10.1 | 24.3 | 10.1 | 24.2 | 6.4 | 19.2 | 21.0 | 4.2 |
| Doubao-Seed-1.6 | 13.9 | 5.4 | 6.9 | 26.4 | 10.3 | 23.6 | 6.3 | 20.1 | 17.5 | 2.3 |
| Claude Sonnet 4 | 5.7 | 3.7 | 2.5 | 10.8 | 4.3 | 10.3 | 1.4 | 8.4 | 6.6 | 3.5 |
| *Open-Source LLMs* | | | | | | | | | | |
| Qwen3-235B-A22B-Thinking | 17.4 | 8.9 | 16.2 | 31.9 | 15.1 | 23.7 | 5.6 | 22.3 | 21.1 | 2.3 |
| DeepSeek-R1 | 11.4 | 5.1 | 11.9 | 25.4 | 8.7 | 22.5 | 4.7 | 16.3 | 9.8 | 3.5 |
| gpt-oss-120b | 7.3 | 6.3 | 5.7 | 18.3 | 4.9 | 14.5 | 1.6 | 12.5 | 5.5 | 0.0 |
| Mistral Small 3.1 | 6.9 | 3.0 | 2.7 | 5.5 | 7.0 | 2.0 | 8.5 | 4.0 | 12.2 | 3.0 |
| Llama 4 Maverick | 2.3 | 1.5 | 2.0 | 4.8 | 3.0 | 3.6 | 0.0 | 5.8 | 1.6 | 0.2 |
| Gemma 3 27B | 2.0 | 2.1 | 1.6 | 3.0 | 2.7 | 0.2 | 0.7 | 7.7 | 1.0 | 0.0 |
| Qwen3-235B-A22B-Instruct | 1.7 | 1.2 | 0.0 | 2.7 | 0.4 | 1.0 | 0.1 | 4.3 | 2.5 | 1.1 |
| DeepSeek-V3.1 | 1.7 | 1.2 | 2.0 | 1.7 | 1.0 | 5.8 | 0.5 | 2.2 | 2.1 | 0.0 |
| Qwen2.5-VL-72B | 0.2 | 0.0 | 0.7 | 0.0 | 0.0 | 0.0 | 0.0 | 0.0 | 0.6 | 0.0 |

## 4.2 MAIN RESULTS

Table 1 presents our evaluation results. Our main findings are summarized as follows:

**Overall performance remains unsatisfactory.** GPT-5 achieves the highest average score in the image input group (19.2), while Gemini 2.5 Pro, the best-performing model in the text input setting, still fails to surpass the 60-point threshold in any error category. Even in the SG category, which yields the best overall performance, nearly half of the models receive single-digit scores. Most

models perform poorly under the scan-oriented task paradigm and fail to detect any issues in many papers. This challenge is particularly pronounced for open-source models.

**Reasoning-enhanced models demonstrate clear advantages.** Across both input configurations, reasoning-enhanced variants consistently achieve higher scores. Almost all best-performing models, measured by metrics for both specific error categories and overall performance, fall into this category. In particular, Qwen3-235B-A22B-Thinking and Deepseek-R1 outperform their base versions by more than 10% in average scores, with substantial gains observed across all error categories. These results indicate that reasoning-enhanced models are better able to simulate the iterative process of extraction followed by reasoning, which is essential for effectively handling scan-oriented tasks and producing higher-quality responses.

**MLLMs face significant bottlenecks in handling long multimodal inputs.** Across most error categories, text inputs outperform image inputs. Among the nine MLLMs tested, the average performance gap between text and image inputs reaches 4.81 points, highlighting visual processing as a key limitation in current MLLM capabilities.

**Although overall performance is generally weaker, multimodal input remains indispensable.** In certain categories such as CF, where OCR-based text extraction leads to substantial loss of formulaic or tabular content, image inputs outperform their text counterparts. This highlights the essential role of multimodal reasoning and the irreplaceable value of visual information in addressing specific error categories.

## 4.3 FINE-GRAINED ANALYSIS

**Capability Dimensions.** We compute pairwise Spearman correlations between error categories across two input configurations (text and image) for the eight evaluated MLLMs excluding Qwen2.5-VL-72B, as shown in Figure 4. We derive the following insights:

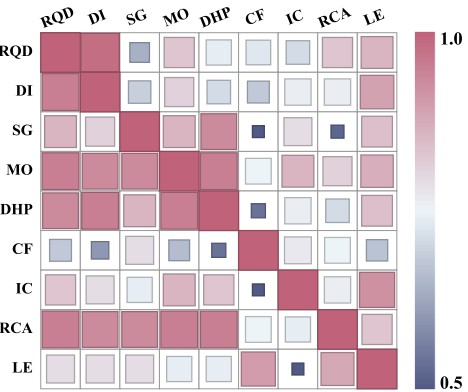

Figure 4: Spearman correlation matrix among the 9 error categories.

*(i) With image input, CF exhibits consistently low correlations with other error categories, suggesting that the skills required for mathematical reasoning are relatively distinct.* In contrast, with text input, CF shows moderate correlation with LE, indicating that OCR-flattened formulas lose their structural specificity and are interpreted by models in a manner more akin to natural language. Combined with the overall poor performance on CF tasks, this underscores the unique challenges of this category and the need for targeted improvements.

*(ii) Although DI is also related to experimental settings, it does not exhibit strong correlations with SG, MO, or DHP.* This indicates that DI primarily emphasizes causal framing and variable identifiability, rather than the procedural understanding of experimental operations.

*(iii) OCR severely degrades structured content such as figures and formulas, making questions that depend on multimodal information unanswerable.* This diminishes the expression of multimodal reasoning capabilities and artificially inflates inter-category correlations under text input.

Based on the above analysis, we consolidate the original nine error categories, each defined by its objective target, into five core latent skill dimensions evaluated by ScholScan under image input. While each dimension emphasizes the primary competence of its corresponding error categories, they are not mutually exclusive, as many questions involve overlapping reasoning abilities.

RQD and DI correspond to *research concept comprehension*, which requires models to identify the scope and definition of research objectives by integrating contextual cues and prior knowledge. SG, MO and DHP fall under *experimental process modeling*, which tests a model's ability to reconstruct procedural workflows such as sampling, measurement, and data handling. CF captures

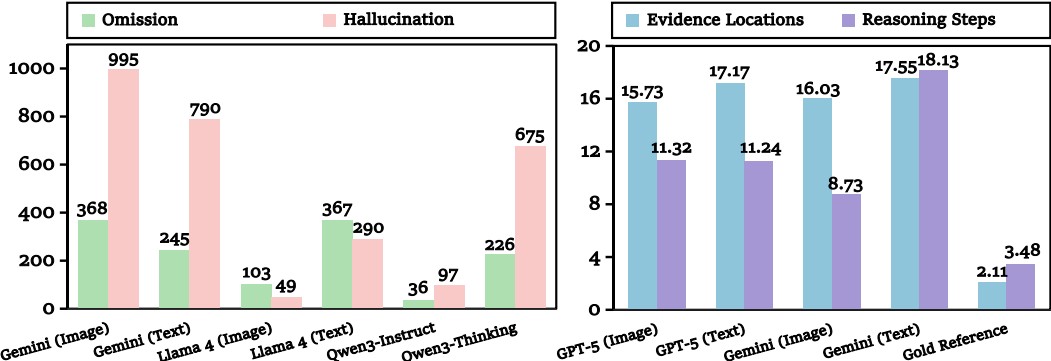

Figure 5: **Left**: Distribution of omission and hallucination errors. **Right**: Average reasoning steps and evidence locations involved in the answer generation, compared against the gold reference.

*formal reasoning and symbolic computation*, focusing on syntactic parsing and numerical logic. IC evaluates *causal inference*, where models must synthesize dispersed causal evidence to reach sound conclusions. RCA and LE reflect *referential alignment and linguistic consistency*, which assess the ability to verify citations and maintain coherent expression throughout the document.

**Hidden Complexity in Scan-Oriented Tasks.** We analyze the reasoning traces of GPT-5 and Gemini 2.5 Pro across both input configurations, focusing on the number of evidence pieces scanned and the reasoning steps performed. As illustrated in Figure 5, even the most advanced models often scan up to 8 times more evidence and execute 3.5 times more reasoning steps than the reference answers, merely to approximate a correct response, yet they still frequently fail. This highlights the substantial hidden complexity inherent in scan-oriented tasks, which significantly amplifies the challenge of successful task completion.

## 4.4 ERROR ANALYSIS

**Omission and Hallucination.** Most zero-score cases fall into two categories: either the model fails to detect any errors in the paper, or it becomes overwhelmed by hallucinations and entirely overlooks the actual errors present in the reference answer. We analyze the number of zero-score questions and the proportion of these two failure modes across models, as shown in Figure 5. Stronger models tend to have fewer zero-score cases overall, but are more prone to overconfident hallucinations.

**Fragile Reasoning under Complex Evidence.** Figure 6 shows how best-performing models behave under different numbers of reasoning steps and evidence locations. As reasoning steps increase, both reasoning and overall scores steadily decline, revealing a clear bottleneck in MLLMs' ability to construct long causal chains. In contrast, variation in evidence locations has a weaker and less consistent impact. However, this does not imply that multi-evidence questions pose only marginal difficulty. Since the evaluation metric allows partial evidence omissions, more evidence items do not necessarily incur large score penalties. Still, heavier evidence loads often require longer reasoning chains, which substantially affect the coherence and completeness of inferred logic. These results highlight the persistent challenge for MLLMs in integrating evidence and maintaining logical structure as task complexity increases.

## 4.5 RAG ANALYSIS

We evaluated 8 RAG methods across both input configurations (Robertson et al., 1994; Chen et al., 2024; Lee et al., 2025; Faysse et al., 2025; Yu et al., 2025; Wang et al., 2025; Izacard et al., 2022). Key findings are presented below, with detailed results shown in Tables 2 and 3.

**The Oracle condition yields significant accuracy gains.** Providing gold images alleviates the scanning burden in long-context inputs, increasing the chances of generating correct answers. Although overall performance improves, the gains are limited for CF errors and minimal for LE errors. For CF, the sparse formulaic content means gold images offer limited assistance. For LE, the dense text distribution makes even direct access to target regions insufficient for current models to reduce complexity.

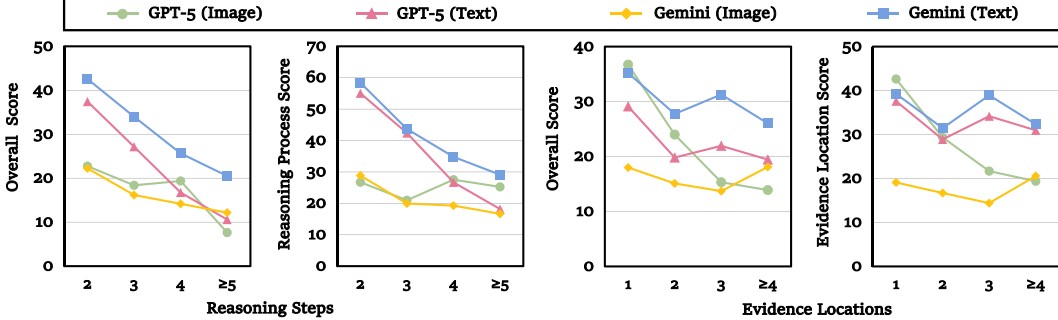

Figure 6: Model performance trends across reasoning steps and evidence locations (scaled by 100).

Table 2: Overall scores of RAG methods across the 9 error categories (scaled by 100).

| Model | Avg. | RQD | DI | SG | MO | DHP | CF | IC | RCA | LE |
|---|---|---|---|---|---|---|---|---|---|---|
| *Text Input (Base Model: Qwen3-Thinking)* | | | | | | | | | | |
| Baseline | 17.4 | 8.9 | 16.2 | 31.9 | 15.1 | 23.7 | 5.6 | 22.3 | 21.1 | 2.3 |
| Oracle | 24.5 | 20.6 | 27.9 | 43.6 | 21.3 | 40.8 | 7.4 | 26.9 | 26.0 | 1.9 |
| BM25 | 16.7 | 9.7 | 13.7 | 33.0 | 17.3 | 23.8 | 6.8 | 25.4 | 16.5 | 3.0 |
| Contriever | 16.6 | 9.7 | 18.2 | 33.7 | 10.7 | 20.8 | 6.4 | 18.5 | 19.8 | 1.8 |
| BGE-M3 | 11.3 | 8.6 | 7.5 | 24.8 | 9.1 | 15.4 | 5.3 | 15.6 | 11.4 | 1.0 |
| NV-Embed-v2 | 6.8 | 4.0 | 4.0 | 9.4 | 6.1 | 4.9 | 5.5 | 5.7 | 10.0 | 2.0 |
| *Image Input (Base Model: Llama 4 Maverick)* | | | | | | | | | | |
| Baseline | 7.0 | 7.0 | 7.3 | 9.4 | 4.5 | 4.0 | 6.5 | 6.7 | 8.8 | 3.0 |
| Oracle | 6.5 | 3.0 | 4.5 | 15.6 | 8.2 | 9.4 | 4.9 | 10.0 | 4.4 | 1.4 |
| VRAG-RL | 10.9 | 9.8 | 11.6 | 17.8 | 8.2 | 11.0 | 6.8 | 13.1 | 10.8 | 8.1 |
| ColQwen2.5 | 1.2 | 2.1 | 0.7 | 0.5 | 0.0 | 1.2 | 0.2 | 2.7 | 2.0 | 0.0 |
| VisRAG | 1.0 | 2.0 | 0.0 | 1.0 | 0.0 | 1.0 | 1.6 | 1.3 | 1.2 | 0.0 |
| ColPali-v1.3 | 0.8 | 1.5 | 0.0 | 0.5 | 0.0 | 0.9 | 0.5 | 1.3 | 1.4 | 0.0 |

**In consistency-centric scan-oriented tasks, most retrieval-based enhancement methods show minimal effectiveness.** All embedding models exhibit poor retrieval accuracy. None achieves recall of 50% within the top-5 retrieved items. More critically, performance deteriorates after retrieval, especially for multimodal embedding models, where post-retrieval responses are almost entirely incorrect and scores approach 0.

**Complex embedding model architectures do not yield better performance.** Under the text-input setting, BM25 achieves the highest retrieval metrics, outperforming Contriever and NV-Embed-v2. Under the image-input setting, although VisRAG shows certain advantages in retrieval performance, its overall score

Table 3: Retrieval performance of RAG methods.

| Model | MRR@5 | Recall@5 |
|---|---|---|
| *Text Input (Base Model: Qwen3-Thinking)* | | |
| BM25 | 0.41 | 0.48 |
| Contriever | 0.31 | 0.39 |
| NV-Embed-v2 | 0.30 | 0.38 |
| BGE-M3 | 0.16 | 0.21 |
| *Image Input (Base Model: Llama 4 Maverick)* | | |
| VisRAG | 0.41 | 0.46 |
| ColQwen2.5 | 0.30 | 0.35 |
| ColPali-v1.3 | 0.26 | 0.31 |

remains comparably low and converges with methods such as ColPali-v1.3. Under such circumstances, comparisons between retrieval metrics lose their substantive significance. The underlying reason lies in the fact that existing embedding models are primarily designed to enhance retrieval performance at the level of semantic relevance. They already struggle with traditional multi-hop reasoning tasks, let alone scan-oriented tasks with target suppression.

**Reinforcement learning frameworks with visual focus have emerged as leading approaches.** Despite being built on a compact 7B model, VRAG-RL consistently delivers improved performance and is the only method that achieves gains under image input following RL optimization. Its enhanced retrieval sharpens evidence selection, while strong reasoning provides effective guidance during document scanning. The retrieval and reasoning components are interleaved in the design, with each stage informing the other in an iterative loop. This tightly coupled interaction contributes to the method's superior performance potential.

# 5    CONCLUSION

In this paper, we introduce ScholScan, a benchmark designed to evaluate the performance of MLLMs on scan-oriented tasks that require the detection of scientific errors across entire academic papers. We conduct a comprehensive evaluation and in-depth analysis of mainstream MLLMs and RAG methods. The results demonstrate that current MLLMs remain far from capable of reliably addressing such tasks and that existing RAG methods provide little to no improvement. This highlights the complexity, integrative demands, and originality of the ScholScan benchmark. Looking ahead, our goal is to develop scan-oriented task paradigms suited to diverse academic scenarios and explore new techniques to improve model performance on target-suppressed inputs. These directions support the larger goal of advancing MLLMs from passive assistants to active participants in scientific research.

## ETHICS STATEMENT

**Data Provenance.** All data used in this paper were constructed by the authors and do not include external public or proprietary datasets. The academic papers and author names referenced are publicly available through arXiv and OpenReview.

**Annotation Process.** A team of 10 domain experts was assembled to thoroughly review all tasks initially generated by Gemini 2.5 Pro. All annotators provided informed consent to participate. To ensure accuracy and neutrality of both model-generated and human-verified content, we employed a rigorous multi-stage validation process involving cross-review and third-party adjudication.

**Model Evaluation.** Evaluation of 15 mainstream models in 24 input configurations was carried out using legally authorized API access through VolcEngine, Alibaba Cloud's LLM services, and OpenRouter.

**Dissemination.** ScholScan is open source and freely available for academic and non-commercial research. All personally identifiable information has been removed from the dataset and its collection and release comply with the ethical and legal requirements in place at the time of data acquisition.

## REPRODUCIBILITY STATEMENT

All results presented in this paper are fully reproducible. To facilitate verification and extension, we provide the complete dataset on Hugging Face, source code and detailed documentation on GitHub. The GitHub repository includes step-by-step instructions and the exact hyperparameter configurations used in our experiments, ensuring full reproducibility. The retrieval components in all RAG experiments were executed on a server equipped with 8 NVIDIA A40 GPUs.

## ACKNOWLEDGMENTS

This work is supported by the Beijing Natural Science Foundation (Grant No. QY25345), the National Natural Science Foundation of China (Grant Nos. 62473271, 62176026), and the Fundamental Research Funds for the Beijing University of Posts and Telecommunications (Grant No. 2025AI4S03). This work is also supported by the Engineering Research Center of Information Networks, Ministry of Education, China. We would also like to thank the anonymous reviewers and area chairs for constructive discussions and feedback.

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

# Appendix Contents

# A  PROMPT TEMPLATES

## A.1  WITHIN-PAPER GENERATION PROMPT

---

### Within-Paper Generation Prompt

```
You will receive a high-quality, already accepted scientific
   paper as a PDF. Working only with the PDF itself (and any
    appendix embedded in the same PDF), edit specific
   textual spans to inject one or more errors chosen only
   from the taxonomy below, such that the errors are hard
   yet clearly identifiable by a professional reviewer
   reading the PDF alone.

Error Types (fixed):
Research Question & Definitions
   Definition: The core construct/hypothesis/variable is
      insufficiently or inconsistently defined (conceptual
      vs. operational), leaving the estimand ambiguous.
Design & Identifiability
   Definition: Given a clear estimand, the design violates
      structural identification conditions so the effect is
      not identifiable even with infinite data and perfect
      measurement.
Sampling & Generalizability
   Definition: The sampling frame/process/composition or
      cluster/power setup does not support valid or stable
      sample→population claims.
Measurement & Operationalization
   Definition: Measures/manipulations lack feasibility/
      reliability/validity/timing, so observed variables
      systematically diverge from the intended construct/
      treatment.
Data Handling & Preprocessing
   Definition: Pipeline choices in missing handling, joins/
      keys, temporal splitting, feature construction, or
      partitioning introduce bias (including leakage or unit
      /scale conflicts).
Computation & Formulae
   Definition: Arithmetic/algebra/notation errors (totals/
      ratios, unit conversion, CI vs. point estimate, p-
      value vs. label, symbol reuse, undefined variables,
      dimension mismatch).
Inference & Conclusions
   Definition: Interpretations or causal statements exceed
      what methods/data support, or contradict the shown
      statistics/tables/captions.
Referential & Citation Alignment
   Definition: Contradictions about the same quantity/term
      across text, tables, captions, or appendix within the
      paper.
Language & Expression
   Definition: Terminology/capitalization/grammar ambiguities
       that affect meaning or domain-critical term
      consistency (not cosmetic typos).
```

---

**Within-Paper Generation Prompt (Continued)**

```
Global constraints (must comply):
1. Each error must map to exactly one primary category in the
     taxonomy. Do not mix causes.
2. Each error must involve more than 2 micro-edits (each edit
     ≤ 20 English words) spread across distinct pages or
    paragraphs.
3. If an edit would create an immediate contradiction in the
    same sentence/paragraph/caption, you may add shadow patch
    (es) for the same error to keep the text natural (still
    counted as edit locations).
4. Independence across errors (per-copy generation):
   Generate each error on a separate copy of the original PDF
       . Different errors must be logically and operationally
        independent:
   No progression or variant relations: an error must not be
       a stricter/looser version, superset/subset, or minor
       wording variant of another error.
   No anchor reuse: do not target the same sentence/caption/
       table cell or reuse the same old_str (or a near-
       duplicate paraphrase) across different errors.
   Applying any single error in isolation to the original PDF
        must still yield a detectable, clearly categorizable
       error according to the taxonomy.
5. Every error must be supportable using text inside the PDF.
     Do not rely on external supplementary files or prior
    knowledge.
6. Design as difficult as possible but clean errors. Prefer
    edits that force cross-checking between two spots (e.g.,
    Methods vs. Results). Avoid trivialities. Edits must
    remain locally plausible and not advertise themselves via
     obviously artificial phrases (e.g., avoid contrived
    tokens purely added to be detectable).
7. "No cosmetic issues" applies except for I (Language &
    Expression). For I, edits must affect meaning or domain-
    critical terminology (e.g., ambiguous phrasing,
    inconsistent technical terms). Pure typos, punctuation
    tweaks, or layout nits are not allowed.
8. Do not edit titles, author lists, bibliography entries,
    equation numbering, figure images, or add new figures/
    tables/references.
9. Frame each question as a neutral imperative that asks for
    a decision about a specific condition, using (but not
    limited to) Decide/Determine/Judge/Evaluate/Assess
    whether... Do not presuppose an outcome or use suggestive
     intensifiers (e.g., clearly/obviously/likely/suspicious)
    .
```

---

**Within-Paper Generation Prompt (Continued)**

```
10. Output in English only and strictly follow the JSON
    schema below. Do not include any additional text outside
    the JSON:
[
  {
    "id": "1-based integer as string",
    "modify": [
      {
        "location": "Page number + short unique nearby quote (
            ≤ 15 tokens).",
        "old_str": "Exact original text from the PDF (verbatim)
            .",
        "new_str": "Edited text after your change."
      }
      /* Add 1-2 more locations; each location ≤ 20 words
          changed.
        Shadow patches for local coherence count as locations.
            */
    ],
    "question": "One neutral audit-style task (1-25 words).",
    "explanation": "Explain in 2-4 sentences why a reviewer
        can detect this error from the edited PDF alone.",
    "Type": "Name the primary category (e.g., Inference &
        Conclusions)."
  }
  /* More Errors */
]
```

---

## A.2 WITHIN-PAPER SAMPLING PROMPT

---

**Within-Paper Sampling Prompt**

You will receive a paper PDF and the weaknesses mentioned in
    its peer-review comments. Your task is, based only on the
     content of that PDF, to sample from the review comments
    and verify possible errors related to the categories
    below, and for each confirmed or highly plausible error,
    generate one question and one explanation.

Error Types (fixed):
Research Question & Definitions
    Definition: The core construct/hypothesis/variable is
        insufficiently or inconsistently defined (conceptual
        vs. operational), leaving the estimand ambiguous.
Design & Identifiability
    Definition: Given a clear estimand, the design violates
        structural identification conditions so the effect is
        not identifiable even with infinite data and perfect
        measurement.
Sampling & Generalizability
    Definition: The sampling frame/process/composition or
        cluster/power setup does not support valid or stable
        sample→population claims.
Measurement & Operationalization
    Definition: Measures/manipulations lack feasibility/
        reliability/validity/timing, so observed variables
        systematically diverge from the intended construct/
        treatment.
Data Handling & Preprocessing
    Definition: Pipeline choices in missing handling, joins/
        keys, temporal splitting, feature construction, or
        partitioning introduce bias (including leakage or unit
        /scale conflicts).
Computation & Formulae
    Definition: Arithmetic/algebra/notation errors (totals/
        ratios, unit conversion, CI vs. point estimate, p-
        value vs. label, symbol reuse, undefined variables,
        dimension mismatch).
Inference & Conclusions
    Definition: Interpretations or causal statements exceed
        what methods/data support, or contradict the shown
        statistics/tables/captions.
Referential & Citation Alignment
    Definition: Contradictions about the same quantity/term
        across text, tables, captions, or appendix within the
        paper.
Language & Expression
    Definition: Terminology/capitalization/grammar ambiguities
         that affect meaning or domain-critical term
        consistency (not cosmetic typos).

---

---

**Within-Paper Sampling Prompt (Continued)**

```
Global constraints (must comply):
1. Output only the specified categories; even if other error
   types appear in the reviews, do not output them.
2. Sample first, then verify: extract candidates from the
   review comments, then confirm them in the PDF. If you
   cannot locate supporting anchors in the PDF (page number
   plus phrase/label), do not output that candidate.
3. Questions must be neutral and non-leading: use an "audit
   task + decision" style, avoiding yes/no bias.
4. Independence: each question must target a different figure
    or different textual anchor; no minor variants of the
   same issue.
5. Evidence first: the explanation must cite locatable
   anchors in the PDF (page number + original phrase/caption
   ). You may mention a key short phrase from the review as
   a clue, but write the question and explanation in your
   own words.
6. Language & format: both question and explanation must be
   in English; output JSON only, with no extra text.
7. Quantity: sort by evidence strength and output up to 5
   items; if none qualify, output an empty array [].
Output JSON schema:
[
  {
    "id": "1",
    "question": "Audit y-axis baselines and possible axis
        breaks in Figure 2; decide presence/absence and cite
        evidence.",
    "explanation": "The review flags possible exaggeration in
        Fig.2. In the PDF (p.6, caption 'Performance vs.
        baseline'), the y-axis starts at 0.85 with a break,
        magnifying small differences; panels use different
        ranges.",
    "Type": "Data Handling & Preprocessing"
  }
]
```

## A.3   CROSS-PAPER GENERATION PROMPT

---

### Cross-Paper Generation Prompt

```
You will receive two PDFs: a "focus paper" P and a cited "
    evidence paper" S (exactly one pair per run). Edit only P
     with textual changes so that P's statements about S
    become incompatible with or unsupported by S along one or
     more dimensions (direction, scope, condition, metric,
    unit, version, protocol). Do not modify S. Each error
    must be detectable by a professional reviewer who reads
    both PDFs.

Error Types (fixed):
Research Question & Definitions (A)
    Definition: The core construct/hypothesis/variable is
        insufficiently or inconsistently defined (conceptual
        vs. operational), leaving the estimand ambiguous.
Design & Identifiability (B)
    Definition: Given a clear estimand, the design violates
        structural identification conditions so the effect is
        not identifiable even with infinite data and perfect
        measurement.
Sampling & Generalizability (C)
    Definition: The sampling frame/process/composition or
        cluster/power setup does not support valid or stable
        sample-population claims.
Measurement & Operationalization (D)
    Definition: Measures/manipulations lack feasibility/
        reliability/validity/timing, so observed variables
        systematically diverge from the intended construct/
        treatment.
Data Handling & Preprocessing (E)
    Definition: Pipeline choices in missing handling, joins/
        keys, temporal splitting, feature construction, or
        partitioning introduce bias (including leakage or unit
        /scale conflicts).
Computation & Formulae (F)
    Definition: Arithmetic/algebra/notation errors (totals/
        ratios, unit conversion, CI vs. point estimate, p-
        value vs. label, symbol reuse, undefined variables,
        dimension mismatch).
Inference & Conclusions (G)
    Definition: Interpretations or causal statements exceed
        what methods/data support, or contradict the shown
        statistics/tables/captions.
Referential & Citation Alignment (H)
    Definition: Contradictions about the same quantity/term
        across text, tables, captions, or appendix within the
        paper.
Language & Expression (I)
    Definition: Terminology/capitalization/grammar ambiguities
         that affect meaning or domain-critical term
        consistency (not cosmetic typos).
```

---

### Cross-Paper Generation Prompt (Continued)

```
Global constraints (must comply):
1. Edit P only, never S. Do not add/remove references,
    figures, or external links.
2. Micro-edits, not rewrites.
3. Cross-paper verifiability. For every error, your
    explanation must quote >=1 anchor from P and >=1 anchor
    from S (page number + a short verbatim span, 10-20 words)
     that, together, demonstrate the conflict or lack of
    support.
4. Local coherence. The edited text in P must read naturally
    in context; the contradiction should emerge only when P
    is compared with S.
5. Independence across errors. You may output multiple errors
    , but generate each on an independent copy of P. Errors
    must not reuse the same P anchor or be minor variants (no
     strengthen/weaken/same-sentence paraphrases).
6. PDF-only evidence. Rely solely on content present in the
    two PDFs. No outside knowledge or materials.
7. Scope exclusions. Do not edit titles, author lists,
    bibliography entries, equation numbering, LaTeX commands,
     or figure pixels; do not add new figures/tables/
    references.
8. Type must be a single letter in {A,B,C,D,E,F,G,H,I}.
9. Inter-paper note. "Inconsistency" describes the evidence
    relation (P vs. S). The Type must reflect the primary
    upstream cause (e.g., B/C/D/F/G/I/A).
10. Quantity & ordering. Sort by evidence strength and output
     up to 5 errors; if none qualify, output an empty array
    [].
11. Language & format. Output English-only JSON exactly in
    the schema below; include no extra text.
12. Multiple mentions of S. The evidence paper S may be cited
     multiple times in P. You may (i) introduce a single
    error spanning >=2 of those citation points, or (ii)
    generate multiple independent errors, each based on a
    different citation of S.
```

---

**Cross-Paper Generation Prompt (Continued)**

```
Output JSON schema:
[
  {
    "id": "1-based string",
    "modify": [
      {
        "location": "P: page X + short nearby quote (≤ 15
            tokens).",
        "old_str": "Exact original text from P (verbatim).",
        "new_str": "Edited text in P after your change."
      }
      /* could add 1-2 more P locations */
    ],
    "question": "One neutral audit-style task (1-25 words) for
        checking P against S.",
    "explanation": "Diagnostic voice: state that the edited P
        now contains an error of Type <letter>, and
        substantiate it with anchors from both PDFs. Include
        page numbers and short verbatim quotes from P and S
        showing the mismatch (e.g., scope/condition/metric/
        unit/version/protocol). Do not describe how to create
        the error; assert and evidence the error as present.",
    "Type": "A|B|C|D|E|F|G|H|I"
  }
]
```

---

## A.4 EXTRACTOR PROMPT

---

### Extractor Prompt

```
You will receive three inputs:
Q: the open-ended question;
E: the gold explanation (describes exactly one error; extra
    details still belong to the same single error);
A: the model's answer to be evaluated.
Your job is to extract counts only and output a single JSON
    object with the exact schema below. Do not compute any
    scores. Do not add fields.

Core selection rule (multiple errors in A)
1. Parse E into a single gold error (the "target error").
2. From A, identify how many distinct error claims are made.
    Cluster together mentions that support the same error (
    multiple locations for one error are still one error).
3. Existence decision (binary correctness only):
Let the gold existence be 1 if E asserts an error exists,
    else 0.
Let the predicted existence be 1 if A asserts any error, else
     0 (e.g., states no error).
Set existence = 1 if predicted existence equals gold
    existence; otherwise set existence = 0.
4. If existence = 0: set contains_target_error = 0; set all
    location and reasoning counts to 0; and set
    unrelated_errors to the total number of distinct error
    claims in A. Then output the JSON.
5. If existence = 1:
If the gold existence is 1: determine whether A contains the
    target error (match by the main error idea in E: category
    /intent/scope; treat E's subpoints as the same error).
    If yes, set contains_target_error = 1 and compute location
        and reasoning only for the target error. Count all
        other error claims in A as unrelated_errors.
    If no, set contains_target_error = 0; set all location and
        reasoning counts to 0; set unrelated_errors to the
        total number of distinct error claims in A.
If the gold existence is 0: set contains_target_error = 0;
    set all location and reasoning counts to 0; set
    unrelated_errors to the total number of distinct error
    claims in A. (These negative items are for binary
    accuracy only; they are not used for detailed scoring.)

Matching guidance (A error ↔ target error): match by the
    main error idea in E (category/intent/scope), not by
    wording. Treat E's subpoints as part of the same single
    error. Prefer the best-matching cluster in A; if ties,
    choose the one with stronger alignment to E's core claim.
```

---

**Extractor Prompt (Continued)**

```
Counting rules:
Location (for the target error only when existence=1 and
    contains_target_error=1):
gold_steps: number of unique error locations described in E (
    after normalization and deduplication).
hit_steps: number of predicted locations in A that match any
    gold location for the target error.
extra_steps: number of predicted locations in A for the
    target error that do not match any gold location.

Reasoning (for the target error only when existence=1 and
    contains_target_error=1):
Convert E into a canonical set or ordered chain of reasoning
    steps for the target error.
gold_steps: total number of such steps.
reached_steps:
    single-chain tasks: length of the longest valid prefix of
        A along the gold chain;
    multi-path/parallel tasks: size of the intersection
        between A's steps and the gold step set (or the
        maximum across gold paths if multiple are defined).
missing_steps: gold_steps - reached_steps (non-negative
    integer).
Unrelated errors:
unrelated_errors: number of distinct error claims in A that
    are not the target error (0 if none).
Output JSON schema (return exactly this JSON; integers only):
{
  "existence": 0,
  "contains_target_error": 0,
  "location": {
    "gold_steps": 0,
    "hit_steps": 0,
    "extra_steps": 0
  },
  "reasoning": {
    "gold_steps": 0,
    "reached_steps": 0,
    "missing_steps": 0
  },
  "unrelated_errors": 0
}
```

---

## A.5 EVALUATION SYSTEM PROMPT

---

### Evaluation System Prompt

You are a neutral, careful academic reviewer. You will
    receive an open-ended question and the paper content. The
     paper may or may not have issues related to the question
    . Do not assume there are errors. If the question is
    about citations, you will be given a citing paper and a
    cited paper; evaluate only the citing paper for possible
    issues and use the cited paper only as the reference for
    comparison. Write in natural prose with no fixed template
    .

Rules:
Speak only when sure. State an error only if you are
    confident it is a real error (not a mere weakness).
Stay on scope. Discuss only what the question asks about.
Evidence completeness. For every error you state, list all
    distinct evidence cues you are confident about from the
    PDF. Include plain identifiers (figure/table/section/
    equation/citation) or quotes. Avoid redundant repeats of
    the exact same instance; include all distinct locations
    needed to support the error.
Be clear and brief. Use short, direct sentences.
No metaphors. No fancy wording. No guesses or outside sources
    . Do not invent figures, tables, equations, citations, or
     results.
Report as many distinct, well-supported errors as you can
    within scope. If none are clear, write exactly: "No clear
     issue relevant to the question." and nothing else.

---

# B    EXAMPLES FROM EXISTING DATASETS

## B.1    DOCMATH-EVAL



**DocMath-Eval**

**Question ID**: complong-testmini-30
**Question**: What is the percentage of total offering cost on the total amount raised in the IPO if the total offering cost is $14,528,328 and each unit sold is $10?



**Context Modalities**: **Text + Text Document**
1. Offering costs consist of legal, accounting and other costs incurred through the balance sheet date that are directly related to the Initial Public Offering. Offering costs amounting to $14,528,328 were charged to shareholders' equity upon the completion of the Initial Public Offering.
2. Pursuant to the Initial Public Offering on July 20, 2020, the Company sold 25,300,000 Units, which includes the full exercise by the underwriter of its option to purchase an additional 3,300,000 Units, at a purchase price of $10.00 per Unit. Each Unit consists of one Class A ordinary share and one-half of one redeemable warrant ("Public Warrant"). Each whole Public Warrant entitles the holder to purchase one Class A ordinary share at an exercise price of $11.50 per whole share (see Note 7).

**Covered Areas**:

**Only Mathematics**

**Cross-Evidence Reasoning**:

**Limited**

**Task Paradigm**:

**Search-Oriented**

## B.2 MMLONGBENCH-DOC



### MMLongBench-Doc

**Document ID**: afe620b9beac86c1027b96d31d396407.pdf

**Question**: How much higher was the proposed dividend paid (Rupees in lacs) in 2002 compared to 2001?



**Context Modalities**: **Text + Multimodal Document**

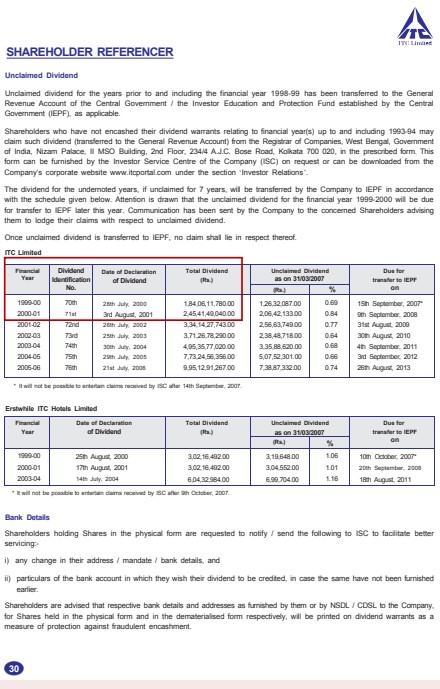

**Covered Areas**:

**Limited (7 areas)**

**Cross-Evidence Reasoning**:

**Limited**

**Task Paradigm**:

**Search-Oriented**

## B.3 FINMMDOCR

---

### FinMMDocR

**Question ID**: test-41

**Question**: Based on the detailed financial forecast tables provided at the end of the report, analyze the company's working capital management efficiency related to its customer collections for the fiscal year 2025. Using the year-end balances of (Accounts Receivable) for 2024 and 2025 to calculate the average balance for 2025, and the corresponding (Total Operating Revenue) for 2025, calculate the implied average collection period for receivables during 2025 (assume 365 days in a year, round to one decimal place, unit: days).

---

**Context Modalities**: **Text + Multimodal Document**

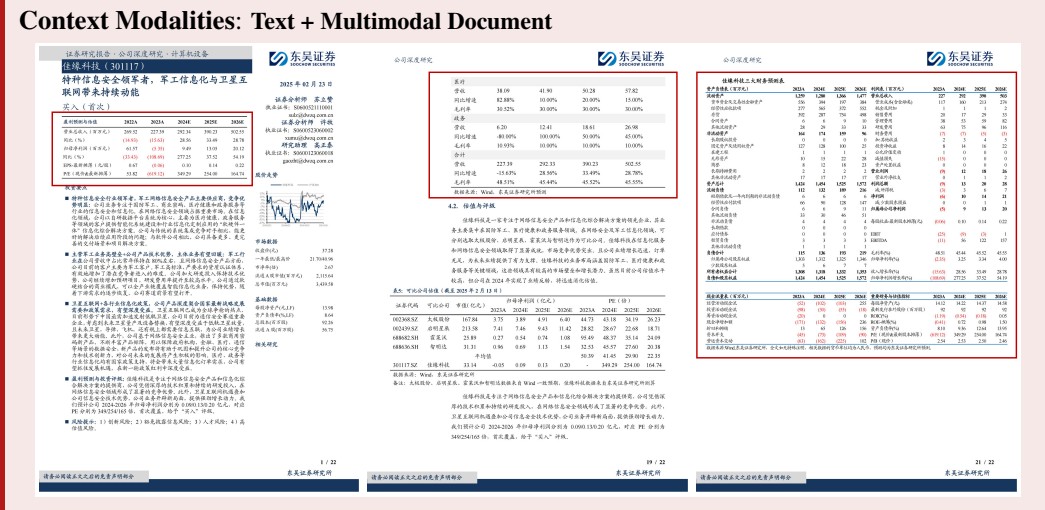

---

**Covered Areas**:

**Limited (Finance Only)**

---

**Cross-Evidence Reasoning**:

**Full**

---

**Task Paradigm**:

**Search-Oriented**

## B.4 LONGDOCURL

---

### LongDocURL

**Question ID**: free_gemini15_pro_4061601_47_71_8
**Question**: What was the total fair value of options that vested in 2016, 2015, and 2014, in millions of Canadian dollars?

---

**Context Modalities**: **Text + Multimodal Document**

The following table summarizes additional stock option information:

| year ended December 31 (millions of Canadian $, unless otherwise noted) | 2016 | 2015 | 2014 |
|---|---|---|---|
| Total intrinsic value of options exercised | 31 | 10 | 21 |
| Fair value of options that have vested | 126 | 91 | 95 |
| Total options vested | 2.1 million | 2.0 million | 1.7 million |

As at December 31, 2016, the aggregate intrinsic value of the total options exercisable was $86 million and the total intrinsic value of options outstanding was $130 million.

**21. PREFERRED SHARES**

In March 2014, TCPL redeemed all of the 4 million outstanding Series Y preferred shares at a redemption price of $50 per share for a gross payment of $200 million.

**22. OTHER COMPREHENSIVE (LOSS)/INCOME AND ACCUMULATED OTHER COMPREHENSIVE LOSS**

Components of Other comprehensive (loss)/income, including the portion attributable to non-controlling interests and related tax effects, are as follows:

| year ended December 31, 2016 (millions of Canadian $) | Before Tax Amount | Income Tax Recovery/ (Expense) | Net of Tax Amount |
|---|---|---|---|
| Foreign currency translation gains on net investment in foreign operations | 3 | — | 3 |
| Change in fair value of net investment hedges | (14) | 4 | (10) |
| Change in fair value of cash flow hedges | 44 | (14) | 30 |
| Reclassification to net income of gains and losses on cash flow hedges | 71 | (29) | 42 |
| Unrealized actuarial gains and losses on pension and other post-retirement benefit plans | (38) | 12 | (26) |
| Reclassification to net income of actuarial loss on pension and other post-retirement benefit plans | 22 | (6) | 16 |
| Other comprehensive loss on equity investments | (117) | 30 | (87) |
| **Other Comprehensive Loss** | **(29)** | **(3)** | **(32)** |

| year ended December 31, 2015 (millions of Canadian $) | Before Tax Amount | Income Tax Recovery/ (Expense) | Net of Tax Amount |
|---|---|---|---|
| Foreign currency translation gains on net investment in foreign operations | 798 | 15 | 813 |
| Change in fair value of net investment hedges | (505) | 133 | (372) |
| Change in fair value of cash flow hedges | (92) | 35 | (57) |
| Reclassification to net income of gains and losses on cash flow hedges | 144 | (56) | 88 |
| Unrealized actuarial gains and losses on pension and other post-retirement benefit plans | 74 | (23) | 51 |
| Reclassification to net income of actuarial loss and prior service costs on pension and other post-retirement benefit plans | 41 | (9) | 32 |
| Other comprehensive income on equity investments | 62 | (15) | 47 |
| **Other Comprehensive Income** | **522** | **80** | **602** |

155  TCPL **Consolidated financial statements** 2016

---

**Covered Areas**:

**Limited**

---

**Cross-Evidence Reasoning**:

**Limited**

---

**Task Paradigm**:

**Search-Oriented**

B.5   SLIDEVQA

**SlideVQA**

**Question ID**: 1
**Question**: How much difference in INR is there between the average order value of CY2013 and that of CY2012?

**Context Modalities**: **Text + Multimodal Document**

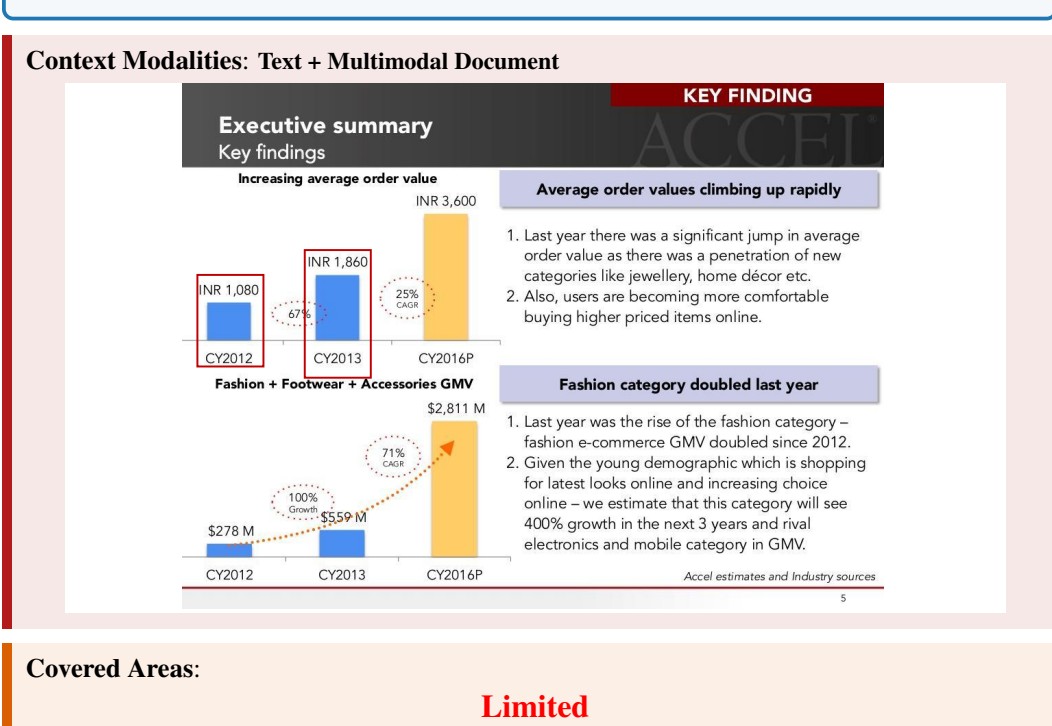

**Covered Areas**:

**Limited**

**Cross-Evidence Reasoning**:

**None**

**Task Paradigm**:

**Search-Oriented**

## B.6 DocVQA

---

**DocVQA**

**Question ID**: 24581
**Question**: What is name of university?

---

**Context Modalities**: **Text + Multimodal Document**

UNIVERSITY OF CALIFORNIA, SAN DIEGO

To _____ Paul
Date 11/30/82 Time 2:04 A.M./P.M.

**WHILE YOU WERE OUT**

Dr./Mr./Ms. _____ Wilson 455-8056
From _____ Scripps Clinic

☑ Telephoned   ☐ Will phone again   ☐ Please phone
☐ Came to see you   ☐ Will come again   ☐ Rush

**MESSAGE**

Re Program Committee — today Fdn. It will probably be 1st or 2nd week in March (1983) rather than latter half. Phone party at (None to call for) Taken by _____

---

**Covered Areas**:

**Limited**

---

**Cross-Evidence Reasoning**:

**None**

---

**Task Paradigm**:

**Search-Oriented**

## B.7   CHARXIV

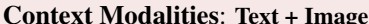

**CharXiv**

**Question ID**: 2004.10956
**Question**: Which model shows a greater decline in accuracy from Session 1 to Session 9 in the 5-way full-shot scenario?

**Context Modalities**: Text + Image

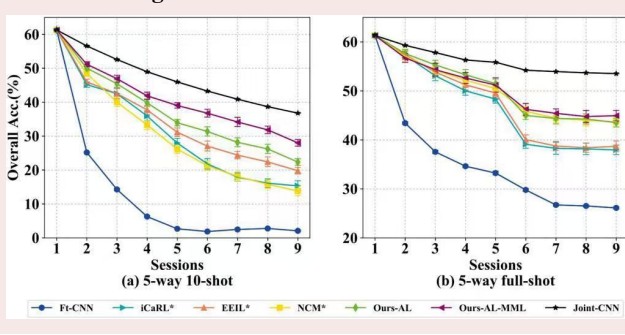

**Covered Areas**:

**Limited (8 areas)**

**Cross-Evidence Reasoning**:

**None**

**Task Paradigm**:

**Search-Oriented**

## B.8 ARXIVQA

> ### ArXivQA
>
> **Question ID**: physics-8049
> **Question**: Based on the top-right graph, how would you describe the behavior of P(z) as z approaches zero?

**Context Modalities**: **Text + Image**

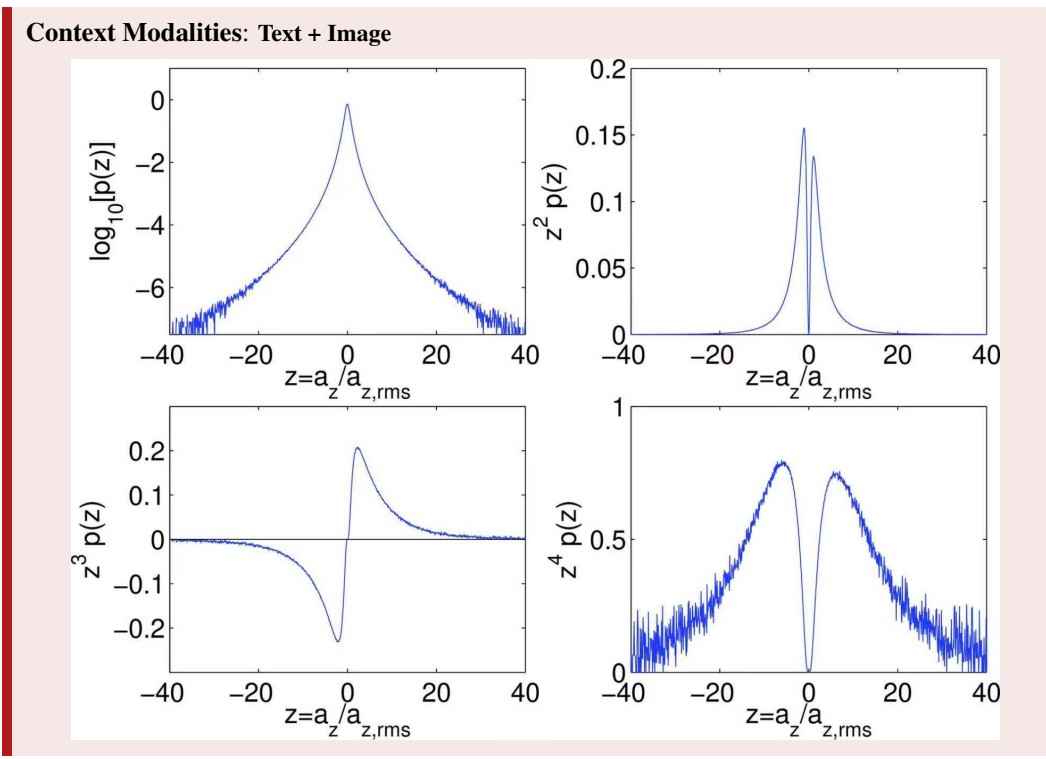

**Covered Areas**:

**Limited**

**Cross-Evidence Reasoning**:

**None**

**Task Paradigm**:

**Search-Oriented**

## B.9 SPIQA

**Question ID**: 1611.04684v1
**Question**: What is the role of the knowledge gates in the KEHNN architecture?

**Context Modalities**: **Text + Image**

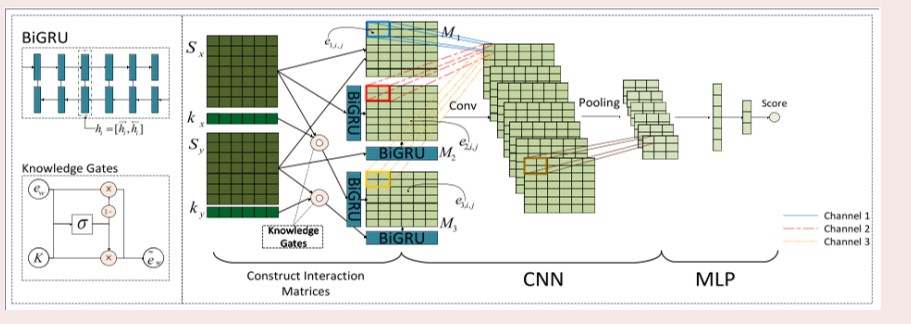

**Covered Areas**:

**Limited**

**Cross-Evidence Reasoning**:

**None**

**Task Paradigm**:

**Search-Oriented**

## B.10 MMCR

---

**MMCR**

**Question ID**: 1
**Question**: Which module's weights are frozen?

---

**Context Modalities**: **Text + Multimodal Document**

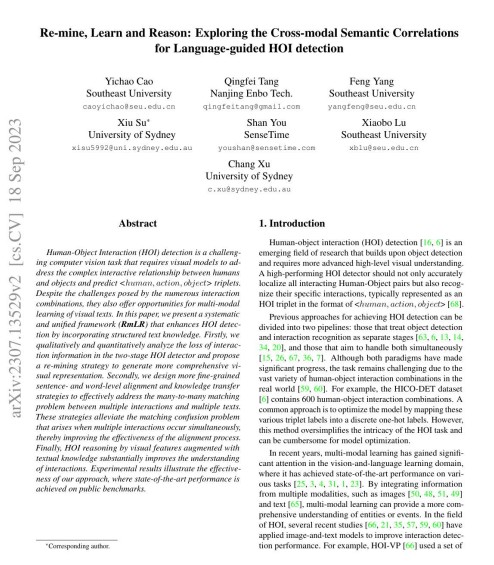

**Covered Areas**:

**Limited**

**Cross-Evidence Reasoning**:

**Limited**

**Task Paradigm**:

**Search-Oriented**

## B.11    AAAR-1.0

---

### AAAR-1.0

**Question ID**: 1902.00751
**Question**: What experiments do you suggest doing? Why do you suggest these experiments?

---

**Context Modalities**: **Text + Multimodal Document**

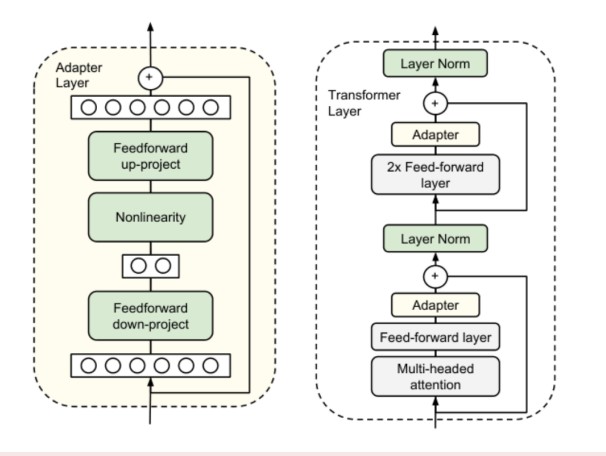

---

**Covered Areas**:

**Limited**

---

**Cross-Evidence Reasoning**:

**Full**

---

**Task Paradigm**:

**Search-Oriented**

# C    DATASET ANNOTATION AND CONSTRUCTION

## C.1    DATA SOURCING AND QUALITY CONTROL

The defective academic papers in our dataset are curated from three primary sources:

- We synthetically injected nine types of errors into papers accepted at ICLR and Nature Communications.
- For papers rejected by ICLR, we identified the shortcomings based on reviewers' comments and categorize them into the same nine error types.
- For accepted ICLR papers, we generated consistency-related errors by cross-referencing their content against the cited literature.

To ensure the quality of each error, all entries underwent a rigorous multistage validation protocol executed by human annotators. For synthetically generated errors, annotators manually embedded them into the source papers following this protocol:

- **Credibility Validation.** Each error must be logically sound and verifiable. For generated errors, annotators first confirm their logical coherence and unambiguity. Flawed error descriptions are revised whenever possible; only irrepairable cases are discarded.
- **Evidence Verification.** All evidence substantiating an error must be either directly traceable to the source document or grounded in established domain-specific knowledge. Annotators are required to meticulously verify the origin and accuracy of all supporting data and background information.
- **Category Classification.** Each error must be accurately classified into one of the nine predefined categories according to their formal definitions. Annotators verify the correctness of the assigned category and reclassify it if necessary.
- **Manuscript Editing.** Upon successful validation, annotators embedded the generated error into the original manuscript by adding, deleting, or modifying relevant text segments as dictated by the error's specification.

This unified and standardized annotation protocol enables the creation of a high-quality dataset of academic papers with curated errors, providing a robust benchmark for evaluating the document scanning and error detection capabilities of MLLMs.

## C.2    ANNOTATION STATISTICS

Initially, we generated or sampled a pool of 3,500 academic paper instances containing potential errors. During the manual annotation phase, following the protocol described above, we discarded 1,700 instances to ensure the logical rigor of the errors, the accuracy of the evidence, and a balanced distribution of categories.

Of the remaining 1,800 instances, 1,541 (85.6%) underwent manual revision. The distribution of these modifications is as follows:

- **535 questions** were rewritten to eliminate ambiguity or to increase their retrieval and reasoning difficulty.
- **1,207 explanations** were revised to correct erroneous evidence references and resolve logical flaws.
- **1,141 instances** underwent category reclassification or manual paper editing. This process served to fix classifications that were inconsistent with our definitions and, for errors generated, to manually inject them into the source papers to create the flawed documents.

## C.3 ANNOTATION EXAMPLES

### C.3.1 CASE 1: DISCARD DIRECTLY

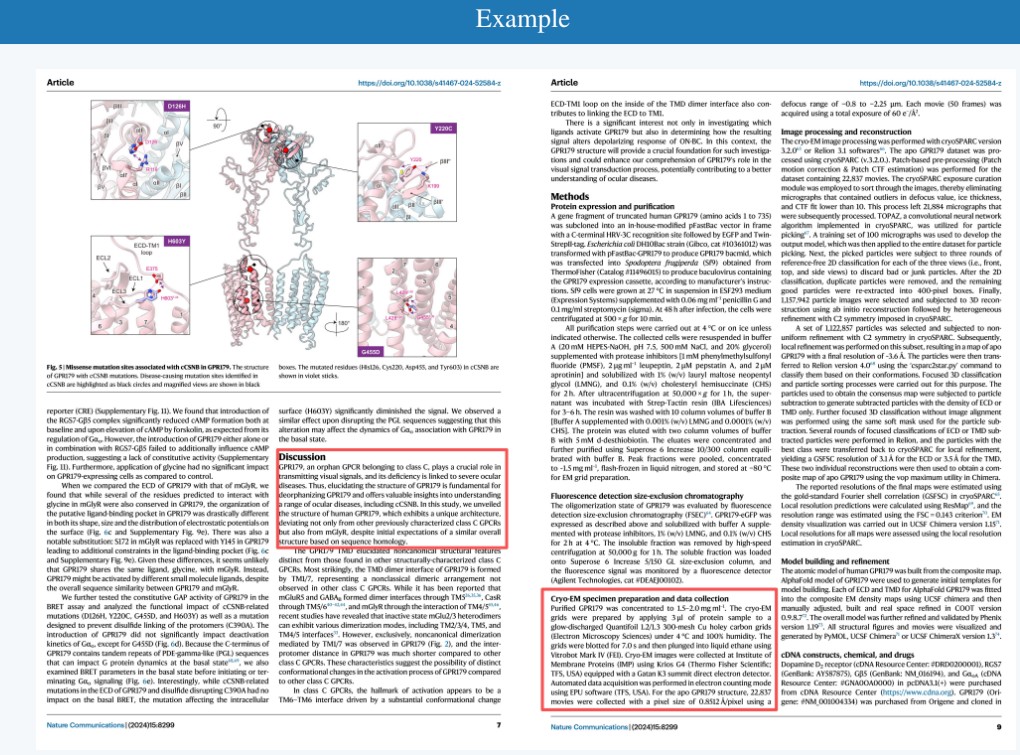

**Question**: Assess whether the conclusions drawn about the protein's functional state and therapeutic applicability are supported by the presented methods and results.

**Explanation**: Edits in the abstract and discussion claim the paper presents an active-state structure that reveals the activation mechanism and provides a roadmap for drug design. This overstates the findings, as the paper repeatedly describes solving the "apo" (unbound) structure and explicitly states the activating ligand is unknown (p.6). To make the error subtle, a contradictory sentence was added to the methods (p.9) claiming a stabilizing agonist was used, but this is falsified by the numerous, unmodified mentions of the "apo GPR179" structure throughout the results and methods.

**Error Type**: IC (Inference & Conclusions)

**Decision**: **Discard**

**Analysis**: Based on the modifications, the revised abstract and conclusion claim that the paper elucidates the protein's "active-state" structure and provides a roadmap for drug design. However, the original text repeatedly states (e.g., on pages 5 and 9) that it is the "apo" (inactive) structure that was resolved, and critically notes on page 6 that the "activating ligand is still unknown". This constitutes a clear RCA-type error, defined by the inconsistent description of a concept within the article. Yet, the large model misclassifies this as an IC-type (Inference & Conclusions) error, which is a significant mistake. Considering that the inconsistency regarding the "active-state" description is overly superficial and obvious, a type of error almost never encountered in actual academic literature, it lacks practical value. Even reclassifying it as an H-type question would be of little significance. Therefore, we have decided to delete this instance.

### C.3.2 CASE 2: MODIFY QUESTION

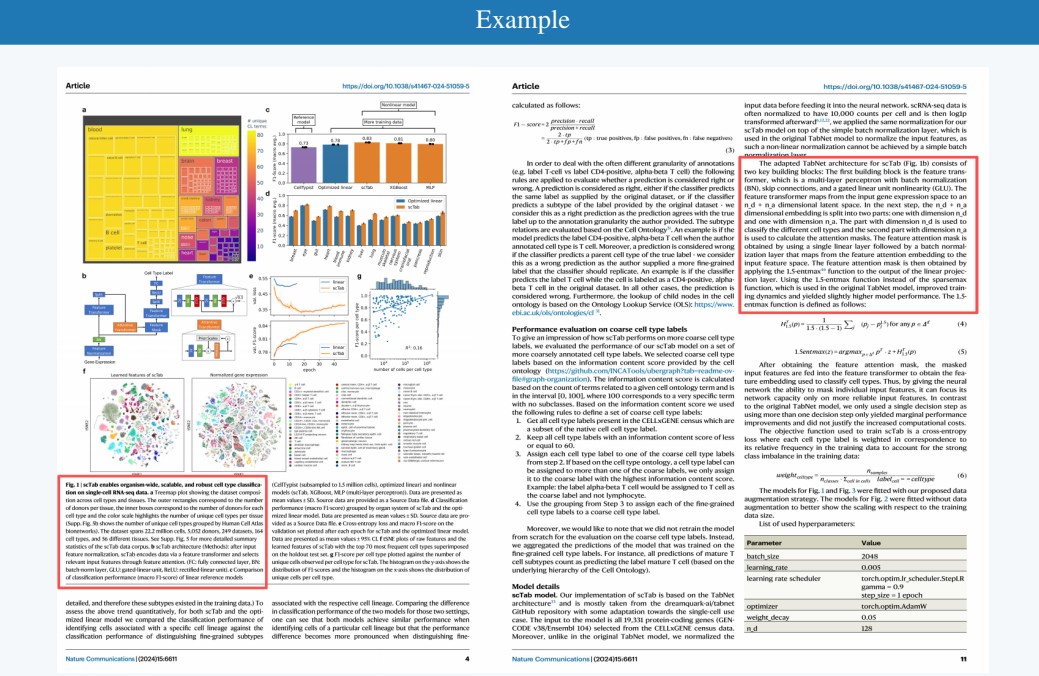

**Question**: Judge whether the mathematical description of the scTab model architecture presents any dimensional mismatches.

**Explanation**: The Methods section on page 11 (and the Figure 1b caption) now states that the model's feature transformer creates an embedding of dimension $n_d$. However, the same paragraph then describes splitting this embedding into two components of dimension $n_d$ and $n_d$ respectively. This is algebraically impossible for any non-zero $n_a$, and the hyperparameter table on page 12 confirms that $n_a$ is set to 64, creating a fundamental dimensional mismatch in the model's description.

**Error Type**: CF (Computation & Formulae)

---

**Before**:
**Question**: Judge whether the mathematical description of the scTab model architecture presents any dimensional mismatches.

---

**Decision**: **Modify**

---

**After**:
**Question**: Assess the Methods section for Computation & Formulae issues.

---

**Analysis**: Based on the error information and the text, the modified model description states that a vector of dimension $n_d$ is split into two parts: one of dimension $n_d$ and another of dimension $n_a$. This is algebraically impossible, as the total dimension ($n_d$) cannot equal the dimension of one of its parts ($n_d$) plus another non-zero part ($n_a$ is set to 64). This constitutes a clear dimensional mismatch, rendering the model's architectural description logically invalid. The original question was overly specific, as it explicitly prompted an assessment of whether the mathematical description of the scTab model architecture contained "any dimensional mismatches". This hint was too detailed, reducing the analytical difficulty for the model. To increase the difficulty, we have revised the question's phrasing to ask only whether the mathematical description of the scTab model architecture presents any problems.

### C.3.3 CASE 3: MODIFY EXPLANATION

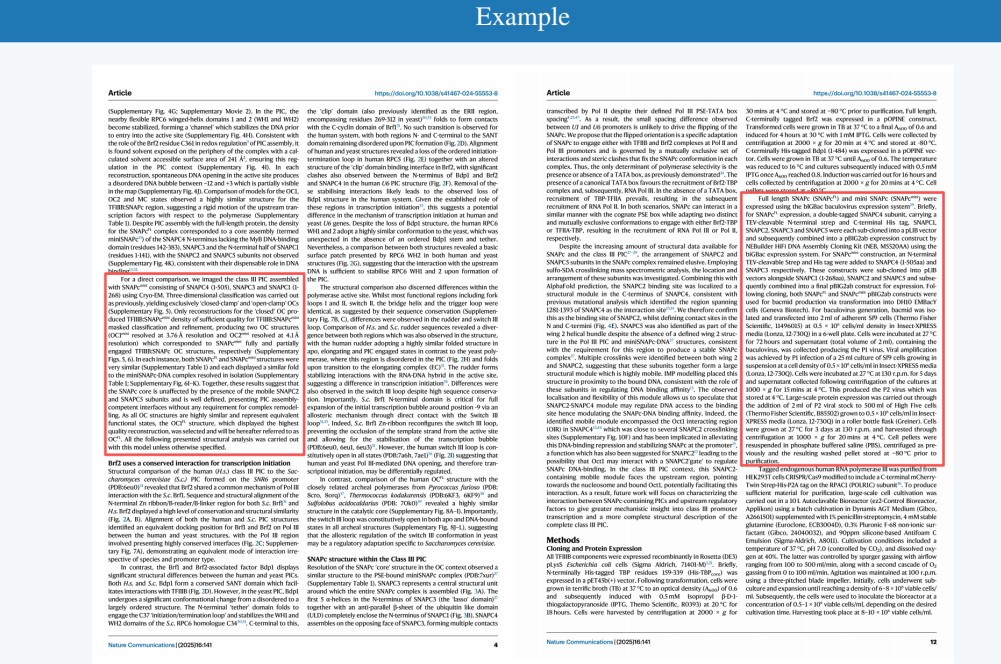

**Question**: Evaluate if the composition of the SNAPcmini construct is consistently defined throughout the paper.

**Explanation**: The results on page 4 state that the assembled SNAPcmini construct includes the SNAPC2 subunit. However, the methods on page 12 describe the construction of SNAPcmini using only SNAPC4, SNAPC3, and SNAPC1, with SNAPC2 explicitly removed from the cloning description. A third conflicting statement on page 6 implies SNAPC2 was expected to be part of the minimal core, creating conceptual and operational inconsistency regarding this key experimental complex.

**Error Type**: RQD (Research Question & Definitions)

---

**Before**:
**Explanation**: ...with SNAPC2 explicitly removed from the cloning description. A third conflicting statement on page 6...

---

**Decision**: **Modify**

---

**After**:
**Explanation**: ...with SNAPC2 explicitly removed from the cloning description.

---

**Analysis**: This instance targets an inconsistency in the operational definition of the SNAPcmini construct. Specifically, the results section states that the assembled SNAPcmini complex includes the SNAPC2 subunit, while the methods section explicitly describes the construction of SNAPcmini using only SNAPC4, SNAPC3, and SNAPC1, with SNAPC2 removed from the cloning procedure. The original explanation additionally referenced a speculative statement regarding SNAPC2's expected presence in the minimal core, which introduced unnecessary ambiguity and reduced the clarity of the definition-level inconsistency. By removing this auxiliary statement, the modified instance focuses on a clear and realistic mismatch between experimental description and implementation, which is representative of RQD-type errors commonly encountered in academic writing.

# D  EXAMPLES FROM SCHOLSCAN

## D.1  RQD (RESEARCH QUESTION AND DEFINITIONS)

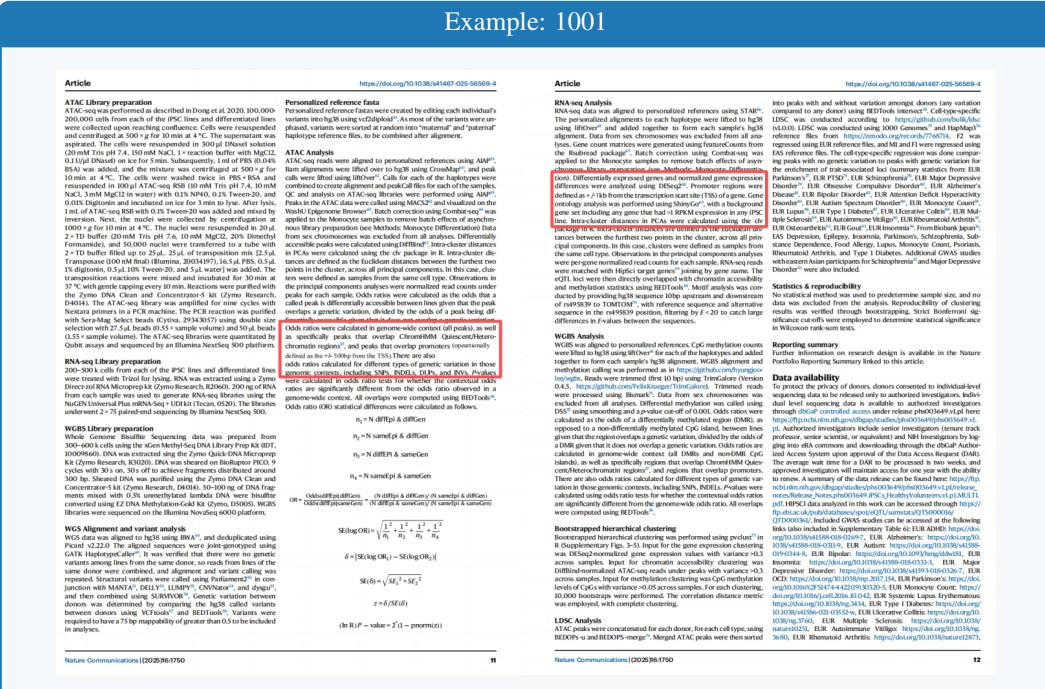

**Question**: Assess the Methods section for Research Question & Definitions issues.

**Explanation**: The definition of a "promoter region", a key analytical construct, is inconsistent across the paper, making the estimand ambiguous. The RNA-seq methods (page 12) define it as +/-1kb from the TSS, the ATAC-seq analysis methods (page 11) define it as +/-500bp from the TSS, and the Results section (page 4) defines it as +/-2kb from the TSS. These three conflicting operational definitions mean that analyses involving "promoters" are not comparable and the construct is insufficiently defined.

**Error Type**: RQD (Research Question & Definitions)

**Type**: Within-Generate

## D.2 DI (DESIGN AND IDENTIFIABILITY)



<p align="center">Example: 1006</p>

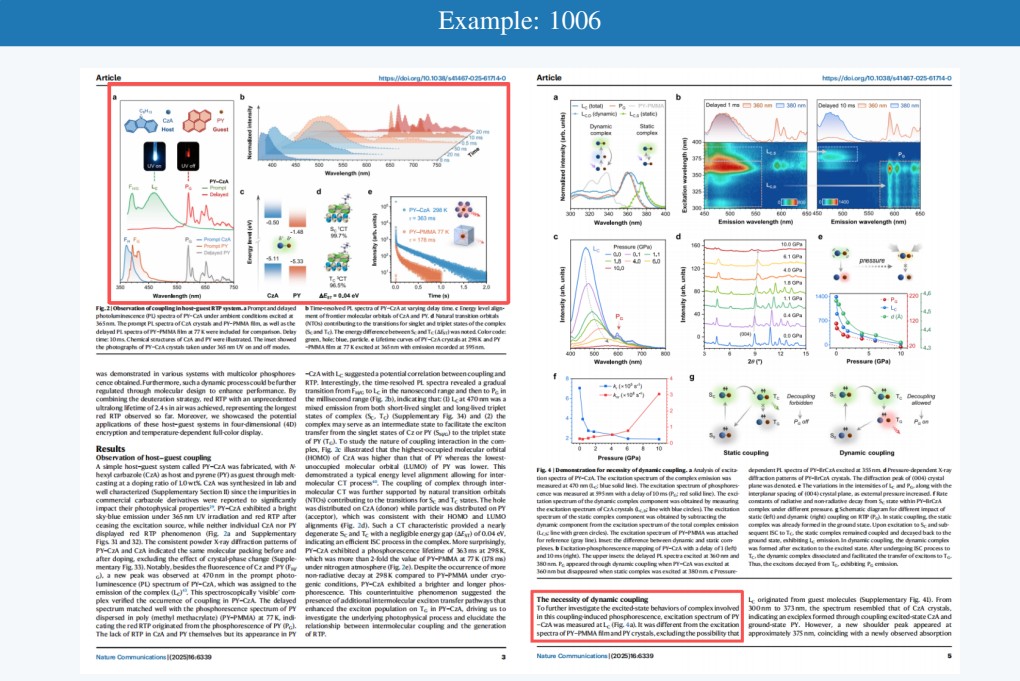

**Question**: Assess the Experiments section for Design & Identifiability issues.

**Explanation**: The paper's core argument is that it identifies a specific "dynamic coupling" pathway as essential for RTP, distinct from a "static coupling" pathway. The edits state that the key experiment (excitation-phosphorescence mapping) cannot distinguish between these two pathways, as the final phosphorescence shows spectral signatures of originating from both. This introduces a structural identification problem: with two potential causal pathways leading to the same outcome and no way to isolate their effects, the claim that the dynamic pathway is the definitive mechanism is not identifiable from the data presented.

**Error Type**: DI (Design & Identifiability)

**Type**: Within-Generate



## D.3    SG (SAMPLING AND GENERALIZABILITY)

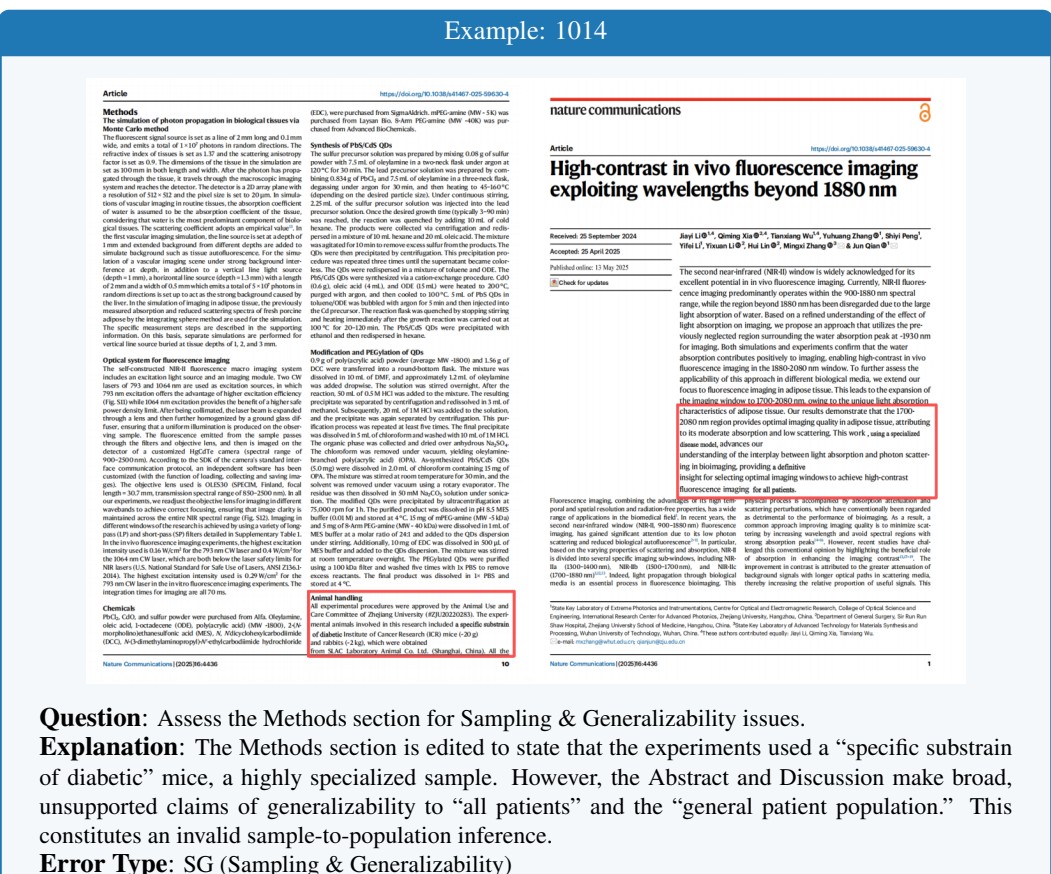

**Question**: Assess the Methods section for Sampling & Generalizability issues.

**Explanation**: The Methods section is edited to state that the experiments used a "specific substrain of diabetic" mice, a highly specialized sample. However, the Abstract and Discussion make broad, unsupported claims of generalizability to "all patients" and the "general patient population." This constitutes an invalid sample-to-population inference.

**Error Type**: SG (Sampling & Generalizability)

**Type**: Within-Generate

## D.4 MO (MEASUREMENT AND OPERATIONALIZATION)

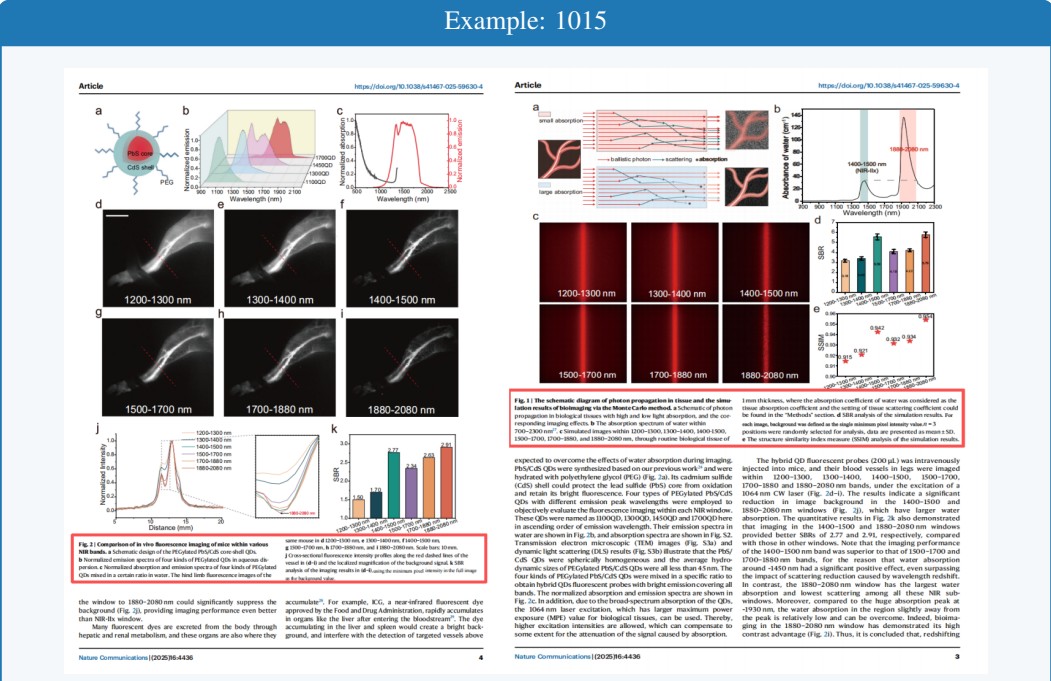

**Example: 1015**

**Question**: Assess the Figures/Tables section for Measurement & Operationalization issues.

**Explanation**: The figure captions on pages 3 and 4 have been edited to specify that the background for Signal-to-Background Ratio (SBR) calculations was defined as the single minimum pixel intensity in the image. This is not a valid or reliable operationalization of the "background" construct, as it is highly susceptible to single-point noise or detector artifacts. This flawed measurement procedure systematically undermines all conclusions based on the SBR metric.

**Error Type**: MO (Measurement & Operationalization)

**Type**: Within-Generate

## D.5 DHP (Data Handling and Preprocessing)

---

**Example: 528**

the likelihood that the output configuration will satisfy the desired constraints. In our case, the constraint is energy, the output of the energy network, must decrease. We specifically employed GBI to directly update the task-net's predictions using gradient signals derived from the energy network.

The implementation of GBI involves three main steps. The task-net, serving as our baseline model, is trained in a supervised manner to predict 3D poses. Next, a structured energy network is trained using the predictions from the task-net as negative samples. Lastly, the trained energy network is employed to iteratively update the task-net's predictions through gradient-based optimization.

**Algorithm 2 Gradient-Based Inference**

**Require:** $(x, y)$: training data (2D inputs and 3D ground-truth outputs)
**Require:** $F_\phi$: task-bot, $E_\theta$: energy network
**Require:** optimizer$_\phi$: optimizer for $E_\theta$
**Require:** $T$: training iterations, $K$: GBI steps.
1:  **Phase 1: train Task-Net**
2:  **for** $t = 1$ to $T$ **do**
3:     Sample batch $B_t = \{(x_i, y_i)\}_{i=1}^N$
4:     Update $\phi$: $\phi \leftarrow \phi - \eta_\phi \nabla_\phi \frac{1}{|B_t|} \sum_{(x_i, y_i) \in B_t} \text{MSE}(F_\phi(x_i) - y_i)$
5:  **end for**
6:  **Phase 2: train energy network**
7:  **for** $t = 1$ to $T$ **do**
8:     Sample batch $B_t = \{(x_i, y_i)\}_{i=1}^N$
9:     Generate $\hat{y}_i = F_\phi(x_i)$ for $x_i \in B_t$
10:    Update $\theta$: $\theta \leftarrow \theta - \eta_\theta \nabla_\theta \frac{1}{|B_t|} \sum_{(x_i, y_i) \in B_t} [E_\theta(x_i, y_i) - E_\theta(x_i, \hat{y}_i)]$
11: **end for**
12: **Phase 3: gradient-based inference**
13: Initialize $\hat{y}_i^{(0)} = F_\phi(x_i)$ for $x_i \in B_t$
14: **for** $k = 1$ to $K$ **do**
15:    Refine $\hat{y}_i$: $\hat{y}_i^{(k)} \leftarrow \hat{y}_i^{(k-1)} - \eta \nabla_{\hat{y}} E_\theta(x_i, \hat{y}_i^{(k-1)})$
16: **end for**

3.3 SETTING

**Datasets** We conduct our experiments on Human3.6M 3D WholeBody dataset (H3WB) (Zhu et al., 2023) and Human3.6M dataset (H36M) (Ionescu et al., 2014). H36M is one of the most widely used datasets for 3D human pose estimation (Zheng et al., 2023; Liu et al., 2024). H3WB extends H36M by providing whole-body annotations using the COCO WholeBody layout, which includes 133 whole-body keypoint annotations, capturing detailed information about hands, face, and feet, making it suitable for tasks that require fine-grained pose estimation. We utilize the ground truth 2D joint locations provided in the dataset to align the 3D and 2D poses. For the H36M dataset, we zero-center the 3D poses around the pelvis joint, following standard protocols and prior work. For the H3WB dataset, we zero-center the 3D poses around the midpoint of the two hip joints.

**Implementation Details** We employ the SimpleBaseline (Martinez et al., 2017), SemGCN (Zhao et al., 2019) and single frame version of VideoPose (Pavllo et al., 2019) as task-nets. For the H3WB dataset, we modify the input and output layers of these task-net to align with data. For the loss-net, we adjusted the SimpleBaseline by modifying the dimensions and depth of the hidden layers. We set the hidden size to 2048 with 2 residual block stages without batch normalization and dropout layers for H3WB and SimpleBaseline task-net for H36M. For the other task-net for H36M, we set the hidden size to 256 with 3 residual block stages with dropout layers. We use separate Adam optimizers (Kingma & Ba, 2015) without learning rate decay for the loss-net and the task-net. All models are trained with a batch size of 1024 for 50 epochs on H36M and a batch size of 64 for 200 epochs on H3WB. For hyperparameter tuning, we employed Bayesian optimization with the wandb sweep tool (Biewald, 2020), aiming to minimize MPJPE for the S9 and S11 in the H36M dataset and PA-MPJPE for the S8 in the H3WB dataset, following the convention of prior works. To avoid overfitting to a specific random seed, we reported the average results from experiments with different random seeds using the optimized hyperparameters.

5

**Question**: Assess the Methods section for Data Handling & Preprocessing issues.

**Explanation**: The reviewer correctly identifies that the authors tuned hyperparameters on the test set. The paper's "Implementation Details" section on page 5 states: "For hyperparameter tuning, we employed Bayesian optimization with the wandb sweep tool (Biewald, 2020), aiming to minimize MPJPE for the S9 and S11 in the H36M dataset and PA-MPJPE for the S8 in the H3WB dataset, following the convention of prior works." According to standard protocols for the H36M dataset, subjects S9 and S11 constitute the test set. Tuning hyperparameters directly on the test set introduces data leakage, leading to an optimistic bias in the reported results and invalidating claims of generalization. This is a critical violation of machine learning best practices and fits the Data Handling & Preprocessing category, as a pipeline choice introduces bias.

**Error Type**: DHP (Data Handling & Preprocessing)

**Type**: Within-Sample

## D.6   CF (COMPUTATION AND FORMULAE)

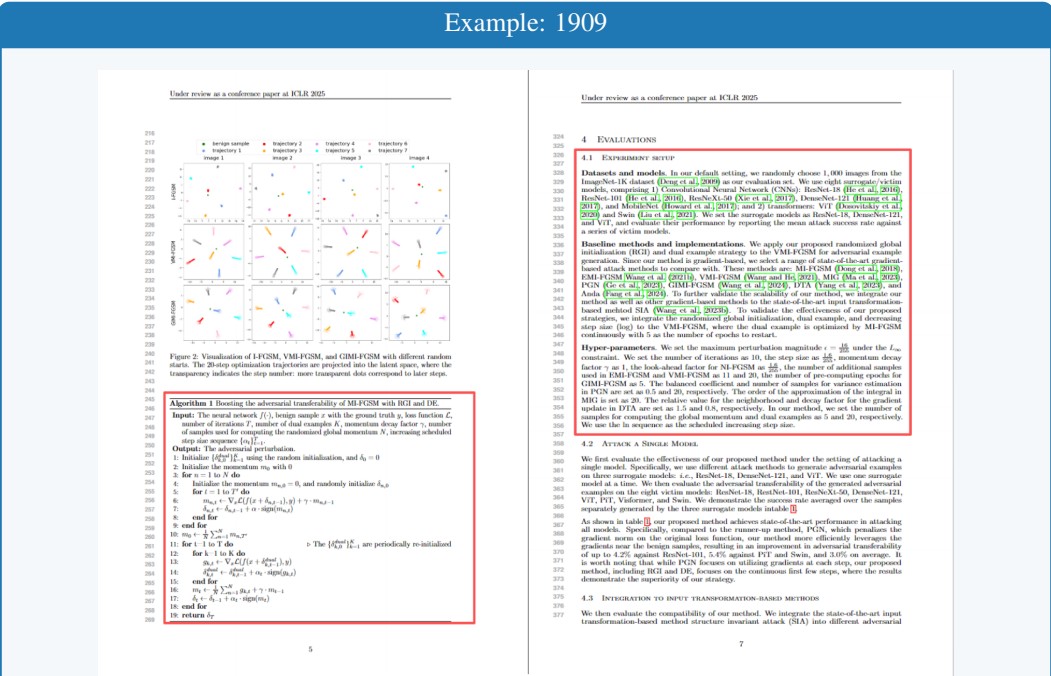

**Question**: Scan the Methods section for Computation & Formulae errors.

**Explanation**: Algorithm 1 on page 5 uses the parameter $T'$ in the loop definition on line 5: **for** $t = 1$ to $T'$ **do**. This parameter determines the number of iterations for the Randomized Global Initialization phase. However, the value of $T'$ is never specified anywhere in the paper, including the "Hyper-parameters" section (Section 4.1 on page 7). An algorithm cannot be implemented or reproduced with an undefined critical parameter. This fits the Computation & Formulae category as an "undefined variable".

**Error Type**: CF (Computation & Formulae)

**Type**: Within-Sample

## D.7   IC (INFERENCE AND CONCLUSIONS)

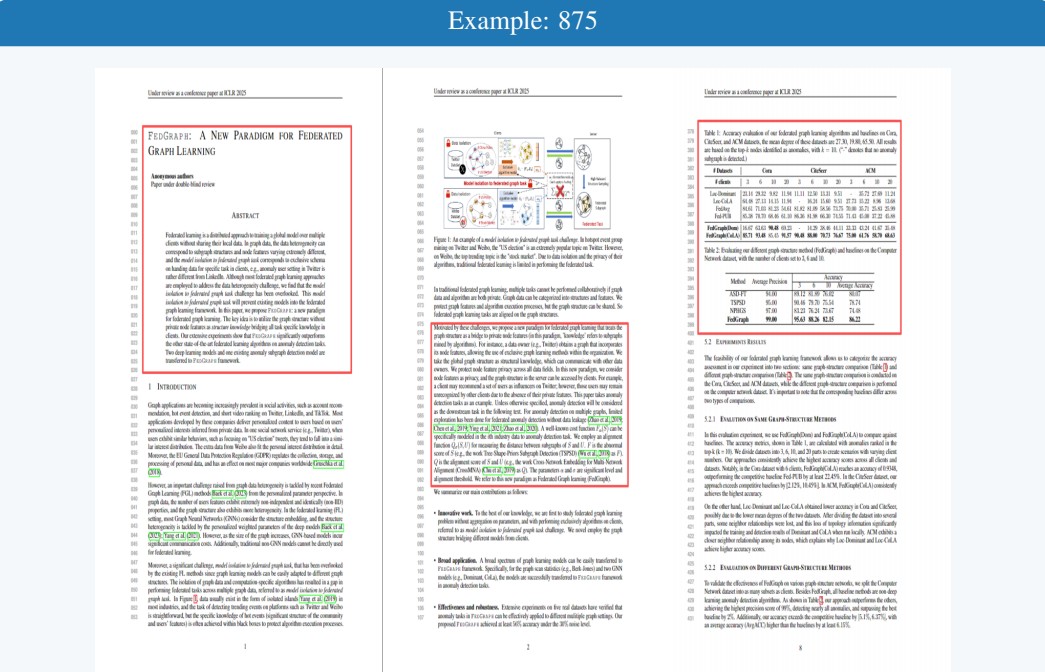

**Question**: Evaluate the Abstract, Introduction, and Experiment sections for issues in Inference & Conclusions.

**Explanation**: The paper's claims of generality are not supported by its evidence. The title and abstract introduce "FedGraph: A New Paradigm for Federated Graph Learning" (page 1), suggesting a broadly applicable framework. However, the methodology is heavily tailored to, and the experiments are exclusively focused on, the single downstream task of anomaly detection. For example, a stated contribution is "Broad application," but this is immediately qualified with "the models are successfully transferred to FEDGRAPH framework in anomaly detection tasks" (page 2). Furthermore, Section 5, "EXPERIMENTS", exclusively reports results on anomaly detection tasks. This discrepancy represents an issue of Inference & Conclusions, as the broad conclusion of having created a new "paradigm" for FGL is an overstatement that exceeds what the narrow experimental results can support.

**Error Type**: IC (Inference & Conclusions)

**Type**: Within-Sample

## D.8 RCA (REFERENTIAL AND CITATION ALIGNMENT)

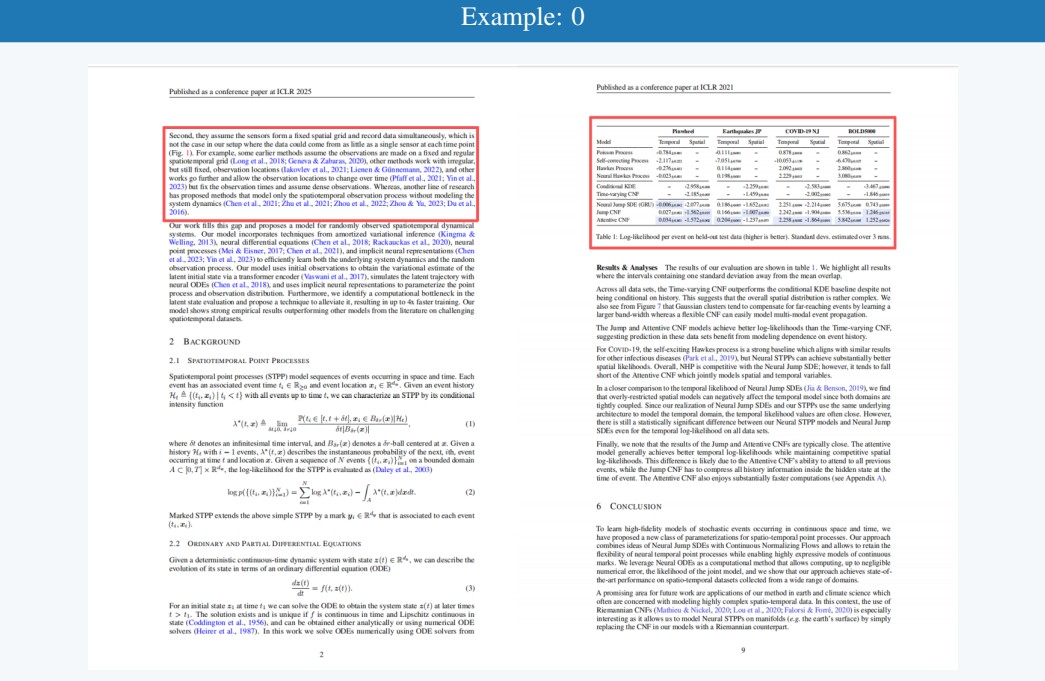

**Question**: Scan the errors in the cited reference Chen et al. (2021).

**Explanation**: The edited P contains a Type H error by misrepresenting the performance of the cited model. P (p. 8) claims that the NSTPP model from Chen et al. (2021) "reported performance comparable to a standard Hawkes process baseline". This contradicts the results in S, where the proposed models (i.e., NSTPP) consistently outperform the Hawkes process baseline, often by a large margin. For example, S (p. 9, Table 1) shows on the BOLD5000 dataset that the "Attentive CNF" model achieves a temporal log-likelihood of 5.842 ± 0.005, which is substantially better than the Hawkes process at 2.860 ± 0.050.

**Error Type**: RCA (Referential & Citation Alignment)

**Type**: Cross-Generate

## D.9 LE (LANGUAGE AND EXPRESSION)

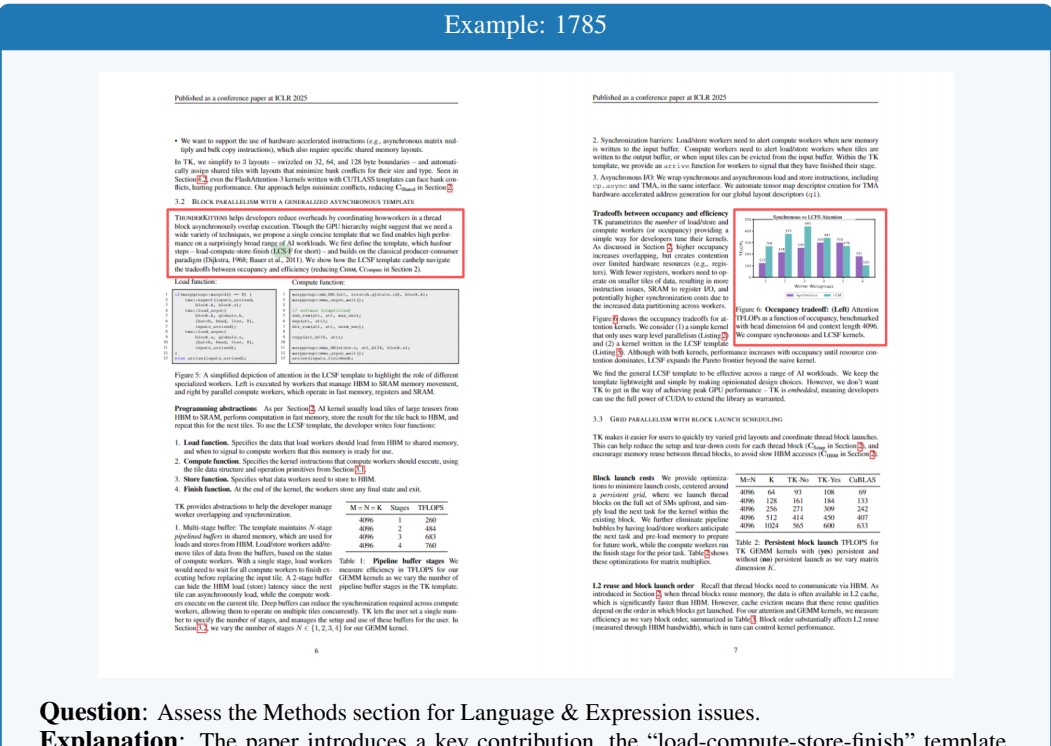

**Question**: Assess the Methods section for Language & Expression issues.

**Explanation**: The paper introduces a key contribution, the "load-compute-store-finish" template, and its acronym "LCSF". This error introduces inconsistencies in this critical term: it is defined as "LCS-F" on page 6, called "LCFS" in a figure title on page 7, and written out in full in the conclusion on page 10, while the original "LCSF" acronym remains elsewhere. This terminological inconsistency for a central, paper-defined concept creates ambiguity and undermines the paper's precision.

**Error Type**: LE (Language & Expression)

**Type**: Within-Generate

# E  HUMAN–MACHINE CONSISTENCY EVALUATION

To evaluate whether GPT-4.1 accurately extracts detailed information from model responses, we conducted a human–machine consistency evaluation. We first randomly sampled 200 questions from the dataset. Then, we invited human experts to analyze the corresponding model-generated responses for these questions and to manually extract key information, including evidence sets, reasoning chains, and the number of unrelated errors. The results are presented in Table 4.

Table 4: Spearman's correlation coefficients among $S$, $S_{\text{location}}$, $S_{\text{reasoning}}$, and $P_{\text{unrelated\_err}}$.

|  | $S$ | $S_{\text{location}}$ | $S_{\text{reasoning}}$ | $P_{\text{unrelated\_err}}$ |
|---|---|---|---|---|
| Correlation Coefficients | 0.841 | 0.806 | 0.842 | 0.954 |

In summary, GPT-4.1 can extract relevant evidence and reasoning steps with considerable accuracy, leading to precise evaluation scores.

In addition, we substituted GPT-4.1 with Qwen3-32B and Gemini 2.5 Flash to independently re-evaluate the same 200 samples (Tables 5 and 6). The results further confirm that our evaluation framework is not dependent on any particular LLM and exhibits strong robustness.

Table 5: Model performance under Qwen3-32B evaluation (scaled by 100).

| Model | Avg. | RQD | DI | SG | MO | DHP | CF | IC | RCA | LE |
|---|---|---|---|---|---|---|---|---|---|---|
| **MLLM (Image Input)** | | | | | | | | | | |
| GPT-5 | 21.2 | 12.6 | 11.8 | 33.5 | 15.4 | 26.6 | 16.0 | 27.1 | 27.0 | 6.4 |
| Gemini 2.5 Pro | 19.7 | 15.0 | 23.2 | 44.7 | 13.2 | 31.4 | 7.8 | 17.3 | 17.8 | 12.8 |
| Doubao-Seed-1.6 | 12.0 | 4.8 | 5.9 | 36.2 | 7.5 | 13.3 | 5.7 | 19.9 | 9.9 | 5.5 |
| Doubao-Seed-1.6-thinking | 11.6 | 5.1 | 6.9 | 28.0 | 7.7 | 14.3 | 10.5 | 15.6 | 10.8 | 4.5 |
| Grok 4 | 4.2 | 0.0 | 1.3 | 20.5 | 2.6 | 5.5 | 1.2 | 3.9 | 2.6 | 0.8 |
| **OCR + LLM (Text Input)** | | | | | | | | | | |
| Gemini 2.5 Pro | 35.0 | 26.6 | 39.6 | 53.9 | 31.4 | 56.2 | 15.9 | 35.6 | 39.0 | 10.0 |
| GPT-5 | 25.3 | 19.0 | 28.3 | 28.3 | 24.1 | 38.8 | 10.3 | 32.2 | 31.8 | 3.1 |
| Grok 4 | 22.6 | 10.7 | 8.9 | 40.6 | 14.1 | 31.3 | 11.1 | 22.6 | 33.6 | 7.3 |
| Doubao-Seed-1.6-thinking | 17.4 | 11.0 | 14.9 | 31.9 | 9.5 | 26.3 | 9.2 | 20.5 | 21.1 | 4.7 |
| Doubao-Seed-1.6 | 15.3 | 6.4 | 9.8 | 31.7 | 10.8 | 22.4 | 8.5 | 21.9 | 17.8 | 2.4 |
| Claude Sonnet 4 | 6.3 | 5.7 | 2.4 | 12.4 | 4.2 | 8.3 | 2.8 | 9.5 | 6.6 | 4.4 |

Table 6: Model performance under Gemini 2.5 Flash evaluation (scaled by 100).

| Model | Avg. | RQD | DI | SG | MO | DHP | CF | IC | RCA | LE |
|---|---|---|---|---|---|---|---|---|---|---|
| **MLLM (Image Input)** | | | | | | | | | | |
| GPT-5 | 20.5 | 11.2 | 12.0 | 32.4 | 17.9 | 27.4 | 10.5 | 25.9 | 27.4 | 3.1 |
| Gemini 2.5 Pro | 14.6 | 9.1 | 11.3 | 34.2 | 11.4 | 28.2 | 4.9 | 12.6 | 14.8 | 5.3 |
| Doubao-Seed-1.6 | 10.9 | 5.3 | 3.9 | 34.0 | 6.1 | 15.6 | 6.1 | 17.9 | 8.0 | 4.7 |
| Doubao-Seed-1.6-thinking | 10.3 | 3.2 | 4.7 | 26.4 | 7.3 | 14.2 | 9.2 | 14.8 | 9.3 | 3.1 |
| Grok 4 | 3.9 | 0.5 | 1.8 | 15.9 | 1.7 | 4.1 | 1.5 | 1.1 | 4.4 | 0.0 |
| **OCR + LLM (Text Input)** | | | | | | | | | | |
| Gemini 2.5 Pro | 30.1 | 20.0 | 30.6 | 47.8 | 27.5 | 47.8 | 11.7 | 30.2 | 36.6 | 5.9 |
| GPT-5 | 23.5 | 15.3 | 21.8 | 26.5 | 23.1 | 37.7 | 7.1 | 31.9 | 31.6 | 2.7 |
| Grok 4 | 20.2 | 9.2 | 7.8 | 36.1 | 11.5 | 32.4 | 8.0 | 20.7 | 30.6 | 5.8 |
| Doubao-Seed-1.6-thinking | 15.9 | 8.4 | 11.1 | 30.9 | 10.1 | 23.7 | 6.3 | 19.9 | 20.2 | 3.5 |
| Doubao-Seed-1.6 | 14.0 | 4.8 | 8.2 | 29.1 | 11.4 | 23.9 | 6.4 | 21.2 | 16.0 | 0.8 |
| Claude Sonnet 4 | 5.7 | 3.8 | 1.4 | 11.1 | 3.9 | 9.4 | 2.0 | 8.7 | 6.5 | 3.1 |

## F  HYPERPARAMETER SENSITIVITY ANALYSIS

We conducted a sensitivity analysis of all four hyperparameters involved in scoring. We varied each independently and re-computed the overall score $S$ across 11 proprietary model configurations (Tables 7, 8, 9, and 10). The results demonstrate that our evaluation metric exhibits strong robustness.

Table 7: Sensitivity under image input: variations of $\lambda$ and $\mu$ (scaled by 100).

| Model | $\lambda$=0.6 | $\lambda$=0.8 | $\lambda$=1.0 | $\mu$=0.85 | $\mu$=0.9 | $\mu$=0.95 |
|---|---|---|---|---|---|---|
| GPT-5 | 19.3 | 19.2 | 19.0 | 18.5 | 19.2 | 19.9 |
| Gemini 2.5 Pro | 15.8 | 15.6 | 15.3 | 15.0 | 15.6 | 16.1 |
| Doubao-Seed-1.6-thinking | 10.4 | 10.2 | 10.0 | 9.9 | 10.2 | 10.5 |
| Doubao-Seed-1.6 | 10.1 | 9.9 | 9.8 | 9.7 | 9.9 | 10.2 |
| Grok 4 | 4.0 | 4.0 | 3.9 | 3.9 | 4.0 | 4.1 |

Table 8: Sensitivity under image input: variations of $\gamma$ and $q$ (scaled by 100).

| Model | $\gamma$=0.4 | $\gamma$=0.6 | $\gamma$=0.8 | $q$=1.0 | $q$=1.5 | $q$=2.0 |
|---|---|---|---|---|---|---|
| GPT-5 | 19.4 | 19.2 | 19.0 | 19.3 | 19.2 | 19.1 |
| Gemini 2.5 Pro | 15.7 | 15.6 | 15.5 | 15.6 | 15.6 | 15.5 |
| Doubao-Seed-1.6-thinking | 10.4 | 10.2 | 10.0 | 10.3 | 10.2 | 10.1 |
| Doubao-Seed-1.6 | 10.1 | 9.9 | 9.8 | 10.0 | 9.9 | 9.9 |
| Grok 4 | 4.0 | 4.0 | 4.0 | 4.0 | 4.0 | 4.0 |

Table 9: Sensitivity under text input: variations of $\lambda$ and $\mu$ (scaled by 100).

| Model | $\lambda$=0.6 | $\lambda$=0.8 | $\lambda$=1.0 | $\mu$=0.85 | $\mu$=0.9 | $\mu$=0.95 |
|---|---|---|---|---|---|---|
| Gemini 2.5 Pro | 30.6 | 30.2 | 29.9 | 29.1 | 30.2 | 31.5 |
| GPT-5 | 22.7 | 22.5 | 22.3 | 21.4 | 22.5 | 23.7 |
| Grok 4 | 21.1 | 20.8 | 20.6 | 20.2 | 20.8 | 21.4 |
| Doubao-Seed-1.6-thinking | 15.6 | 15.3 | 15.0 | 14.8 | 15.3 | 15.8 |
| Doubao-Seed-1.6 | 14.1 | 13.9 | 13.7 | 13.6 | 13.9 | 14.3 |
| Claude Sonnet 4 | 6.0 | 5.9 | 5.8 | 5.6 | 5.9 | 6.1 |

Table 10: Sensitivity under text input: variations of $\gamma$ and $q$ (scaled by 100).

| Model | $\gamma$=0.4 | $\gamma$=0.6 | $\gamma$=0.8 | $q$=1.0 | $q$=1.5 | $q$=2.0 |
|---|---|---|---|---|---|---|
| Gemini 2.5 Pro | 30.7 | 30.2 | 29.9 | 30.5 | 30.2 | 30.1 |
| GPT-5 | 23.4 | 22.5 | 21.9 | 23.0 | 22.5 | 22.2 |
| Grok 4 | 21.0 | 20.8 | 20.7 | 20.9 | 20.8 | 20.8 |
| Doubao-Seed-1.6-thinking | 15.6 | 15.3 | 15.1 | 15.5 | 15.3 | 15.2 |
| Doubao-Seed-1.6 | 14.2 | 13.9 | 13.7 | 14.1 | 13.9 | 13.8 |
| Claude Sonnet 4 | 6.3 | 5.9 | 5.6 | 6.1 | 5.9 | 5.7 |

# G    USE OF LLMS

LLMs were used for language editing and stylistic refinement during manuscript preparation. In addition, Gemini 2.5 Pro was used in a controlled manner to synthesize data for dataset construction. Details are provided in Section 3.2, Appendix A, and Appendix C. All research ideas, experimental design, evaluation protocols, and result analysis were conceived, implemented, and validated entirely by the authors. The use of LLMs did not influence the scientific conclusions of this paper.

