# OpenReview forum: "Not Search, But Scan: Benchmarking MLLMs on Scan-Oriented Academic Paper Reasoning"
_ICLR.cc/2026/Conference — ICLR 2026 Poster_

### Official Review · Reviewer_co8M · 2025-10-30

**Soundness:** 3
**Presentation:** 3
**Contribution:** 3
**Rating:** 6
**Confidence:** 4

**Summary:**

This paper introduces ScholScan, a novel scan-oriented reading benchmark designed to address the limitation of Multimodal Large Language Models (MLLMs) in overlooking holistic and cross-document understanding. The benchmark's questions are initially generated through a sampling and generation process, then manually screened to form the final set. Corresponding metrics are employed to evaluate model performance, and multi-dimensional experiments provide valuable insights into future directions for model development.

**Strengths:**

ScholScan effectively captures a key aspect of human reading: the ability to process full documents or read across multiple articles. The data construction and filtering process is methodologically sound. The evaluation metrics are multi-faceted, reasonable, and well-aligned with human judgments. Consequently, the experimental results offer valuable insights that can guide the future development of Multi-modal Large Language Models (MLLMs).

**Weaknesses:**

The development of ScholScan heavily relies on the use of a Large Language Model (LLM), specifically Gemini-2.5 Pro. Although manual screening and validation were subsequently incorporated to ensure quality, the questions generated by the LLM may inherently differ from those posed by humans.

**Questions:**

- On line 88, there appears to be a typo. "Process-Aware Rvaluation Framework" should be corrected to "Process-Aware Evaluation Framework".
- Some entries in the bibliography seem overly long, which takes up considerable page space. Please consider condensing them.
- In the section on evaluation metrics, several hyper-parameters are introduced (e.g., 0.8, 0.9, 0.6). The rationale for selecting these specific values is not entirely clear. To strengthen the paper, I would recommend including a justification for these choices. For instance, a sensitivity analysis or an ablation study could demonstrate the robustness and appropriateness of the selected parameters.

---

> ### Author Response · Authors · 2025-11-22
> **Rebuttal to Reviewer co8M's Comments**
>
> Below we provide our detailed response to your question; we would be grateful if you could let us know whether this clarifies your concerns.
>
> **Clarifying the Role of Gemini-2.5 Pro in ScholScan Construction**
>
> We would like to clarify that the benchmark questions are not synthetic data that “heavily rely on the use of an LLM.” As described in the paper, ScholScan is constructed from two sources: Sampling and Generation.
>
> For the Sampling subset, the construction steps are:
>
> **Step1.** We first collect real peer-review comments from OpenReview.
>
> **Step2.** Gemini 2.5 Pro is then used only for an initial filtering pass and for drafting candidate questions.
>
> **Step3.** Human experts subsequently perform thorough rewriting and validation of these questions.
>
> Thus, the Sampling questions are fundamentally grounded in human evaluations: Gemini’s role is limited to extracting and drafting, while the underlying review comments and question formulations are rigorously checked and rewritten by human annotators during the labeling process. Characterizing this subset as “heavily relying on the use of LLM” is therefore not accurate.
>
> For the Generation subset, the construction steps are:
>
> **Step1.** Gemini 2.5 Pro reads the paper and proposes candidate edits and questions.
>
> **Step2.** Human experts then integrate, add, or discard these candidates, and further conduct evidence annotation, question rewriting, and related steps.
>
> In this subset, it is true that the initial candidates are produced by an LLM. However, these items are substantially revised by human experts during the annotation process, to the point that the final questions are essentially indistinguishable from those that would have been authored directly by humans.
>
> **On Typos and Bibliography Length**
>
> Thank you for pointing out these typos and stylistic issues. We have corrected “Process-Aware Rvaluation Framework” to “Process-Aware Evaluation Framework” on line 88, and we have also condensed several overly long bibliography entries in the revised version to save space.
>
> **Sensitivity analysis of the hyperparameters**
>
> We conduct a sensitivity analysis of the four hyperparameters discussed In our paper. In the main paper, they are set to λ=0.8, μ=0.9, γ=0.5, and q=1.6. We vary each hyperparameter individually while keeping the others fixed, and obtain the results below:
>
> | Model                          | λ=0.6 | λ=0.8 | λ=1.0 | μ=0.85 | μ=0.9 | μ=0.95 | γ=0.4 | γ=0.6 | γ=0.8 | q=1.0 | q=1.5 | q=2.0 |
> |--------------------------------|-------|-------|-------|--------|-------|--------|-------|-------|-------|-------|-------|-------|
> | GPT-5_Image                    | 19.3  | 19.2  | 19.0  | 18.5   | 19.2  | 19.9   | 19.4  | 19.2  | 19.0  | 19.3  | 19.2  | 19.1  |
> | Gemini 2.5 Pro_Image           | 15.8  | 15.6  | 15.3  | 15.0   | 15.6  | 16.1   | 15.7  | 15.6  | 15.5  | 15.6  | 15.6  | 15.5  |
> | Doubao-Seed-1.6-Thinking_Image | 10.4  | 10.2  | 10.0  | 9.9    | 10.2  | 10.5   | 10.4  | 10.2  | 10.0  | 10.3  | 10.2  | 10.1  |
> | Doubao-Seed-1.6_Image          | 10.1  | 9.9   | 9.8   | 9.7    | 9.9   | 10.2   | 10.1  | 9.9   | 9.8   | 10.0  | 9.9   | 9.9   |
> | Grok 4_Image                   | 4.0   | 4.0   | 3.9   | 3.9    | 4.0   | 4.1    | 4.0   | 4.0   | 4.0   | 4.0   | 4.0   | 4.0   |
> | Gemini 2.5 Pro_Text            | 30.6  | 30.2  | 29.9  | 29.1   | 30.2  | 31.5   | 30.7  | 30.2  | 29.9  | 30.5  | 30.2  | 30.1  |
> | GPT-5_Text                     | 22.7  | 22.5  | 22.3  | 21.4   | 22.5  | 23.7   | 23.4  | 22.5  | 21.9  | 23.0  | 22.5  | 22.2  |
> | Grok 4_Text                    | 21.1  | 20.8  | 20.6  | 20.2   | 20.8  | 21.4   | 21.0  | 20.8  | 20.7  | 20.9  | 20.8  | 20.8  |
> | Doubao-Seed-1.6-Thinking_Text  | 15.6  | 15.3  | 15.0  | 14.8   | 15.3  | 15.8   | 15.6  | 15.3  | 15.1  | 15.5  | 15.3  | 15.2  |
> | Doubao-Seed-1.6_Text           | 14.1  | 13.9  | 13.7  | 13.6   | 13.9  | 14.3   | 14.2  | 13.9  | 13.7  | 14.1  | 13.9  | 13.8  |
> | Claude Sonnet 4_Text           | 6.0   | 5.9   | 5.8   | 5.6    | 5.9   | 6.1    | 6.3   | 5.9   | 5.6   | 6.1   | 5.9   | 5.7   |
>
> (Here, Image and Text denote the input configurations.)
>
> Our conclusions are:
>
> 1. Across all tested values, the relative ranking of the 11 proprietary model configurations remains unchanged. This indicates that **the evaluation metric is highly robust** at the level of comparative conclusions.
>
> 2. For each individual model, the score variation across different hyperparameter settings is very small, and much smaller than both the gap between humans and the strongest current MLLMs and the differences between models. In other words, **changing the hyperparameters within a reasonable range does not alter our core conclusions**.
>
> If you still have questions about the interpretability or other aspects of the evaluation metric, we kindly refer you to our response to Reviewer 6ouw, specifically the section “Systematic Explanation of Our Evaluation Metric Design”.

---

> > ### Comment · Reviewer_co8M · 2025-11-25
> >
> > Thanks to the response, I will keep the positive score.

---

> > > ### Author Response · Authors · 2025-11-25
> > >
> > > Thanks for your reply! If there are any remaining questions or points requiring further clarification, we would be pleased to engage in further discussion.

---

> ### Author Response · Authors · 2025-11-22
> **Rebuttal to Reviewer co8M's Comments (2)**
>
> ***
> At last, we sincerely appreciate your valuable feedback. If our responses address your concerns, we would be deeply grateful if you could kindly consider raising the score further to support our work. Thank you very much!

---

### Official Review · Reviewer_a2Tc · 2025-10-30

**Soundness:** 3
**Presentation:** 3
**Contribution:** 3
**Rating:** 6
**Confidence:** 2

**Summary:**

The paper introduces ScholScan, a benchmark designed to evaluate multimodal large language models (MLLMs) on scan-oriented scholarly reasoning: instead of answering prespecified, “search-oriented” questions, models receive one or more full academic papers and must proactively scan the entire document to uncover consistency issues. Key aspects:
- Task formulation: “target-absent” audit-style queries (e.g., “Assess the Methods section for Measurement & Operationalization issues”) requiring evidence construction across the whole paper.
- Dataset: 1,800 questions across 715 papers from 13 natural-science domains, covering 9 error families (Research Question & Definitions; Design & Identifiability; Sampling & Generalizability; Measurement & Operationalization; Data Handling & Preprocessing; Computation & Formulae; Inference & Conclusions; Referential & Citation Alignment; Language & Expression). Instances come from two sources: (i) LLM-edited accepted papers (synthetic error injection) and (ii) sampling verifiable issues from ICLR public reviews of rejected papers. Cross-paper citation inconsistencies are generated.
- Process-aware evaluation: A unified evaluation pipeline parses open-ended model outputs into existence, evidence localization (Dice-like score with over-reporting penalty), reasoning-chain prefix-match, and a penalty for unrelated/hallucinated errors.
- Experiments: 15 models with 24 input settings (image vs OCR text) and 8 RAG frameworks. Main findings: (a) overall performance is weak; (b) reasoning-optimized models perform better; (c) text (OCR) inputs often outperform image inputs, but images can matter for formula/table-heavy tasks; (d) standard RAG brings little benefit; an RL-based visual-centric RAG (VRAG-RL) shows the most promise on image inputs.
- Analyses: Correlations across error types suggest latent “skill” dimensions; evidence-and-reasoning workload correlates with failures; failure mode breakdown (omission vs hallucination); human–machine agreement for the judge-extractor shows strong Spearman correlations.

**Strengths:**

- A New "Scan-Oriented" Task Paradigm: The paper formalizes a novel and challenging task setting that moves beyond simple question-answering to require exhaustive, document-level understanding and consistency checking without pre-specified targets.
- The ScholScan Benchmark: A comprehensive and large-scale benchmark consisting of 1,800 questions derived from 715 scientific papers across 13 natural science domains. The benchmark covers 9 distinct families of scientific errors, from issues in experimental design to referential inconsistencies.
- High-Quality Data and Annotation: The dataset was constructed through a rigorous process involving both LLM-based error generation/sampling and meticulous verification by 10 domain experts, ensuring the errors are subtle, realistic, and verifiable. The annotations include not just the final answer but also evidence locations and reasoning traces.
- A Process-Aware Evaluation Framework: The authors propose a detailed evaluation protocol and a composite scoring metric that assesses a model's performance on multiple dimensions: correctly identifying an error's existence, localizing all relevant evidence, completing the correct reasoning chain, and avoiding the hallucination of unrelated errors.
- Extensive Empirical Analysis: The paper presents a thorough evaluation of 15 state-of-the-art MLLMs (across 24 configurations) and 8 Retrieval-Augmented Generation (RAG) methods. The results show that all current models, including proprietary ones like GPT-5 and Gemini 2.5 Pro, perform poorly on this task, and that standard RAG techniques fail to provide any significant improvement, highlighting the unique challenges posed by the scan-oriented paradigm.

**Weaknesses:**

- Dependence on LLM-based Evaluation: The evaluation protocol relies on GPT-4.1 to parse model responses and extract structured information (evidence, reasoning steps). While the authors validate this with a human-machine consistency check (Appendix F), which shows strong correlation, this dependency introduces a potential point of failure or bias. The evaluation of one model is contingent on the capabilities of another, which is a common but not ideal practice. The evaluation could be influenced by the specific style or formatting of the model being evaluated.
- Limited Scope of "Generated" Errors: The data generation process relies on an LLM (Gemini 2.5 Pro) to inject errors into high-quality papers. While curated by experts, these synthetic errors might have subtle, systematic patterns that differ from genuine human errors. The benchmark could be further strengthened by including more examples of errors "sampled" from real-world peer reviews or errata, although the authors are transparent about their methodology.
- Ambiguity in the "Reasoning Process Score": The reasoning score (`S_reasoning`) is based on a prefix match of the reasoning chain. This assumes a single, linear "gold" reasoning path. However, a valid logical argument could potentially be structured in different ways or use slightly different intermediate steps. This metric might unduly penalize models that arrive at the correct conclusion through a logically sound but differently structured argument.

**Questions:**

- Beyond RAG - Other Potential Architectures: The RAG analysis is excellent. However, have the authors considered or could they comment on other architectural approaches that might be better suited for "scan" tasks? For example, models with explicit memory B, graph-based representations of the document's claims and evidence, or iterative self-refinement loops where the model repeatedly scans the paper to build and verify a "consistency map." A brief discussion in the conclusion could enrich the paper.
- LLM-as-Judge Robustness: Regarding the GPT-4.1-based evaluation, did the authors observe any failure modes? For instance, did the evaluator struggle with particularly verbose or poorly structured responses from certain open-source models? Providing some qualitative examples of the evaluator's performance could further strengthen confidence in this method.
- Feasibility of Human Evaluation: The paper mentions high agreement between the GPT-4.1 pipeline and expert annotations on a subset. Given the complexity of the task, what would be the estimated cost or time for a full human evaluation of a single model's output on the benchmark? A brief comment on this could help contextualize the necessity of the automated pipeline.
- Suggestion on Score Calibration: The final scores are quite low across the board (the best model scores ~30). This is a strong testament to the benchmark's difficulty. However, it can make it hard to differentiate between poorly performing models. Have the authors considered if a simpler, "binary" success metric (e.g., did the model identify the correct error type and at least one piece of correct evidence?) could complement the main score and provide another view on performance, especially for the lower-performing models? This is not a required change, but a suggestion for a potential additional analysis.

---

> ### Author Response · Authors · 2025-11-22
> **Rebuttal to Reviewer a2Tc's Comments (1)**
>
> Below we provide our detailed response to your question; we would be grateful if you could let us know whether this clarifies your concerns.
>
> **On the Dependence on LLM-Based Evaluation**
>
> We re-evaluated part of the main experiments by replacing GPT-4.1 with Gemini 2.5 Flash and Qwen3-32B under exactly the same prompts. Across these three evaluators, all proprietary models exhibit highly consistent score trends and relative rankings:
>
> For Qwen3-32B:
>
> | Models               | Score | RQD    | DI    | SG    | MO    | DHP    | CF    | IC    | RCA    | LE    |
> |----------------------|-------|------|------|------|------|------|------|------|------|------|
> | Gemini 2.5 Pro_Image     | 19.7  | 15.0 | 23.2 | 44.7 | 13.2 | 31.4 | 7.8  | 17.3 | 17.8 | 12.8 |
> | GPT-5_Image       | 21.2  | 12.6 | 11.8 | 33.5 | 15.4 | 26.6 | 16.0 | 27.1 | 27.0 | 6.4  |
> | Grok 4_Image       | 4.2   | 0.0  | 1.3  | 20.5 | 2.6  | 5.5  | 1.2  | 3.9  | 2.6  | 0.8  |
> | Doubao-Seed-1.6-thinking_Image        | 11.6  | 5.1  | 6.9  | 28.0 | 7.7  | 14.3 | 10.5 | 15.6 | 10.8 | 4.5  |
> | Doubao-Seed-1.6_Image  | 12.0  | 4.8  | 5.9  | 36.2 | 7.5  | 13.3 | 5.7  | 19.9 | 9.9  | 5.5  |
> | Gemini 2.5 Pro_Text      | 35.0  | 26.6 | 39.6 | 53.9 | 31.4 | 56.2 | 15.9 | 35.6 | 39.0 | 10.0 |
> | GPT-5_Text        | 25.3  | 19.0 | 28.3 | 28.3 | 24.1 | 38.8 | 10.3 | 32.2 | 31.8 | 3.1  |
> | Claude Sonnet 4_Text      | 6.3   | 5.7  | 2.4  | 12.4 | 4.2  | 8.3  | 2.8  | 9.5  | 6.6  | 4.4  |
> | Grok 4_Text        | 22.6  | 10.7 | 8.9  | 40.6 | 14.1 | 31.3 | 11.1 | 22.6 | 33.6 | 7.3  |
> | Doubao-Seed-1.6-thinking_Text         | 17.4  | 11.0 | 14.9 | 31.9 | 9.5  | 26.3 | 9.2  | 20.5 | 21.1 | 4.7  |
> | Doubao-Seed-1.6_Text   | 15.3  | 6.4  | 9.8  | 31.7 | 10.8 | 22.4 | 8.5  | 21.9 | 17.8 | 2.4  |
>
> For Gemini 2.5 Flash:
>
> | Models               | Score | RQD    | DI    | SG    | MO    | DHP    | CF    | IC    | RCA    | LE    |
> |--------------------------------|-------|------|------|------|------|------|------|------|------|------|
> | Gemini 2.5 Pro_Image            | 14.6  | 9.1  | 11.3 | 34.2 | 11.4 | 28.2 | 4.9  | 12.6 | 14.8 | 5.3  |
> | GPT-5_Image                     | 20.5  | 11.2 | 12.0 | 32.4 | 17.9 | 27.4 | 10.5 | 25.9 | 27.4 | 3.1  |
> | Grok 4_Image                    | 3.9   | 0.5  | 1.8  | 15.9 | 1.7  | 4.1  | 1.5  | 1.1  | 4.4  | 0.0  |
> | Doubao-Seed-1.6-thinking_Image | 10.3  | 3.2  | 4.7  | 26.4 | 7.3  | 14.2 | 9.2  | 14.8 | 9.3  | 3.1  |
> | Doubao-Seed-1.6_Image          | 10.9  | 5.3  | 3.9  | 34.0 | 6.1  | 15.6 | 6.1  | 17.9 | 8.0  | 4.7  |
> | Gemini 2.5 Pro_Text            | 30.1  | 20.0 | 30.6 | 47.8 | 27.5 | 47.8 | 11.7 | 30.2 | 36.6 | 5.9  |
> | GPT-5_Text                     | 23.5  | 15.3 | 21.8 | 26.5 | 23.1 | 37.7 | 7.1  | 31.9 | 31.6 | 2.7  || GPT-5_Text                     | 23.5  | 15.3 | 21.8 | 26.5 | 23.1 | 37.7 | 7.1  | 31.9 | 31.6 | 2.7  |
> | Claude Sonnet 4_Text           | 5.7   | 3.8  | 1.4  | 11.1 | 3.9  | 9.4  | 2.0  | 8.7  | 6.5  | 3.1  |
> | Grok 4_Text                    | 20.2  | 9.2  | 7.8  | 36.1 | 11.5 | 32.4 | 8.0  | 20.7 | 30.6 | 5.8  |
> | Doubao-Seed-1.6-thinking_Text  | 15.9  | 8.4  | 11.1 | 30.9 | 10.1 | 23.7 | 6.3  | 19.9 | 20.2 | 3.5  |
> | Doubao-Seed-1.6_Text           | 14.0  | 4.8  | 8.2  | 29.1 | 11.4 | 23.9 | 6.4  | 21.2 | 16.0 | 0.8  |
>
> (Here, Image and Text denote the input configurations.)
>
> This suggests that our conclusions do not depend on any single evaluation model. Together with the human–model agreement results in Appendix F, this provides two complementary pieces of evidence for the robustness of our evaluation pipeline.
>
> Second, our setup does not follow the traditional LLM-as-a-judge paradigm where the LLM directly outputs a final score. Instead, the evaluator is instructed to first produce structured annotations (e.g., evidence indices, lists of reasoning steps, flags for unrelated errors), and we then compute all metrics using a unified scoring formula. In practice, this decoupled design substantially reduces the impact of any single evaluator’s subjective preferences or scoring scale, and also mitigates the influence of model-specific style factors on the final scores.

---

> ### Author Response · Authors · 2025-11-22
> **Rebuttal to Reviewer a2Tc's Comments (2)**
>
> **On the Scope of LLM-Generated Errors**
>
> We agree that if a benchmark relied solely on raw LLM-injected edits and only underwent light human filtering, it could indeed end up reflecting the generator model’s own biases more than the error patterns that actually occur in real research writing. However, this is not how ScholScan is constructed. As described in the paper, our data consist of two parallel subsets: Sampling and Generation.
>
> For the Sampling subset, the construction process is:
>
> **Step1.** Collect real peer-review comments and their corresponding papers from OpenReview;
>
> **Step2.** Use Gemini 2.5 Pro only for coarse filtering and drafting initial question candidates;
>
> **Step3.** Have human experts substantially rewrite, merge, and verify the questions and evidence.
>
> Thus, the Sampling subset is fundamentally rooted in genuine error patterns described in human peer reviews, with Gemini serving only as an auxiliary tool for extraction and initial drafting. The final error forms and their distribution are primarily determined by our human annotation guidelines, so it would be inaccurate to regard this subset as “automatically generated by an LLM.”
> For the Generation subset, the construction process is:
>
> **Step1.** Use Gemini 2.5 Pro to propose candidate edits and corresponding questions on a given paper;
>
> **Step2.** Have human experts apply multiple rounds of filtering, necessary rewriting and consolidation, and add evidence annotations and error-type labels to align the samples with our predefined nine error categories and annotation protocol.
>
> In other words, within the Generation subset, Gemini’s outputs are not simply “carefully selected” and then used as-is; they are actively filtered and reshaped. A large portion of candidates are discarded, and the retained ones are rewritten and reorganized according to our error taxonomy and annotation scheme. As a result, the final error forms and their frequency distribution are shaped primarily by our human-defined error taxonomy and annotation process, rather than directly inheriting the raw output distribution of a single LLM.
>
> Overall, the Sampling subset introduces error sources grounded in real peer review, while the Generation subset actively reshapes LLM-proposed candidates through the annotation pipeline. Together, they mitigate the concern that ScholScan would merely reflect the idiosyncrasies of one specific generator model. We also agree that further expanding the portion of examples drawn directly from real peer review or errata in future versions could further strengthen the long-term validity of the benchmark.
>
> **On Ambiguity in the "Reasoning Process Score"**
>
> We agree that, in principle, the prefix-matching-based S_reasoning implicitly assumes the existence of a “canonical” reasoning path, and may under-estimate some alternative reasoning structures that are logically sound but organized differently.
>
> In our current task, however, we believe the practical impact of this issue is limited, for several reasons:
>
> 1. The prefix-matching scheme used for scoring allows for a certain degree of paraphrasing and parallelization of reasoning steps; it does not require the model’s answer to follow a fully linear, fine-grained chain.
>
> 2. In our human-model agreement study, we indeed observed almost no cases where a “completely different but equally correct” reasoning path was systematically under-scored.
>
> 3. From the task design itself, this phenomenon is also inherently constrained in ScholScan: each question is built around a set of multiple, closely related evidence locations, and any valid reasoning chain must return to these locations and rely on the corresponding evidence. Our evidence annotation is intended to be as complete as possible, precisely to avoid situations where a model can legitimately base its reasoning on entirely different evidence. Under this setup, the space for disjoint yet equally sufficient alternative reasoning paths is relatively limited, especially compared with more open-ended complex mathematical reasoning tasks.
>
> Thank you for raising this point, and we plan to further refine the annotation and scoring of reasoning chains in future work.

---

> ### Author Response · Authors · 2025-11-22
> **Rebuttal to Reviewer a2Tc's Comments (3)**
>
> **On Architectural Alternatives to RAG for Scholarly Scan Tasks**
>
> We are very pleased that you are interested in discussing architectural paradigms for enhancing current methods; designing better solutions for the scan-oriented task setting is exactly the direction we are actively exploring. On this topic, we would like to share a few of our current thoughts:
>
> 1. When we directly apply agentic RAG systems designed for long- or multi-document settings (such as ViDoRAG, M3DocRAG, and SimpleDoc) to ScholScan, we do not yet observe consistent or substantial performance gains. In contrast, their complex multi-step retrieval and planning procedures noticeably increase inference latency and computation cost. We therefore tend to believe that these systems and their predefined workflows are not fully aligned with the task characteristics of ScholScan’s scan-oriented setting.
>
> 2. We agree that graph-based representations of a paper’s claims and evidence may be promising for some error types. However, we see at least two limitations: (i) research papers are highly multimodal and layout-heavy documents, and current multimodal knowledge-graph techniques still have limited coverage and fidelity; (ii) compressing an entire paper into a graph often discards a large amount of “semantically weak but reasoning-critical” detail, such as concrete experimental numbers, confidence intervals, and hyperparameter settings. For this reason, we view graph representations as better suited as auxiliary modules for particular error types, rather than as a single unified architecture that could robustly handle all nine error categories.
>
> Overall, we strongly agree with the richer architectural directions you suggested. Although these ideas go beyond the main focus of this work: namely, establishing a unified evaluation protocol and baseline analysis. They are precisely the kinds of follow-up research directions we hope ScholScan will motivate.
>
> [1] Qiuchen Wang, Ruixue Ding, Zehui Chen, Weiqi Wu, Shihang Wang, Pengjun Xie, and Feng Zhao. ViDoRAG: Visual Document Retrieval-Augmented Generation via Dynamic Iterative Reasoning Agents. In Proceedings of EMNLP 2025, 2025. (arXiv:2502.18017)
>
> [2] Jaemin Cho, Debanjan Mahata, Ozan Irsoy, Yujie He, and Mohit Bansal. M3DocRAG: Multi-modal Retrieval is What You Need for Multi-page Multi-document Understanding. arXiv preprint arXiv:2411.04952, 2024.
>
> [3] Chelsi Jain, Yiran Wu, Yifan Zeng, Jiale Liu, Shengyu Dai, Zhenwen Shao, Qingyun Wu, and Huazheng Wang. SimpleDoc: Multi-Modal Document Understanding with Dual-Cue Page Retrieval and Iterative Refinement. arXiv preprint arXiv:2506.14035, 2025. (to appear in EMNLP 2025)
>
> [4] Haoran Luo, Haihong E, Guanting Chen, Yandan Zheng, Xiaobao Wu, Yikai Guo, Qika Lin, Yu Feng, Zemin Kuang, Meina Song, Yifan Zhu, and Luu Anh Tuan. HyperGraphRAG: Retrieval-Augmented Generation with Hypergraph-Structured Knowledge Representation. arXiv preprint arXiv:2503.21322, 2025.
>
> [5] Bernal Jiménez Gutiérrez, Yiheng Shu, Yu Gu, Michihiro Yasunaga, and Yu Su. HippoRAG: Neurobiologically Inspired Long-Term Memory for Large Language Models. In Advances in Neural Information Processing Systems (NeurIPS), 2024. (arXiv:2405.14831)
>
> **On LLM-as-Judge Robustness**
>
> We chose GPT-4.1 because, before constructing the benchmark, we conducted comparative experiments using multiple LLMs as evaluators, and GPT-4.1 achieved the highest agreement with human annotations. Based on the human-model agreement study in this paper, we systematically examined the cases where human judgments and GPT-4.1 evaluations disagreed and did not observe any consistently triggered failure modes; most discrepancies are concentrated in relatively fine-grained counting differences, such as the exact number of evidence items or slight variations in how reasoning steps are segmented. Therefore, we regard the overall evaluation pipeline as stable and reliable.
>
> We will add a subsection in the appendix of the revised manuscript providing several illustrative examples to more transparently demonstrate the evaluator’s behavior.

---

> ### Author Response · Authors · 2025-11-22
> **Rebuttal to Reviewer a2Tc's Comments (4)**
>
> **On Feasibility of Human Evaluation**
>
> We are pleased to report that, when conducting the consistency study between the GPT-4.1 evaluation pipeline and human experts, we explicitly recorded the annotation time.
>
> In this experiment, the experts annotated 200 responses, with a total annotation time of 41.4 hours, which corresponds to an average of 12.4 minutes per response. Extrapolating from this, a full human evaluation of a single model under a single input configuration on the 1,800-question benchmark would require approximately 1,800 × 12.4 min ≈ 372 h, i.e., about 9 weeks of full-time work for one expert.
>
> Considering that the main experiment (excluding the RAG settings) already reports 24 input configurations, replacing the automated pipeline with human evaluation for all of them would amount to several thousand person-hours, which is practically infeasible.
>
> **On Suggestion on Score Calibration**
>
> In line with your suggestion, we constructed and experimented with a simpler “binary success” metric (whether the model identifies the correct error type and hits at least one piece of correct evidence):
>
> | Models               | Score(Binary) | RQD    | DI    | SG    | MO    | DHP    | CF    | IC    | RCA    | LE    |
> |--------------------------------|----------------|------|------|------|------|------|------|------|------|------|
> | Gemini 2.5 Pro_Image           | 21.3           | 18.0 | 14.0 | 48.0 | 16.0 | 29.0 | 13.0 | 18.0 | 20.6 | 14.0 |
> | GPT-5_Image                    | 25.0           | 14.0 | 12.0 | 43.0 | 19.5 | 28.0 | 20.0 | 32.0 | 28.8 | 19.0 |
> | Grok 4_Image                   | 4.9            | 0.0  | 2.0  | 23.0 | 3.5  | 8.0  | 1.0  | 2.0  | 3.6  | 1.0  |
> | Doubao-Seed-1.6_Image          | 13.9           | 4.5  | 4.7  | 48.0 | 11.5 | 17.0 | 7.5  | 23.3 | 8.8  | 5.0  |
> | Doubao-Seed-1.6-thinking_Image | 15.4           | 5.5  | 4.7  | 44.5 | 14.0 | 17.0 | 15.5 | 17.3 | 12.4 | 6.0  |
> | Gemini 2.5 Pro_Text            | 41.3           | 28.5 | 38.7 | 58.0 | 36.0 | 62.0 | 29.0 | 37.3 | 49.0 | 19.0 |
> | GPT-5_Text                     | 36.6           | 28.5 | 28.7 | 53.0 | 39.5 | 41.0 | 13.5 | 41.3 | 44.8 | 20.0 |
> | Claude Sonnet 4_Text           | 11.1           | 10.5 | 2.7  | 29.5 | 9.5  | 15.0 | 6.0  | 16.7 | 7.6  | 7.0  |
> | Grok 4_Text                    | 26.3           | 11.0 | 9.3  | 44.5 | 17.0 | 40.0 | 16.5 | 27.3 | 37.6 | 13.0 |
> | Doubao-Seed-1.6_Text           | 19.9           | 9.5  | 9.3  | 46.5 | 19.0 | 26.0 | 11.5 | 24.0 | 19.6 | 11.0 |
> | Doubao-Seed-1.6-thinking_Text  | 23.4           | 13.0 | 12.7 | 52.0 | 21.0 | 26.0 | 15.5 | 26.7 | 23.8 | 15.0 |
>
> The results show that this metric indeed improves the separation between models and can, to some extent, distinguish models that otherwise obtain uniformly low scores. At the same time, we believe under the present scan-oriented setting, models struggle simultaneously with error localization, evidence coverage, and reasoning completeness; and simply binary metrics may collapse these fundamentally different capabilities and weakens our sensitivity to improvements at the process level.
>
> We appereciate it for your new perspective. In future work, once models have largely mastered some core aspects of the scan-oriented paradigm (e.g., reliably detecting the main errors), such a binary metric will become a valuable complementary dimension of evaluation.
>
> ***
> At last, we sincerely appreciate your valuable feedback. If our responses address your concerns, we would be deeply grateful if you could kindly consider raising the score further to support our work. Thank you very much!

---

> ### Author Response · Authors · 2025-11-26
> **Follow-Up on Our Rebuttal Responses**
>
> Dear Reviewer,
>
> Thank you again for your careful review and insightful comments on our paper. We have now provided detailed responses and additional analyses in the rebuttal to address your concerns. If you have a moment to look over our replies and let us know whether there are any remaining points that would benefit from further clarification, we would be very grateful.
>
> Wishing you a joyful Thanksgiving! :-)
>
> Best regards,
> The Authors

---

### Official Review · Reviewer_78Mn · 2025-10-31

**Soundness:** 3
**Presentation:** 3
**Contribution:** 3
**Rating:** 6
**Confidence:** 3

**Summary:**

This paper introduces ScholScan, a benchmark for scholarly reasoning that contrasts with existing search-oriented evaluations. ScholScan uses a scan-oriented question design: models are required to read entire papers and derive all necessary concepts and inferences solely from the provided documents, rather than answering pre-specified queries grounded in retrieval. The benchmark covers 13 scientific domains and 9 error types, enabling a comprehensive assessment of error detection. This paper evaluates multiple models and further highlights the paradigm difference by testing the RAG method, finding that simple retrieval offers no improvement on scan-oriented tasks. These results suggest that current MLLMs still struggle with addressing such scan-oriented tasks

**Strengths:**

**Introduction of a new paradigm:** This paper proposes a novel scan-oriented paradigm, which differs fundamentally from traditional search-oriented settings with pre-specified queries. In this setup, models must autonomously and comprehensively understand the entire document and identify the error by themselves.

**Comprehensive evaluation and analysis:** This paper conducts large-scale evaluations covering both instruction-following and thinking-style models, providing detailed statistical analyses of errors. It further reveals that model performance deteriorates as the required number of documents and reasoning steps increases, offering valuable diagnostic insights.

**Well-structured evaluation metric:** The proposed metric jointly measures error detection, evidence accuracy, and reasoning quality. It rewards correct identification of target errors, penalizes noisy or irrelevant evidence, and assesses reasoning completeness through prefix matching.

**Weaknesses:**

**Synthetic task generation and authenticity concerns:**
All benchmark tasks are generated by LLMs rather than collected from real-world review scenarios, which raises questions about the authenticity of the task distribution. Although the authors include human verification, the underlying errors are created by modifying accepted papers via LLMs. As a result, the error types and frequency distributions may still deviate from those naturally occurring in real research writing. Moreover, since the tasks are generated using Gemini-2.5-Pro, this may partially explain its superior performance on the benchmark.

**Potential data contamination and long-term validity:**
There is also a risk of benchmark contamination as newer models may already include papers from sources such as ICLR 2024/2025 or Nature Communications in their pretraining corpora. Over time, this contamination could become increasingly severe, diminishing the benchmark’s long-term validity. In particular, for tasks generated from the "sampling" method based on the public reviews, future models may memorize or overfit to these examples, reducing the benchmark’s long-term validity.

**Questions:**

Have you analyzed why model performance remains low across all error categories, even when oracle evidence is provided? Is this primarily due to limitations in the models’ reasoning capability, or are there other contributing factors?

Additionally, have you evaluated human performance under the same oracle condition? From the current results, it seems that even with oracle evidence, the benchmark is still extremely challenging.

---

> ### Author Response · Authors · 2025-11-21
> **Rebuttal to Reviewer 78Mn's Comments (1)**
>
> Below we provide our detailed response to your question; we would be grateful if you could let us know whether this clarifies your concerns.
>
> **On Synthetic Task Generation and Authenticity Concerns**
>
> We would like to clarify that the benchmark is not “entirely synthetic data generated by Gemini-2.5 Pro.” As described in the paper, the dataset has two sources: Sampling and Generation.
>
> For the Sampling part, the construction procedure is:
>
> **Step1.** Retrieve real peer-review comments from OpenReview;
>
> **Step2.** Use Gemini 2.5 Pro for coarse filtering and to draft initial question candidates;
>
> **Step3.** Have human experts substantially rewrite and verify the questions.
>
> Thus, items in the Sampling subset are essentially rooted in human judgments. Gemini only plays the role of assisting with initial extraction and drafting. These review comments and question candidates are rigorously checked and rewritten by human experts during annotation, so it would be inappropriate to regard them as data “automatically generated by an LLM.”
>
> For the Generation part, the construction procedure is:
>
> **Step1.** Use Gemini 2.5 Pro to read the paper and propose candidate edits and questions;
>
> **Step2.** Have human experts integrate, add to, or discard these candidate edits, and perform evidence annotation, question rewriting, and related steps.
>
> The items in the Generation subset do indeed originate from model-generated candidates. However, they are extensively revised by human experts during the annotation process, and as a result, their final form differs substantially from the original distribution of Gemini’s raw generations.
>
> **On Potential Data Contamination and Long-term Validity**
>
> We agree that when a benchmark partially draws on high-quality, publicly available ICLR accepted papers, it is almost inevitable that future models will be exposed to some of this content during pre-training or post-training. This is a shared challenge for all open benchmarks constructed from public corpora.
>
> To directly examine this issue, we re-aggregated all proprietary model results by splitting them into the Sampling and Generation subsets, with the following outcome:
>
> **Image input**
>
> | Model              | Generation | Sampling |
> |--------------------|------------|----------|
> | Gemini 2.5 Pro     |      18.16      |     12.48     |
> | GPT-5              |      24.34      |     13.01     |
> | Grok 4             |      4.20      |     3.71     |
> | Doubao-1.6         |     11.52       |     8.06     |
> | Doubao-1.6-Thinking|      12.83      |     7.07     |
>
> **Text input**
>
> | Model              | Generation | Sampling |
> |--------------------|------------|----------|
> | Gemini 2.5 Pro     |      42.84      |     15.30     |
> | GPT-5              |      34.31      |     8.47     |
> | Grok 4             |     28.63       |     11.53     |
> | Doubao-1.6         |      19.18      |     7.68     |
> | Doubao-1.6-Thinking|      21.57      |     7.81     |
>
> Models actually perform worse on the Sampling subset than on the Generation subset, even though the former is built from errors that have already been publicly discussed in practice. This indicates that merely “having seen the paper or its reviews” does not provide a clear advantage (even for ICLR 2025 papers, the papers and reviews have been publicly available since last year, i.e., more than half a year before the release of Gemini 2.5 Pro). Our rewriting of the examples, together with the process-aware annotation, substantially reshapes the tasks, keeping them challenging even in the presence of potential exposure.
>
> In summary, our dataset construction pipeline involves extensive rewriting and refinement of public reviews, so that even if future models include these papers or reviews in their training data, overfitting to ScholScan is unlikely. Moreover, our work provides a reusable construction recipe, making ScholScan amenable to future updates.
>
> **Analysis of the Oracle Setting and Remaining Performance Gap**
>
> First, we would like to clarify what “oracle evidence” means in our setting: we provide models with the entire pages that contain the gold evidence, rather than pre-cropped evidence sentences or paragraphs. The task therefore remains fundamentally scan-oriented, rather than a setting where the correct evidence sentence is simply handed to the model.
>
> We believe that under this page-level oracle, the difficulty of global retrieval and coarse document-level localization is partially alleviated, but not fundamentally resolved. Put differently, even in the oracle condition, the requirements and core challenges of the scan-oriented task are still very much present.

---

> ### Author Response · Authors · 2025-11-21
> **Rebuttal to Reviewer 78Mn's Comments (2)**
>
> **Human Performance Under the Oracle Setting**
>
> We conducted a human baseline experiment that is strictly aligned with the oracle setting in this paper. Specifically, we randomly sampled 20 questions that match the invited experts’ research areas and that did not appear in either the data annotation process or any prior human study (corresponding to 20 distinct papers), allocating 30 minutes per question. We then divided experts into two groups: in the baseline condition, experts were given the full paper; in the oracle condition, experts were given the oracle evidence pages for these papers. We used different experts for the baseline and oracle groups to avoid confounds such as memory effects, repeated reading, or familiarity with the same paper; the results are summarized as follows.
>
> | Group | Avg. time used | Avg. score | % questions unanswered (timeout) |
> |----------------------|----------------|-----------:|----------------------------------:|
> |  Baseline           | 27.0 minutes   | 62.04      | 20%                               |
> |  Oracle          | 22.2 minutes   | 79.52      | 5%                               |
>
> The results show that under the oracle condition, human experts achieve somewhat higher scores than in the baseline full-paper setting, and their performance remains substantially higher than that of the strongest current MLLMs. This further indicates that oracle evidence only partially alleviates the retrieval and localization burden and does not remove the core reasoning difficulty of the tasks; under this setting, the scan-oriented evaluation continues to pose a substantial challenge for current MLLMs.
>
> ***
> At last, we sincerely appreciate your valuable feedback. If our responses address your concerns, we would be deeply grateful if you could kindly consider raising the score further to support our work. Thank you very much!

---

> > ### Comment · Reviewer_78Mn · 2025-11-25
> >
> > Thanks for the detailed response and additional experiments. I will maintain my positive score.

---

> > > ### Author Response · Authors · 2025-11-25
> > >
> > > Thanks for your reply! If there are any remaining questions or points requiring further clarification, we would be pleased to engage in further discussion.

---

### Official Review · Reviewer_HMvn · 2025-10-31

**Soundness:** 2
**Presentation:** 3
**Contribution:** 2
**Rating:** 4
**Confidence:** 4

**Summary:**

This paper proposes ScholScan, a new benchmark designed to evaluate multimodal large language models (MLLMs) on “scan-oriented” scholarly reasoning—i.e., the ability to read and cross-check entire academic papers like human researchers to identify internal consistency issues. Unlike conventional “search-oriented” paradigms that rely on pre-specified questions and localized retrieval, ScholScan presents target-absent queries that require holistic document understanding. The benchmark includes 1,800 expert-annotated questions across 9 scientific error types, drawn from 715 papers in 13 natural science domains, with detailed evidence localization and reasoning traces.

**Strengths:**

1. The paper clearly articulates a meaningful distinction between search-oriented and scan-oriented reasoning paradigms, highlighting a critical gap in current MLLM capabilities for full-document scientific verification.

2. The dataset construction process is rigorous: 10 domain experts performed dual reviews with third-party adjudication. Of 3,500 initial candidates, 1,700 were discarded and 1,541 revised—demonstrating strong quality control.

3. The evaluation is comprehensive: 15 models under 24 input configurations and 8 RAG frameworks are tested. Results consistently show poor performance across the board, and notably, RAG offers no significant gains—underscoring the challenge of scan-oriented tasks.

**Weaknesses:**

1. The definition of “scan-oriented” remains somewhat vague. Figure 1 contrasts target-prespecified vs. target-absent questions, which is clearer than the term “scan-oriented” itself. It would help to explicitly state that “scan-oriented” means processing the entire document (text or image) without relying on retrieved chunks—as opposed to RAG’s fragment-based approach.

2. While the 9 error types span the scientific workflow (Figure 3), it is not convincingly demonstrated that all require full-document understanding. Many instances are derived from OpenReview comments, which may reflect localized, detail-level critiques rather than issues demanding cross-section or cross-paper reasoning.

3. Model performance is universally low (e.g., even under Oracle RAG, average score is only 24.5/100). This raises the question: how would human experts perform on these tasks? Without a human baseline, it’s unclear whether the low scores reflect model limitations or simply extreme task difficulty.

4. Several figures and tables lack sufficient clarity:
   - Figures 4 and 5 omit axis labels.
   - Figure 5’s “evidence locations” is not well-defined—how are these locations counted or validated?
   - Table 2 reports RAG results but does not describe the detailed setting or related references of these RAG methods, making it hard to understand.

**Questions:**

1. Scan-oriented processing requires ingesting the full paper, which can be very long. Given current context-length and attention limitations, could this approach become impractical for some real-world use? How does performance scale with document length?

2. Section 4.4 notes that stronger models have fewer zero-score cases but more hallucinations. Yet Figure 5 shows Gemini suffering from high omission and hallucination rates—seemingly at odds with its relatively higher scores in Table 1. Can the authors reconcile this apparent contradiction?

3. Figure 6 suggests performance degrades with more reasoning steps or evidence locations. However, this may simply reflect inherently harder tasks rather than a causal effect of reasoning length. Could the authors provide an analysis of task difficulty (e.g., via human expert ratings) to disentangle complexity from model limitations?

4. The Oracle RAG condition does yield significant gains, implying that high-quality retrieval is helpful. Doesn’t this suggest the core issue is not the RAG paradigm itself, but the inadequacy of current retrievers? Might future advances in retrieval close part of the gap?

---

> ### Author Response · Authors · 2025-11-19
> **Rebuttal to Reviewer HMvn's Comments (1)**
>
> Below we provide our detailed response to your question; we would be grateful if you could let us know whether this clarifies your concerns.
>
> **Clarifying the meaning of “scan-oriented” and its contrast to RAG-style retrieval**
>
> In the current manuscript, we introduce this paradigm in the introduction and in Figure 1 by describing tasks where models must actively construct a document-level evidence view, and perform evidence-based reasoning without prespecified targets or hints. In addition, Figure 1 explicitly includes a “retrieval” module under both paradigms, which is intended to denote a RAG-style, fragment-based retrieval component (i.e., first chunking the document and then reasoning over the retrieved fragments).
>
> We are already revising the text to state it more explicitly and will upload an updated version during the rebuttal period.
>
> **Justification for full-document understanding in all 9 error types and clarification on use of OpenReview comments**
>
> All 9 error types in ScholScan are defined as requiring document‑level understanding, even if some of them may appear to be “local details” on the surface. This is particularly true for CF (Computation & Formulae) and LE (Language & Expression), which are also the two categories most likely to be misunderstood as not involving broader context.
>
> For CF (Computation & Formulae), identifying whether a computation is incorrect cannot be done by looking only at the formula itself. The parameters and variables involved are often defined across multiple sections—such as data preprocessing, experimental setup, and ablation studies. Determining the correct value usually requires integrating information from earlier definitions, dataset scale, experimental conditions, and previously reported results.
>
> For LE (Language & Expression), the situation is more evident: issues such as spelling, phrasing, or terminology inconsistencies can potentially occur anywhere in the text. Verifying such issues often requires scanning the entire document, as relevant cues, such as canonical terminology, consistent phrasing, or proper usage, are distributed across many sections rather than confined to a specific passage.
>
> We respectfully invite you to revisit the examples in Figure 2, which aim to illustrate the document-level nature of each error type. If any specific category appears insufficiently justified, we would greatly appreciate the chance to clarify or strengthen its explanation.
>
> Regarding your comment on items constructed from OpenReview feedback, we would also like to clarify the following: review comments serve more as starting points for locating potential issues, not as the questions themselves. In our sampling and annotation pipeline, annotators must return to the original paper to realign and validate any candidate issue. The annotation guidelines explicitly require filtering out comments that are purely local, purely stylistic, or subjective in nature, keeping only those that can be verified or falsified based on the scientific content of the paper. After this stage, we further conduct extensive human screening and rewriting, discarding a substantial number of candidates and substantially modifying those retained, ensuring that the resulting items ultimately require reasoning across multiple paragraphs, sections, or even across different papers.

---

> ### Author Response · Authors · 2025-11-19
> **Rebuttal to Reviewer HMvn's Comments (2)**
>
> **Interpreting Low Model Performance and Human Baseline Evaluation**
>
> We first want to clarify that, in real scientific writing and peer review, the kinds of issues covered by ScholScan are exactly what authors, reviewers, and editors routinely have to check. For domain experts with the relevant background, if you give them sufficient time to read, go back and forth across sections, compare tables and formulas, and verify citations against the original sources, these errors can in principle be systematically identified and corrected. Only under this level of scrutiny can a paper reach the quality threshold required for publication.
>
> The real bottleneck is cost, not solvability. In practice, the publication process can approach a very high human baseline only because it relies on a resource-intensive pipeline: a single paper typically goes through multiple rounds of review and revision involving several reviewers and an area chair/editor, and some journals add an extra professional editing or proofreading pass before final acceptance. Each round means additional hours of expert labour. This multi-stage, multi-person redundancy is a substantial consumption of time and expertise. From this perspective, it is precisely because humans can reach very high accuracy when enough effort is invested that the community is willing to pay this cost to maintain a certain quality floor.
>
> Against this backdrop, the goal of ScholScan is not to design a puzzle that is “almost impossible even for humans”, but to approximate a realistic high-standard reading scenario and measure how far current MLLMs are from a human-like ability to read a whole paper and detect such errors.
>
> To directly address your concern about task difficulty, we also ran a supplemental human-baseline experiment. We recruited the domain experts who participated in dataset construction as subjects, and for each expert we assigned 20 questions from their area of expertise that they had not personally annotated, to avoid memory bias. We used two time limits, 15 minutes and 30 minutes, and under each condition we asked experts to read the full paper and independently answer the corresponding questions. We then applied exactly the same automatic scoring pipeline as for the models, using GPT-4.1 to grade human answers, bringing into consistency with our paper.
>
> | Time limit per paper | Avg. time used | Avg. score | % questions unanswered (timeout) |
> |----------------------|----------------|-----------:|----------------------------------:|
> | 30 minutes           | 27.0 minutes   | 62.04      | 20%                               |
> | 15 minutes           | 13.6 minutes   | 13.66      | 80%                               |
>
> The results show that, on ScholScan’s scan-oriented tasks, domain experts achieve substantially higher scores than current MLLMs when given enough time to read and think. This directly supports our central claim: given sufficient time and resources, humans can reach a high baseline on this benchmark; the fundamental limitation lies in time and human labour cost, rather than in any cognitive impossibility of the tasks themselves.
>
> **Clarifying Figure and Table Presentation, Evidence Counting, and RAG Settings**
>
> Figure 5 involves two models: Gemini 2.5 Pro and GPT-5. Since both models have built-in reasoning abilities, we first collect their full chains of thought and then prompt GPT-4.1 to annotate and count the evidence locations actually used in their reasoning, following exactly the same principles as in the automatic grading stage.
>
> All RAG methods reported in Table 2, together with their references, are already listed in Section 4.5, including BM25, BGE-M3, Contriever-msmarco, NV-Embed-v2, ColPali-v1.3, ColQwen2.5, VisRAG, and VRAG-RL. The hyperparameters and implementation details for these RAG settings are as follows:
>
> • In all RAG configurations, we first score all pages and select the top-10 most similar pages per example. For image RAG, we pass these 10 page screenshots; for text RAG, we pass the OCR text of these 10 pages, in ascending page order.
>
> • For NV-Embed-v2, we set the maximum encoding length to max_length = 3000 tokens.
>
> • For ColPali and ColQwen2.5, we use IMAGE_BATCH = 8 and TEXT_BATCH = 8.
>
> • In VRAG-RL, we use ColQwen2.5 as the multimodal retriever, and we do not impose a fixed upper bound on the number of reasoning iterations.
>
> • All experiments are run on a single server with 8×NVIDIA A40 GPUs.
>
> In line with your suggestion, we will further improve the labeling and explanations of the relevant figures and tables, and we will add the full experimental setup to the appendix.

---

> > ### Comment · Reviewer_HMvn · 2025-11-25
> >
> > I appreciate the authors' extra effort and responses. However, in the supplementary experiments, I found that human experts still only scored 62.04% on 80% of the questions given a half-hour time limit. This supplementary experimental result still does not fully address my concerns about whether these questions can all be answered effectively (perfectly corresponding to standard question answers), although the results from human experts are significantly better than the model's performance. As a benchmark, the validity of the questions is crucial.

---

> > > ### Author Response · Authors · 2025-11-26
> > >
> > > Thank you very much for your response. We would like to offer the following clarifications, which we hope will be helpful:
> > >
> > > > As a benchmark, the validity of the questions is crucial.
> > >
> > > The reported score of 62.04 is the overall average across all questions, including those left unanswered when experts ran out of time; these unanswered questions are counted as 0. This means that, conditioning only on questions that were actually answered, human experts achieve an average score of around 78/100, which is in fact quite a high level under our stringent evaluation protocol.
> > >
> > > Second, our original 15-minute and 30-minute settings were designed precisely to reveal a clear time–accuracy gradient. We have now run an additional experiment in which experts are given 60 minutes per paper; the accuracy is reported as follows:
> > > | Time limit per paper (min) | Avg. time used (min) | Avg. score (Scaled by 100) | % questions unanswered (timeout) |
> > > |-|-|-:|-:|
> > > | 60 | 36.2 | 90.4| 0|
> > > | 30 | 27.0 | 62.0| 20|
> > > | 15 | 13.6 | 13.7| 80 |
> > >
> > > In the 60-minute condition, we observe that: (i) the average time per question is well below the allotted limit; (ii) all questions can be answered; and (iii) the average score reaches about 90/100.
> > >
> > > > My previous concern was that the problems requiring long inference chains might be more difficult, and even human experts wouldn't exhibit similar performance degradation.
> > >
> > > We compute a breakdown of the 60-minute results on these 20 questions by golden reasoning chain length; the results are as follows:
> > >
> > > | Gold reasoning steps | Amount | Avg. score (Scaled by 100) |
> > > |-|-:|-:|
> > > | 1| 5| 90.0|
> > > | 2| 9| 93.2|
> > > | 3| 4| 87.6|
> > > | 5| 2| 84.2|
> > >
> > > With sufficient time, human experts exhibit only a mild and cognitively reasonable decrease in performance on questions that require longer reasoning chains. We hope this helps address your concerns about the validity of the benchmark questions.
> > >
> > > > Regarding the RAG configuration, I think it's a rather naive approach, especially since the authors emphasize that many questions require answers across many pages.
> > >
> > > We do not claim that our current RAG configuration is the optimal way to solve ScholScan. We choose to operate at the page level in order to stay consistent with recent work (e.g., multimodal RAG systems such as ColPali, ViDoRAG, M3DocRAG, SimpleDoc, and multimodal document benchmarks such as MMDocIR), which almost universally treat pages as the basic retrieval units.
> > >
> > > Accordingly, our goal in adopting this standard configuration is to answer a diagnostic question: under mainstream RAG input settings, what do current models actually achieve on scan-oriented tasks, and where are their main bottlenecks?
> > >
> > > As for the choice of k = 10 most similar pages, this is based on empirical statistics and pilot experiments: (i) even for the questions in ScholScan whose oracle evidence spans the largest number of pages, k = 10 is sufficient to cover all relevant pages in theory, making these questions answerable in principle; and (ii) this choice helps control the input length and computational load for the models.
> > >
> > > For graph-based and other knowledge-enhanced approaches, please refer to our response to reviewer a2Tc:
> > >
> > > “We agree that graph-based representations of a paper’s claims and evidence may be promising for some error types. However, we see at least two limitations: (i) research papers are highly multimodal and layout-heavy documents, and current multimodal knowledge-graph techniques still have limited coverage and fidelity; (ii) compressing an entire paper into a graph often discards a large amount of “semantically weak but reasoning-critical” detail, such as concrete experimental numbers, confidence intervals, and hyperparameter settings. For this reason, we view graph representations as better suited as auxiliary modules for particular error types, rather than as a single unified architecture that could robustly handle all nine error categories.”
> > >
> > > We would be grateful for your further response. Wishing you a great Thanksgiving. :-)
> > >
> > > [1] Faysse et al. ColPali: Efficient Document Retrieval with Vision Language Models.
> > >     In Proceedings of the International Conference on Learning Representations (ICLR 2025), 2025.
> > >
> > > [2] Wang et al. ViDoRAG: Visual Document Retrieval-Augmented Generation via Dynamic Iterative Reasoning Agents.
> > >     In Proceedings of EMNLP 2025, 2025. (arXiv:2502.18017)
> > >
> > > [3] Cho et al. M3DocRAG: Multi-modal Retrieval is What You Need for Multi-page Multi-document Understanding.
> > >     arXiv preprint arXiv:2411.04952, 2024.
> > >
> > > [4] Jain et al. SimpleDoc: Multi-Modal Document Understanding with Dual-Cue Page Retrieval and Iterative Refinement.
> > >     arXiv preprint arXiv:2506.14035, 2025. (to appear in EMNLP 2025)
> > >
> > > [5] Dong et al. MMDocIR: Benchmarking Multimodal Retrieval for Long Documents.
> > >     In Proceedings of the 2025 Conference on Empirical Methods in Natural Language Processing (EMNLP 2025), 2025.

---

> > ### Comment · Reviewer_HMvn · 2025-11-25
> >
> > "In all RAG configurations, we first score all pages and select the top-10 most similar pages per example. For image RAG, we pass these 10 page screenshots; for text RAG, we pass the OCR text of these 10 pages, in ascending page order." Regarding the RAG configuration, I think it's a rather naive approach, especially since the authors emphasize that many questions require answers across many pages. Simply vectorizing each page, rather than pre-constructing evidence chains (such as a knowledge graph), makes it difficult to connect all possible evidence, resulting in incomplete evidence retrieval. However, I acknowledge that this is beyond the scope of this paper's research.

---

> > > ### Comment · Reviewer_HMvn · 2025-11-25
> > >
> > > Based on the authors' supplementary experiments and detailed responses, I'm willing to raise my rating.

---

> ### Author Response · Authors · 2025-11-19
> **Rebuttal to Reviewer HMvn's Comments (3)**
>
> **Feasibility of Full-Paper Input and Performance Scaling with Document Length**
>
> To more concretely address your question, we additionally analyze GPT-5 and Gemini 2.5 Pro under both text-input and image-input configurations, and examine their performance across different page-count ranges. The results are summarized as follows (All scores scaled by 100):
>
> | Pages  | Gemini-Text | Gemini-Image | GPT-5-Text | GPT-5-Image |
> |--------|------------:|-------------:|-----------:|------------:|
> | 1–10   |       18.2 |        38.0 |      24.3 |       30.8 |
> | 11–20  |       13.4 |        28.4 |      17.6 |       20.7 |
> | 21–30  |       13.3 |        17.4 |      12.3 |       11.2 |
> | 31–40  |       21.5 |        46.0 |      29.5 |       34.8 |
> | > 40   |       25.6 |        45.1 |      28.0 |       36.1 |
>
> (Here, Image and Text denote the input configurations.)
>
> We find that the differences across length buckets are relatively mild; within the length range covered by ScholScan, model performance is not predominantly driven by context-length limits. Our interpretation is that, in the ScholScan setting, the main challenge lies in locating and integrating evidence across the full paper. In other words, even for longer papers, the primary weakness of current MLLMs is their global reading and reasoning strategy, rather than a purely hard constraint on sequence length. At the same time, even in cases involving cross-paper alignment errors (RCA, where two papers must be fed to the model), the total length of typical scientific articles rarely reaches an extreme scale that is fundamentally beyond what current MLLMs can process.
>
> Regarding the necessity and practical usability of “full-paper input”, we believe it is important to return to the core capability that ScholScan aims to capture: we focus on a type of behaviour that is hard to reliably evaluate under strong summarization—maintaining a global evidence view at the scale of an entire paper, so as to detect subtle yet critical anomalies (e.g., cross-paragraph logical inconsistencies, nuanced misstatements of prior work, or small but meaningful discrepancies in dosage or numerical values between two figures). These signals are exactly the kind of information that is most likely to be lost when the document is heavily summarized or aggressively chunked.
>
> Taken together, we therefore view full-paper input, within the length range of ScholScan, as both feasible and scientifically appropriate for this evaluation setting. The key difficulty lies in global evidence aggregation and reasoning, which is precisely the aspect of MLLM behaviour that we hope ScholScan will encourage the community to improve.
>
> **On the Apparent Contradiction Between Gemini’s Hallucination Rates and Its Higher Average Scores**
>
> In Section 4.4, our statement that “stronger models have fewer zero-score cases but are more likely to exhibit over-confident hallucinations” is based on an analysis within each individual model, rather than a cross-model comparison. Under both input configurations, Gemini 2.5 Pro follows this pattern. This is not in conflict with the relatively higher scores of Gemini 2.5 Pro in Table 1, because we observe that it achieves higher scores on many of the remaining questions.
>
> In other words, the higher average score of Gemini 2.5 Pro does not mean that hallucinations or omissions are rare. Instead, it means that on some questions it (i) successfully detects the target edit, (ii) aligns well with the gold evidence, and (iii) produces a reasoning chain that largely overlaps with the reference answer, while still exhibiting hallucinations or omissions on more questions than other models.
>
> **Clarifying the Interpretation of Figure 6 and the Role of Task Complexity**
>
> Our intention is not to claim that longer reasoning chains causally lead to lower performance. Rather, Figure 6 is meant to descriptively show that current models perform markedly worse on instances that objectively require more evidence and longer gold reasoning chains. In other words, the role of Figure 6 is to characterize how models struggle increasingly on higher process-complexity scan-oriented tasks, rather than to make a causal statement about reasoning length itself.
>
> In the second and fourth subfigures of Figure 6, we already report the average reasoning-process score S_reasoning and evidence-location score S_location under different gold evidence counts and reasoning-chain lengths. It is clear from these plots that both scores exhibit a consistent downward trend as the gold reasoning chains become longer. This provides direct, process-level support for our conclusion that long causal chains pose a particular challenge for current models.

---

> > ### Comment · Reviewer_HMvn · 2025-11-25
> >
> > I understand Figure 6's description of the current model performing significantly worse on instances that objectively require more evidence and longer golden inference chains. My previous concern was that the problems requiring long inference chains might be more difficult, and even human experts wouldn't exhibit similar performance degradation.

---

> ### Author Response · Authors · 2025-11-19
> **Rebuttal to Reviewer HMvn's Comments (4)**
>
> **On Whether RAG Limitations Stem from Retriever Quality or the Paradigm Itself**
>
> We sincerely appreciate your insightful question and would be very glad to further discuss our perspective on why current RAG paradigms struggle with scan-oriented tasks. If the failure of a given RAG paradigm were primarily due to inadequate retrievers, then as retrieval models improve over time we would expect to observe at least one of the following:
>
> **(i)** a relatively stable upward trend in retrieval quality and downstream answer scores as newer retrievers are adopted, **or**
>
> **(ii)** clear case-study evidence that stronger retrievers systematically surface better evidence and thereby yield higher scores.
>
> However, in our experiments we evaluate a range of retrievers whose publication dates span from classical BM25 (1994) to the latest models released in 2025, and across this time span the performance on ScholScan does not exhibit any substantial or monotonic improvement. This suggests that, for scan-oriented tasks that require full-document scanning and global consistency checking, the dominant limitation lies in the mainstream chunk-based RAG paradigm itself, rather than being solely a consequence of current retrievers being insufficiently strong.
>
> Additionally, Table 3 reports the retrieval performance metrics of the RAG methods evaluated. These results also fail to show a consistent improvement across retrievers, which further suggests that the limitations are rooted in the chunk-based RAG framework itself, rather than in the current retrievers alone.
>
> ***
> At last, we sincerely appreciate your valuable feedback. If our responses address your concerns, we would be deeply grateful if you could kindly consider raising the score further to support our work. Thank you very much!

---

### Official Review · Reviewer_6ouw · 2025-11-01

**Soundness:** 3
**Presentation:** 3
**Contribution:** 4
**Rating:** 8
**Confidence:** 4

**Summary:**

The paper proposes ScholScan, a new benchmark meant to evaluate multimodal large language models (MLLMs) on “scan-oriented” scholarly paper reasoning rather than traditional “search-oriented” QA. This paper introduces a task setting where the model is given one or more full academic papers and a target-absent instruction (e.g., “Assess the Methods section for Measurement & Operationalization issues”) and curated 1,800 questions with 9 families of scientific errors. The authors found that existing models perform badly in this new task, and this can not be simply addressed by a RAG pipeline.

**Strengths:**

1. **Task**: The paper draws a sharp conceptual line between search-oriented QA (“find X in this paper”) and scan-oriented critique (“is anything off about this study?”). This framing is intuitive, well motivated, and reflects a real gap between “LLM as literature assistant” and “LLM as junior reviewer.” This is an important task with pressing needs.

2. **Dataset scale, diversity, and annotation rigor**: the benchmark curation process covers good scale (715 papers across 13 scientific domains (including physics, chemistry, computer science, biology, etc.) and 1,800 instances spanning nine error families is impressively broad and goes beyond narrowly technical CS-only corpora. There is rigorous human review of the generated data instances.

3. **Strong analysis and observations**: OCR text often outperforms raw page images overall (likely because long-context language modeling is currently better than long-context vision-language modeling). Longer reasoning chains actually correlate with worse scores. They report that classical retrieval does not solve the “scan-oriented” problem because the hardest part is deciding what to look for in the first place. These are very valuable insights immediately transferrable to the AI-for-science community.

**Weaknesses:**

1. **Synthetic edit to accepted papers**: A large fraction of benchmark items are created by prompting Gemini 2.5 Pro to insert coordinated edits into accepted/high-quality papers, and then asking models to catch those edits. The quality of this generated split of dataset is bounded by the capacity of Gemini 2.5 Pro. As the paper later finds, even best models so far struggle with the "scan-oriented" problems. This would hurt the quality of the benchmark.

2. **Evaluation metric design**: The final score S(m) is an involved product of existence, evidence localization Dice (with squared penalties for over-reporting), reasoning prefix match, and a nonlinear hallucination penalty. There are many tunable constants (e.g. the 0.8 subtraction, exponent 1.5 in the hallucination penalty). It’s not obvious these specific choices are robust or interpretable across domains.

3. **Leakage concern**: The benchmark is partially consisted of high-quality ICLR-accepted papers that future models will train on. Since the benchmark curation process involves extensive human annotations, data leakage problem becomes a major problem.

**Questions:**

1. Can you break down, as exact percentages (or counts): (1) generated-edited accepted papers (Gemini edits), (2) sampled-from-rejected-papers (ICLR reviews), (3) cross-paper citation consistency cases?

2. You mention that human evaluation “confirms high agreement” between the GPT-4.1 extraction/metric pipeline and expert annotations. Can you report some quantitative results?

---

> ### Author Response · Authors · 2025-11-22
> **Rebuttal to Reviewer 6ouw's Comments (1)**
>
> Below we provide our detailed response to your question; we would be grateful if you could let us know whether this clarifies your concerns.
>
> **On the Quality of Gemini-Generated Synthetic Edits**
>
> In our workflow, modifying a local piece of content is much easier than finding that issue again under the scan-oriented setting. Gemini 2.5 Pro is only used to propose candidate edits. Discovering these edits in ScholScan instead requires the evaluated model to read the entire paper, build a document-level evidence view, and perform consistency checking across multiple sections, figures/tables, and even cited references.
>
> Second, within our pipeline Gemini mainly plays the role of suggesting candidates and directions, rather than directly determining the final benchmark items. For the synthetic-edit examples, we first use Gemini to over-generate a large pool of candidates, and then apply strict human filtering and rewriting: a substantial portion of candidates is discarded outright, and the retained ones are manually checked and revised against the full PDF. In this sense, **the final quality of the dataset is in fact elevated by human editing standards, rather than being bounded by Gemini’s raw generation capacity.**
>
> **Systematic Explanation of Our Evaluation Metric Design**
>
> We organize our explanation of the evaluation metric into three parts:
>
> **A. Qualitative analysis of the non-linear scoring behavior**
>
> 1. On S_reasoning: we use the square of the proportion of correctly covered reasoning prefix to define S_reasoning, in order to more strongly reward answers that reproduce the full reasoning chain.
>
> 2. On S_location: we use a Dice-style function to measure the overlap between predicted and gold evidence, and add an explicit over-reporting penalty, to reflect the need in real reviewing scenarios to precisely point to the true inconsistencies.
>
> 3. On P_unrelated_err: since the number of unrelated errors is in principle unbounded, and even a few such errors can quickly degrade answer quality, we introduce an exponential penalty and set a tolerance up to two unrelated errors in the multiplicative factor.
>
> 4. On the overall outcome score: we take the geometric mean of S_reasoning and S_location to obtain the final score, so that if either component is poor, the overall score will be significantly reduced.
>
> **B. Suitability of the current hyperparameters for ScholScan**
>
> We now explain why the four hyperparameters 0.8,0.9,1.6,0.5 are chosen in the paper:
> 1. For S_location: we provide a table below showing several typical cases when the gold evidence set contains 6 items:
>
> | Gold Evidence Count | Hit Count | Extra Count | Predicted Evidence Count | S_location |
> |-------------------|----------|------------|------------------------|-----------:|
> | 6                 | 6        | 0          | 6                      |     1.00   |
> | 6                 | 4        | 0          | 4                      |     0.80   |
> | 6                 | 4        | 2          | 6                      |     0.58   |
> | 6                 | 4        | 4          | 8                      |     0.37   |
>
> The results show that our metric can effectively penalize unrelated evidence: when the model only correctly “finds” 4 gold evidence items, the evidence-location score has already dropped below the passing threshold.
>
> 2. For P_unrelated_err: under the current hyperparameter setting, we obtain the values summarized in the table below:
>
> | Unrelated Error Count (n) | P_unrelated_err |
> |-------------------------|----------------:|
> | 0                       |           1.00  |
> | 1                       |           0.90  |
> | 2                       |           0.81  |
> | 3                       |           0.44  |
> | 4                       |           0.15  |
> | 5                       |           0.04  |
> | 6                       |           0.01  |
>
> Once the number of unrelated errors reaches 3 (i.e., the answer is already proposing many spurious error candidates), the penalty becomes quite substantial.
>
> **C. Sensitivity analysis of the hyperparameters**
>
> We further conduct a sensitivity analysis of the four hyperparameters discussed in part B. In the main paper, they are set to λ=0.8, μ=0.9, γ=0.5, and q=1.6. We vary each hyperparameter individually while keeping the others fixed, and obtain the results reported in the next part of the comment.

---

> ### Author Response · Authors · 2025-11-22
> **Rebuttal to Reviewer 6ouw's Comments (2)**
>
> | Model                          | λ=0.6 | λ=0.8 | λ=1.0 | μ=0.85 | μ=0.9 | μ=0.95 | γ=0.4 | γ=0.6 | γ=0.8 | q=1.0 | q=1.5 | q=2.0 |
> |--------------------------------|-------|-------|-------|--------|-------|--------|-------|-------|-------|-------|-------|-------|
> | GPT-5_Image                    | 19.3  | 19.2  | 19.0  | 18.5   | 19.2  | 19.9   | 19.4  | 19.2  | 19.0  | 19.3  | 19.2  | 19.1  |
> | Gemini 2.5 Pro_Image           | 15.8  | 15.6  | 15.3  | 15.0   | 15.6  | 16.1   | 15.7  | 15.6  | 15.5  | 15.6  | 15.6  | 15.5  |
> | Doubao-Seed-1.6-Thinking_Image | 10.4  | 10.2  | 10.0  | 9.9    | 10.2  | 10.5   | 10.4  | 10.2  | 10.0  | 10.3  | 10.2  | 10.1  |
> | Doubao-Seed-1.6_Image          | 10.1  | 9.9   | 9.8   | 9.7    | 9.9   | 10.2   | 10.1  | 9.9   | 9.8   | 10.0  | 9.9   | 9.9   |
> | Grok 4_Image                   | 4.0   | 4.0   | 3.9   | 3.9    | 4.0   | 4.1    | 4.0   | 4.0   | 4.0   | 4.0   | 4.0   | 4.0   |
> | Gemini 2.5 Pro_Text            | 30.6  | 30.2  | 29.9  | 29.1   | 30.2  | 31.5   | 30.7  | 30.2  | 29.9  | 30.5  | 30.2  | 30.1  |
> | GPT-5_Text                     | 22.7  | 22.5  | 22.3  | 21.4   | 22.5  | 23.7   | 23.4  | 22.5  | 21.9  | 23.0  | 22.5  | 22.2  |
> | Grok 4_Text                    | 21.1  | 20.8  | 20.6  | 20.2   | 20.8  | 21.4   | 21.0  | 20.8  | 20.7  | 20.9  | 20.8  | 20.8  |
> | Doubao-Seed-1.6-Thinking_Text  | 15.6  | 15.3  | 15.0  | 14.8   | 15.3  | 15.8   | 15.6  | 15.3  | 15.1  | 15.5  | 15.3  | 15.2  |
> | Doubao-Seed-1.6_Text           | 14.1  | 13.9  | 13.7  | 13.6   | 13.9  | 14.3   | 14.2  | 13.9  | 13.7  | 14.1  | 13.9  | 13.8  |
> | Claude Sonnet 4_Text           | 6.0   | 5.9   | 5.8   | 5.6    | 5.9   | 6.1    | 6.3   | 5.9   | 5.6   | 6.1   | 5.9   | 5.7   |
>
> (Here, Image and Text denote the input configurations.)
>
> Our conclusions are:
>
> 1. Across all tested values, the relative ranking of the 11 proprietary model configurations remains unchanged. This indicates that **the evaluation metric is highly robust** at the level of comparative conclusions.
>
> 2. For each individual model, the score variation across different hyperparameter settings is very small, and much smaller than both the gap between humans and the strongest current MLLMs and the differences between models. In other words, **changing the hyperparameters within a reasonable range does not alter our core conclusions**.
>
> **On potential data leakage when using accepted papers and human annotations.**
>
> We agree that when a benchmark is partially constructed from high-quality, publicly available ICLR accepted papers, it is almost unavoidable that future models will be exposed to some of this content during pre-training or post-training. This is a challenge shared by essentially all open benchmarks built on public corpora. In the context of ScholScan, we believe it is helpful to distinguish between the generation part and the sampling part when discussing this issue with you.
>
> For the generation part, even if a model has seen the correct version of a paper during training, it has not seen the edited-with-errors version that we later construct, nor the associated explanations and labels. In fact, if a model over-relies on its memory of the correct paper, it can be more easily misled by our injected inconsistencies.
>
> For the sampling part, we do start from real human review comments. Here, we agree that if a model has seen both the original paper and its corresponding public reviews during training, there is theoretically a higher risk of leakage.
>
> To directly probe this risk, we conducted an empirical check: we compared several strong proprietary models on the two subsets, with results summarized below.
>
> | Model               | Generation-Image | Sampling-Image | Generation-Text | Sampling-Text |
> |---------------------|-----------|------------|----------|-----------|
> | Gemini 2.5 Pro      | 18.16     | 12.48      | 42.84    | 15.30     |
> | GPT-5               | 24.34     | 13.01      | 34.31    | 8.47      |
> | Grok 4              | 4.20      | 3.71       | 28.63    | 11.53     |
> | Doubao-1.6          | 11.52     | 8.06       | 19.18    | 7.68      |
> | Doubao-1.6-Thinking | 12.83     | 7.07       | 21.57    | 7.81      |
>
>
> The outcome is that **merely “having seen the paper or the review” does not yield a clear advantage** (even though the ICLR 2025 papers and reviews have been publicly available online since last year). Our rewrites of the examples and the process-based annotations substantially reshape the task, so that it remains challenging even under potential exposure.
>
> Finally, ScholScan is designed as an updatable dataset. Even if the current version is partially used for training in the future, we can construct new versions from newer papers to serve as genuinely “unseen” evaluation sets. Our methodology and construction pipeline therefore help ensure that ScholScan can retain its evaluation value as models and training corpora continue to evolve.

---

> ### Author Response · Authors · 2025-11-22
> **Rebuttal to Reviewer 6ouw's Comments (3)**
>
> **Exact Composition of ScholScan: Generation, Sampling, and Cross-Paper Cases**
>
> ScholScan contains 1800 items in total: **977** questions are built on Gemini-edited accepted/high-quality papers, **823** are sampled and revised from public ICLR reviews of rejected or critiqued papers. **200** questions correspond to our Referential & Citation Alignment cases involving cross-paper citation consistency.
>
> **Quantitative Human–GPT-4.1 Agreement Results**
>
> The detailed procedure and numerical results of this consistency check are presented in Appendix F; we briefly summarize them here.
>
> We randomly sampled 200 questions from ScholScan and asked human experts to manually analyze the corresponding model responses, extracting (i) the evidence sets, (ii) the reasoning chains, and (iii) the number of unrelated errors. Based on these human-derived extractions, we computed four metrics using the formulas in Section 4.1: the evidence localization score S_location, the reasoning-process score S_reasoning, the unrelated-error penalty P_unrelated_err, and the overall score S_total. We then compared these human-based score vectors with the corresponding vectors obtained from the GPT-4.1 extraction pipeline and calculated Spearman’s correlation coefficients. The results reported in Table 4 of Appendix F are as follows:
>
> | S_total | S_location | S_reasoning | P_unrelated_err |
> |--------|------------|-------------|------------------|
> | 0.841  | 0.806      | 0.842       | 0.954            |
>
> In summary, GPT-4.1 can extract relevant evidence and reconstruct reasoning steps with considerable accuracy, leading to reliable process-aware evaluation scores. This supports the effectiveness of our methodology that uses GPT-4.1 as an evaluator to parse and score the responses of the models under evaluation.

---

> ### Author Response · Authors · 2025-11-26
> **Follow-Up on Our Rebuttal Responses**
>
> Dear Reviewer,
>
> Thank you again for your careful review and insightful comments on our paper. We have now provided detailed responses and additional analyses in the rebuttal to address your concerns. If you have a moment to look over our replies and let us know whether there are any remaining points that would benefit from further clarification, we would be very grateful.
>
> Wishing you a joyful Thanksgiving! :-)
>
> Best regards,
> The Authors

---

### Author Response · Authors · 2025-11-28
**Rebuttal Summary: Clarifications and Additional Analyses for ScholScan**

Dear Area Chair and Reviewers,

As the rebuttal period draws to a close, we would like to briefly summarize the current status of our responses.

First, we sincerely thank you for the positive assessments of our work. All reviewers recognize the necessity and challenge of the proposed scan-oriented task paradigm, and acknowledge ScholScan’s strengths in data quality, annotation workflow, and the design of fine-grained evaluation metrics and analyses.

At the same time, several common concerns have emerged across the reviews, for which we have conducted additional experiments and provided detailed replies:

1. **On long-term validity under potential pre-training exposure**

We explicitly separate the benchmark into the Sampling and Generation subsets and report model performance on each subset. The results show that merely “having seen the papers or reviews” does not yield a clear advantage. We also emphasize that the construction pipeline of ScholScan is designed to be updatable, allowing future versions to be built on newer papers so that the benchmark can maintain its evaluation value over time.

2. **On the human baseline and task solvability**

We ran multiple human studies under 15/30/60-minute settings as well as under the oracle condition. The results show that, given a reasonable time budget, domain experts can achieve substantially higher scores than current MLLMs on ScholScan. Taken together, these findings indicate that the benchmark primarily exposes a significant capability gap between current MLLMs and human experts, rather than defining a task on which both humans and models perform uniformly poorly.

3. **On potential over-reliance on LLMs during data construction**

For the Sampling subset, we start from real human peer-review comments, using Gemini only for coarse filtering and initial drafting; human experts then thoroughly rewrite and verify the items. For the Generation subset, Gemini’s candidates are heavily filtered and substantially rewritten. In the final dataset, the error forms are mainly governed by our human-defined error taxonomy and annotation protocol, rather than directly inheriting the idiosyncrasies of a single LLM.

Finally, we are very grateful for the constructive interactions during the rebuttal. Reviewer HMvn has explicitly indicated an increased rating after reading our additional human-baseline and RAG analyses, and Reviewers 78Mn and co8M have both stated in their public comments that they will maintain their positive scores.

We hope that these additional analyses and clarifications help convey the intent behind ScholScan, its main conclusions, and its potential to support future research in this area.

Best Regards,

Authors

---

### Meta-Review · Area_Chair_CyQq · 2026-01-07

**Summary:**

Major concerns:

1. Dataset was created with an LLM in the loop, so its quality is questionable. (6ouw, 78Mn, a2Tc, co8M)
2. Concerns over data leakage. (6ouw, 78Mn)
3. Lack of an estimate of human performance. (HMvn)
4. Evaluation metric uses an LLM, and hence could be biased. (a2Tc)
5. Evaluation metric has many components and tunable parameters, and hence may not transfer across domains. (6ouw)

**Reviewer Concerns:**

Concern (1) seems largely addressed since the authors argue that one of the two subsets of the dataset contains more natural error types and the both the subsets have gone through human verification and filtering. The authors' argument regarding (2) is based on the comparison of performance between the two subsets, one of which is entirely public and the other is partially public. This argument is not entirely convincing, although I do agree that this is issue is extremely hard to avoid. I think the remaining major concerns were addressed well in the discussion.

**Reviewer Scores:**

Scores of HMvn, 78Mn and a2Tc would have likely gone up because at least a few of their concerns were addressed satisfactorily.

---

### Decision · Program_Chairs · 2026-01-26

Accept (Poster)